# Understanding Straight-Through Estimator in Training Activation Quantized Neural Nets

**Penghang Yin,**[*] **Jiancheng Lyu,**[†] **Shuai Zhang,**[‡] **Stanley Osher,**[*] **Yingyong Qi,**[‡] **Jack Xin**[†]

[*]Department of Mathematics, University of California, Los Angeles
 `yph@ucla.edu, sjo@math.ucla.edu`
[†]Department of Mathematics, University of California, Irvine
 `jianchel@uci.edu, jxin@math.uci.edu`
[‡]Qualcomm AI Research, San Diego
 `{shuazhan,yingyong}@qti.qualcomm.com`

## Abstract

Training activation quantized neural networks involves minimizing a piecewise constant function whose gradient vanishes almost everywhere, which is undesirable for the standard back-propagation or chain rule. An empirical way around this issue is to use a straight-through estimator (STE) (Bengio et al., 2013) in the backward pass only, so that the "gradient" through the modified chain rule becomes non-trivial. Since this unusual "gradient" is certainly not the gradient of loss function, the following question arises: *why searching in its negative direction minimizes the training loss?* In this paper, we provide the theoretical justification of the concept of STE by answering this question. We consider the problem of learning a two-linear-layer network with binarized ReLU activation and Gaussian input data. We shall refer to the unusual "gradient" given by the STE-modifed chain rule as coarse gradient. The choice of STE is not unique. We prove that if the STE is properly chosen, the expected coarse gradient correlates positively with the population gradient (not available for the training), and its negation is a descent direction for minimizing the population loss. We further show the associated coarse gradient descent algorithm converges to a critical point of the population loss minimization problem. Moreover, we show that a poor choice of STE leads to instability of the training algorithm near certain local minima, which is verified with CIFAR-10 experiments.

## 1 Introduction

Deep neural networks (DNN) have achieved the remarkable success in many machine learning applications such as computer vision (Krizhevsky et al., 2012; Ren et al., 2015), natural language processing (Collobert & Weston, 2008) and reinforcement learning (Mnih et al., 2015; Silver et al., 2016). However, the deployment of DNN typically require hundreds of megabytes of memory storage for the trainable full-precision floating-point parameters, and billions of floating-point operations to make a single inference. To achieve substantial memory savings and energy efficiency at inference time, many recent efforts have been made to the training of coarsely quantized DNN, meanwhile maintaining the performance of their float counterparts (Courbariaux et al., 2015; Rastegari et al., 2016; Cai et al., 2017; Hubara et al., 2018; Yin et al., 2018b).

Training fully quantized DNN amounts to solving a very challenging optimization problem. It calls for minimizing a piecewise constant and highly nonconvex empirical risk function $f(\boldsymbol{w})$ subject to a discrete set-constraint $\boldsymbol{w} \in \mathcal{Q}$ that characterizes the quantized weights. In particular, weight quantization of DNN have been extensively studied in the literature; see for examples (Li et al., 2016; Zhu et al., 2016; Li et al., 2017; Yin et al., 2016; 2018a; Hou & Kwok, 2018; He et al., 2018; Li & Hao, 2018). On the other hand, the gradient $\nabla f(\boldsymbol{w})$ in training activation quantized DNN is almost everywhere (a.e.) zero, which makes the standard back-propagation inapplicable. The arguably most effective way around this issue is nothing but to construct a non-trivial search direction by

properly modifying the chain rule. Specifically, one can replace the a.e. zero derivative of quantized activation function composited in the chain rule with a related surrogate. This proxy derivative used in the backward pass only is referred as the straight-through estimator (STE) (Bengio et al., 2013). In the same paper, Bengio et al. (2013) proposed an alternative approach based on stochastic neurons. In addition, Friesen & Domingos (2017) proposed the feasible target propagation algorithm for learning hard-threshold (or binary activated) networks (Lee et al., 2015) via convex combinatorial optimization.

## 1.1 RELATED WORKS

The idea of STE originates to the celebrated perceptron algorithm (Rosenblatt, 1957; 1962) in 1950s for learning single-layer perceptrons. The perceptron algorithm essentially does not calculate the "gradient" through the standard chain rule, but instead through a modified chain rule in which the derivative of identity function serves as the proxy of the original derivative of binary output function $1_{\{x>0\}}$. Its convergence has been extensive discussed in the literature; see for examples, (Widrow & Lehr, 1990; Freund & Schapire, 1999) and the references therein. Hinton (2012) extended this idea to train multi-layer networks with binary activations (a.k.a. binary neuron), namely, to back-propagate as if the activation had been the identity function. Bengio et al. (2013) proposed a STE variant which uses the derivative of the sigmoid function instead. In the training of DNN with weights and activations constrained to $\pm 1$, (Hubara et al., 2016) substituted the derivative of the signum activation function with $1_{\{|x|\leq 1\}}$ in the backward pass, known as the saturated STE. Later the idea of STE was readily employed to the training of DNN with general quantized ReLU activations (Hubara et al., 2018; Zhou et al., 2016; Cai et al., 2017; Choi et al., 2018; Yin et al., 2018b), where some other proxies took place including the derivatives of vanilla ReLU and clipped ReLU. Despite all the empirical success of STE, there is very limited theoretical understanding of it in training DNN with stair-case activations.

Goel et al. (2018) considers leaky ReLU activation of a one-hidden-layer network. They showed the convergence of the so-called Convertron algorithm, which uses the identity STE in the backward pass through the leaky ReLU layer. Other similar scenarios, where certain layers are not desirable for back-propagation, have been brought up recently by (Wang et al., 2018) and (Athalye et al., 2018). The former proposed an implicit weighted nonlocal Laplacian layer as the classifier to improve the generalization accuracy of DNN. In the backward pass, the derivative of a pre-trained fully-connected layer was used as a surrogate. To circumvent adversarial defense (Szegedy et al., 2013), (Athalye et al., 2018) introduced the backward pass differentiable approximation, which shares the same spirit as STE, and successfully broke defenses at ICLR 2018 that rely on obfuscated gradients.

## 1.2 MAIN CONTRIBUTIONS

Throughout this paper, we shall refer to the "gradient" of loss function w.r.t. the weight variables through the STE-modified chain rule as coarse gradient. Since the backward and forward passes do not match, the coarse gradient is certainly not the gradient of loss function, and it is generally not the gradient of any function. Why searching in its negative direction minimizes the training loss, as this is not the standard gradient descent algorithm? Apparently, the choice of STE is non-unique, then what makes a good STE? From the optimization perspective, we take a step towards understanding STE in training quantized ReLU nets by attempting these questions.

On the theoretical side, we consider three representative STEs for learning a two-linear-layer network with binary activation and Gaussian data: the derivatives of the identity function (Rosenblatt, 1957; Hinton, 2012; Goel et al., 2018), vanilla ReLU and the clipped ReLUs (Cai et al., 2017; Hubara et al., 2016). We adopt the model of population loss minimization (Brutzkus & Globerson, 2017; Tian, 2017; Li & Yuan, 2017; Du et al., 2018). *For the first time, we prove that proper choices of STE give rise to training algorithms that are descent.* Specifically, the negative expected coarse gradients based on STEs of the vanilla and clipped ReLUs are provably descent directions for the minimizing the population loss, which yield monotonically decreasing energy in the training. In contrast, this is not true for the identity STE. We further prove that the corresponding training algorithm can be *unstable near certain local minima, because the coarse gradient may simply not vanish there.*

Complementary to the analysis, we examine the empirical performances of the three STEs on MNIST and CIFAR-10 classifications with general quantized ReLU. While both vanilla and clipped ReLUs work very well on the relatively shallow LeNet-5, clipped ReLU STE is arguably the best for the deeper VGG-11 and ResNet-20. In our CIFAR experiments in section 4.2, we observe that the training using identity or ReLU STE can be unstable at good minima and repelled to an inferior one with substantially higher training loss and decreased generalization accuracy. This is an implication that *poor STEs generate coarse gradients incompatible with the energy landscape, which is consistent with our theoretical finding about the identity STE.*

To our knowledge, convergence guarantees of perceptron algorithm (Rosenblatt, 1957; 1962) and Convertron algorithm (Goel et al., 2018) were proved for the identity STE. It is worth noting that Convertron (Goel et al., 2018) makes weaker assumptions than in this paper. These results, however, do not generalize to the network with two trainable layers studied here. As aforementioned, the identity STE is actually a poor choice in our case. Moreover, it is not clear if their analyses can be extended to other STEs. Similar to Convertron with leaky ReLU, the monotonicity of quantized activation function plays a role in coarse gradient descent. Indeed, all three STEs considered here exploit this property. But this is not the whole story. A great STE like the clipped ReLU matches quantized ReLU at the extrema, otherwise the instability/incompatibility issue may arise.

**Organization.** In section 2, we study the energy landscape of a two-linear-layer network with binary activation and Gaussian data. We present the main results and sketch the mathematical analysis for STE in section 3. In section 4, we compare the empirical performances of different STEs in 2-bit and 4-bit activation quantization, and report the instability phenomena of the training algorithms associated with poor STEs observed in CIFAR experiments. Due to space limitation, all the technical proofs as well as some figures are deferred to the appendix.

**Notations.** $\| \cdot \|$ denotes the Euclidean norm of a vector or the spectral norm of a matrix. $\mathbf{0}_n \in \mathbb{R}^n$ represents the vector of all zeros, whereas $\mathbf{1}_n \in \mathbb{R}^n$ the vector of all ones. $\boldsymbol{I}_n$ is the identity matrix of order $n$. For any $\boldsymbol{w}, \mathbf{z} \in \mathbb{R}^n$, $\boldsymbol{w}^\top \mathbf{z} = \langle \boldsymbol{w}, \mathbf{z} \rangle = \sum_i w_i z_i$ is their inner product. $\boldsymbol{w} \odot \mathbf{z}$ denotes the Hadamard product whose $i^{\text{th}}$ entry is given by $(\boldsymbol{w} \odot \mathbf{z})_i = w_i z_i$.

## 2 LEARNING TWO-LINEAR-LAYER CNN WITH BINARY ACTIVATION

We consider a model similar to (Du et al., 2018) that outputs the prediction

$$y(\mathbf{Z}, \boldsymbol{v}, \boldsymbol{w}) := \sum_{i=1}^{m} v_i \sigma(\mathbf{Z}_i^\top \boldsymbol{w}) = \boldsymbol{v}^\top \sigma(\mathbf{Z}\boldsymbol{w})$$

for some input $\mathbf{Z} \in \mathbb{R}^{m \times n}$. Here $\boldsymbol{w} \in \mathbb{R}^n$ and $\boldsymbol{v} \in \mathbb{R}^m$ are the trainable weights in the first and second linear layer, respectively; $\mathbf{Z}_i^\top$ denotes the $i$th row vector of $\mathbf{Z}$; the activation function $\sigma$ acts component-wise on the vector $\mathbf{Z}\boldsymbol{w}$, i.e., $\sigma(\mathbf{Z}\boldsymbol{w})_i = \sigma((\mathbf{Z}\boldsymbol{w})_i) = \sigma(\mathbf{Z}_i^\top \boldsymbol{w})$. The first layer serves as a convolutional layer, where each row $\mathbf{Z}_i^\top$ can be viewed as a patch sampled from $\mathbf{Z}$ and the weight filter $\boldsymbol{w}$ is shared among all patches, and the second linear layer is the classifier. The label is generated according to $y^*(\mathbf{Z}) = (\boldsymbol{v}^*)^\top \sigma(\mathbf{Z}\boldsymbol{w}^*)$ for some true (non-zero) parameters $\boldsymbol{v}^*$ and $\boldsymbol{w}^*$. Moreover, we use the following squared sample loss

$$\ell(\boldsymbol{v}, \boldsymbol{w}; \mathbf{Z}) := \frac{1}{2} \left( y(\mathbf{Z}, \boldsymbol{v}, \boldsymbol{w}) - y^*(\mathbf{Z}) \right)^2 = \frac{1}{2} \left( \boldsymbol{v}^\top \sigma(\mathbf{Z}\boldsymbol{w}) - y^*(\mathbf{Z}) \right)^2. \tag{1}$$

Unlike in (Du et al., 2018), the activation function $\sigma$ here is not ReLU, but the binary function $\sigma(x) = 1_{\{x > 0\}}$.

We assume that the entries of $\mathbf{Z} \in \mathbb{R}^{m \times n}$ are i.i.d. sampled from the Gaussian distribution $\mathcal{N}(0, 1)$ (Zhong et al., 2017; Brutzkus & Globerson, 2017). Since $\ell(\boldsymbol{v}, \boldsymbol{w}; \mathbf{Z}) = \ell(\boldsymbol{v}, \boldsymbol{w}/c; \mathbf{Z})$ for any scalar $c > 0$, without loss of generality, we take $\|\boldsymbol{w}^*\| = 1$ and cast the learning task as the following population loss minimization problem:

$$\min_{\boldsymbol{v} \in \mathbb{R}^m, \boldsymbol{w} \in \mathbb{R}^n} f(\boldsymbol{v}, \boldsymbol{w}) := \mathbb{E}_{\mathbf{Z}} \left[ \ell(\boldsymbol{v}, \boldsymbol{w}; \mathbf{Z}) \right], \tag{2}$$

where the sample loss $\ell(\boldsymbol{v}, \boldsymbol{w}; \mathbf{Z})$ is given by (1).

## 2.1 BACK-PROPAGATION AND COARSE GRADIENT DESCENT

With the Gaussian assumption on $\mathbf{Z}$, as will be shown in section 2.2, it is possible to find the analytic expressions of $f(\boldsymbol{v}, \boldsymbol{w})$ and its gradient

$$\nabla f(\boldsymbol{v}, \boldsymbol{w}) := \begin{bmatrix} \frac{\partial f}{\partial \boldsymbol{v}}(\boldsymbol{v}, \boldsymbol{w}) \\ \frac{\partial f}{\partial \boldsymbol{w}}(\boldsymbol{v}, \boldsymbol{w}) \end{bmatrix}.$$

The gradient of objective function, however, is not available for the network training. In fact, we can only access the expected sample gradient, namely,

$$\mathbb{E}_{\mathbf{Z}}\left[\frac{\partial \ell}{\partial \boldsymbol{v}}(\boldsymbol{v}, \boldsymbol{w}; \mathbf{Z})\right] \text{ and } \mathbb{E}_{\mathbf{Z}}\left[\frac{\partial \ell}{\partial \boldsymbol{w}}(\boldsymbol{v}, \boldsymbol{w}; \mathbf{Z})\right].$$

We remark that $\mathbb{E}_{\mathbf{Z}}\left[\frac{\partial \ell}{\partial \boldsymbol{w}}(\boldsymbol{v}, \boldsymbol{w}; \mathbf{Z})\right]$ is not the same as $\frac{\partial f}{\partial \boldsymbol{w}}(\boldsymbol{v}, \boldsymbol{w}) = \frac{\partial \mathbb{E}_{\mathbf{Z}}[\ell(\boldsymbol{v}, \boldsymbol{w}; \mathbf{Z})]}{\partial \boldsymbol{w}}$. By the standard back-propagation or chain rule, we readily check that

$$\frac{\partial \ell}{\partial \boldsymbol{v}}(\boldsymbol{v}, \boldsymbol{w}; \mathbf{Z}) = \sigma(\mathbf{Z}\boldsymbol{w})\left(\boldsymbol{v}^\top \sigma(\mathbf{Z}\boldsymbol{w}) - y^*(\mathbf{Z})\right) \tag{3}$$

and

$$\frac{\partial \ell}{\partial \boldsymbol{w}}(\boldsymbol{v}, \boldsymbol{w}; \mathbf{Z}) = \mathbf{Z}^\top\left(\sigma'(\mathbf{Z}\boldsymbol{w}) \odot \boldsymbol{v}\right)\left(\boldsymbol{v}^\top \sigma(\mathbf{Z}\boldsymbol{w}) - y^*(\mathbf{Z})\right). \tag{4}$$

Note that $\sigma'$ is zero a.e., which makes (4) inapplicable to the training. The idea of STE is to simply replace the a.e. zero component $\sigma'$ in (4) with a related non-trivial function $\mu'$ (Hinton, 2012; Bengio et al., 2013; Hubara et al., 2016; Cai et al., 2017), which is the derivative of some (sub)differentiable function $\mu$. More precisely, back-propagation using the STE $\mu'$ gives the following non-trivial surrogate of $\frac{\partial \ell}{\partial \boldsymbol{w}}(\boldsymbol{v}, \boldsymbol{w}; \mathbf{Z})$, to which we refer as the coarse (partial) gradient

$$\boldsymbol{g}_\mu(\boldsymbol{v}, \boldsymbol{w}; \mathbf{Z}) = \mathbf{Z}^\top\left(\mu'(\mathbf{Z}\boldsymbol{w}) \odot \boldsymbol{v}\right)\left(\boldsymbol{v}^\top \sigma(\mathbf{Z}\boldsymbol{w}) - y^*(\mathbf{Z})\right). \tag{5}$$

Using the STE $\mu'$ to train the two-linear-layer convolutional neural network (CNN) with binary activation gives rise to the (full-batch) coarse gradient descent described in Algorithm 1.

---

**Algorithm 1** Coarse gradient descent for learning two-linear-layer CNN with STE $\mu'$.

---

**Input**: initialization $\boldsymbol{v}^0 \in \mathbb{R}^m$, $\boldsymbol{w}^0 \in \mathbb{R}^n$, learning rate $\eta$.

    **for** $t = 0, 1, \ldots$ **do**
      $\boldsymbol{v}^{t+1} = \boldsymbol{v}^t - \eta\,\mathbb{E}_{\mathbf{Z}}\left[\frac{\partial \ell}{\partial \boldsymbol{v}}(\boldsymbol{v}^t, \boldsymbol{w}^t; \mathbf{Z})\right]$
      $\boldsymbol{w}^{t+1} = \boldsymbol{w}^t - \eta\,\mathbb{E}_{\mathbf{Z}}\left[\boldsymbol{g}_\mu(\boldsymbol{v}^t, \boldsymbol{w}^t; \mathbf{Z})\right]$
    **end for**

---

## 2.2 PRELIMINARIES

Let us present some preliminaries about the landscape of the population loss function $f(\boldsymbol{v}, \boldsymbol{w})$. To this end, we define the angle between $\boldsymbol{w}$ and $\boldsymbol{w}^*$ as $\theta(\boldsymbol{w}, \boldsymbol{w}^*) := \arccos\left(\frac{\boldsymbol{w}^\top \boldsymbol{w}^*}{\|\boldsymbol{w}\|\|\boldsymbol{w}^*\|}\right)$ for any $\boldsymbol{w} \neq \mathbf{0}_n$. Recall that the label is given by $y^*(\mathbf{Z}) = (\boldsymbol{v}^*)^\top \mathbf{Z}\boldsymbol{w}^*$ from (1), we elaborate on the analytic expressions of $f(\boldsymbol{v}, \boldsymbol{w})$ and $\nabla f(\boldsymbol{v}, \boldsymbol{w})$.

**Lemma 1.** *If $\boldsymbol{w} \neq \mathbf{0}_n$, the population loss $f(\boldsymbol{v}, \boldsymbol{w})$ is given by*

$$\frac{1}{8}\left[\boldsymbol{v}^\top\left(\boldsymbol{I}_m + \mathbf{1}_m\mathbf{1}_m^\top\right)\boldsymbol{v} - 2\boldsymbol{v}^\top\left(\left(1 - \frac{2}{\pi}\theta(\boldsymbol{w}, \boldsymbol{w}^*)\right)\boldsymbol{I}_m + \mathbf{1}_m\mathbf{1}_m^\top\right)\boldsymbol{v}^* + (\boldsymbol{v}^*)^\top\left(\boldsymbol{I}_m + \mathbf{1}_m\mathbf{1}_m^\top\right)\boldsymbol{v}^*\right].$$

*In addition, $f(\boldsymbol{v}, \boldsymbol{w}) = \frac{1}{8}(\boldsymbol{v}^*)^\top\left(\boldsymbol{I}_m + \mathbf{1}_m\mathbf{1}_m^\top\right)\boldsymbol{v}^*$ for $\boldsymbol{w} = \mathbf{0}_n$.*

**Lemma 2.** *If $\boldsymbol{w} \neq \mathbf{0}_n$ and $\theta(\boldsymbol{w}, \boldsymbol{w}^*) \in (0, \pi)$, the partial gradients of $f(\boldsymbol{v}, \boldsymbol{w})$ w.r.t. $\boldsymbol{v}$ and $\boldsymbol{w}$ are*

$$\frac{\partial f}{\partial \boldsymbol{v}}(\boldsymbol{v}, \boldsymbol{w}) = \frac{1}{4}\left(\boldsymbol{I}_m + \mathbf{1}_m\mathbf{1}_m^\top\right)\boldsymbol{v} - \frac{1}{4}\left(\left(1 - \frac{2}{\pi}\theta(\boldsymbol{w}, \boldsymbol{w}^*)\right)\boldsymbol{I}_m + \mathbf{1}_m\mathbf{1}_m^\top\right)\boldsymbol{v}^* \tag{6}$$

*and*

$$\frac{\partial f}{\partial \boldsymbol{w}}(\boldsymbol{v}, \boldsymbol{w}) = -\frac{\boldsymbol{v}^\top \boldsymbol{v}^*}{2\pi \|\boldsymbol{w}\|} \frac{\left(\boldsymbol{I}_n - \frac{\boldsymbol{w}\boldsymbol{w}^\top}{\|\boldsymbol{w}\|^2}\right)\boldsymbol{w}^*}{\left\|\left(\boldsymbol{I}_n - \frac{\boldsymbol{w}\boldsymbol{w}^\top}{\|\boldsymbol{w}\|^2}\right)\boldsymbol{w}^*\right\|}, \tag{7}$$

*respectively.*

For any $\boldsymbol{v} \in \mathbb{R}^m$, $(\boldsymbol{v}, \boldsymbol{0}_m)$ is impossible to be a local minimizer. The only possible (local) minimizers of the model (2) are located at

1. Stationary points where the gradients given by (6) and (7) vanish simultaneously (which may not be possible), i.e.,

$$\boldsymbol{v}^\top \boldsymbol{v}^* = 0 \text{ and } \boldsymbol{v} = \left(\boldsymbol{I}_m + \boldsymbol{1}_m \boldsymbol{1}_m^\top\right)^{-1}\left(\left(1 - \frac{2}{\pi}\theta(\boldsymbol{w}, \boldsymbol{w}^*)\right)\boldsymbol{I}_m + \boldsymbol{1}_m \boldsymbol{1}_m^\top\right)\boldsymbol{v}^*. \tag{8}$$

2. Non-differentiable points where $\theta(\boldsymbol{w}, \boldsymbol{w}^*) = 0$ and $\boldsymbol{v} = \boldsymbol{v}^*$, or $\theta(\boldsymbol{w}, \boldsymbol{w}^*) = \pi$ and $\boldsymbol{v} = \left(\boldsymbol{I}_m + \boldsymbol{1}_m \boldsymbol{1}_m^\top\right)^{-1}(\boldsymbol{1}_m \boldsymbol{1}_m^\top - \boldsymbol{I}_m)\boldsymbol{v}^*$.

Among them, $\{(\boldsymbol{v}, \boldsymbol{w}) : \boldsymbol{v} = \boldsymbol{v}^*, \theta(\boldsymbol{w}, \boldsymbol{w}^*) = 0\}$ are obviously the global minimizers of (2). We show that the stationary points, if exist, can only be saddle points, and $\{(\boldsymbol{v}, \boldsymbol{w}) : \theta(\boldsymbol{w}, \boldsymbol{w}^*) = \pi, \ \boldsymbol{v} = \left(\boldsymbol{I}_m + \boldsymbol{1}_m \boldsymbol{1}_m^\top\right)^{-1}(\boldsymbol{1}_m \boldsymbol{1}_m^\top - \boldsymbol{I}_m)\boldsymbol{v}^*\}$ are the only potential spurious local minimizers.

**Proposition 1.** *If the true parameter $\boldsymbol{v}^*$ satisfies $(\boldsymbol{1}_m^\top \boldsymbol{v}^*)^2 < \frac{m+1}{2}\|\boldsymbol{v}^*\|^2$, then*

$$\left\{(\boldsymbol{v}, \boldsymbol{w}) : \boldsymbol{v} = (\boldsymbol{I}_m + \boldsymbol{1}_m \boldsymbol{1}_m^\top)^{-1}\left(\frac{-(\boldsymbol{1}_m^\top \boldsymbol{v}^*)^2}{(m+1)\|\boldsymbol{v}^*\|^2 - (\boldsymbol{1}_m^\top \boldsymbol{v}^*)^2}\boldsymbol{I}_m + \boldsymbol{1}_m \boldsymbol{1}_m^\top\right)\boldsymbol{v}^*,$$

$$\theta(\boldsymbol{w}, \boldsymbol{w}^*) = \frac{\pi}{2}\frac{(m+1)\|\boldsymbol{v}^*\|^2}{(m+1)\|\boldsymbol{v}^*\|^2 - (\boldsymbol{1}_m^\top \boldsymbol{v}^*)^2}\right\} \tag{9}$$

*give the saddle points obeying (8), and $\{(\boldsymbol{v}, \boldsymbol{w}) : \theta(\boldsymbol{w}, \boldsymbol{w}^*) = \pi, \ \boldsymbol{v} = \left(\boldsymbol{I}_m + \boldsymbol{1}_m \boldsymbol{1}_m^\top\right)^{-1}(\boldsymbol{1}_m \boldsymbol{1}_m^\top - \boldsymbol{I}_m)\boldsymbol{v}^*\}$ are the spurious local minimizers. Otherwise, the model (2) has no saddle points or spurious local minimizers.*

We further prove that the population gradient $\nabla f(\boldsymbol{v}, \boldsymbol{w})$ given by (6) and (7), is Lipschitz continuous when restricted to bounded domains.

**Lemma 3.** *For any differentiable points $(\boldsymbol{v}, \boldsymbol{w})$ and $(\tilde{\boldsymbol{v}}, \tilde{\boldsymbol{w}})$ with $\min\{\|\boldsymbol{w}\|, \|\tilde{\boldsymbol{w}}\|\} = c_{\boldsymbol{w}} > 0$ and $\max\{\|\boldsymbol{v}\|, \|\tilde{\boldsymbol{v}}\|\} = C_{\boldsymbol{v}}$, there exists a Lipschitz constant $L > 0$ depending on $C_{\boldsymbol{v}}$ and $c_{\boldsymbol{w}}$, such that*

$$\|\nabla f(\boldsymbol{v}, \boldsymbol{w}) - \nabla f(\tilde{\boldsymbol{v}}, \tilde{\boldsymbol{w}})\| \le L\|(\boldsymbol{v}, \boldsymbol{w}) - (\tilde{\boldsymbol{v}}, \tilde{\boldsymbol{w}})\|.$$

## 3 MAIN RESULTS

We are most interested in the complex case where both the saddle points and spurious local minimizers are present. Our main results are concerned with the behaviors of the coarse gradient descent summarized in Algorithm 1 when the derivatives of the vanilla and clipped ReLUs as well as the identity function serve as the STE, respectively. We shall prove that Algorithm 1 using the derivative of vanilla or clipped ReLU converges to a critical point, whereas that with the identity STE does not.

**Theorem 1** (Convergence). *Let $\{(\boldsymbol{v}^t, \boldsymbol{w}^t)\}$ be the sequence generated by Algorithm 1 with ReLU $\mu(x) = \max\{x, 0\}$ or clipped ReLU $\mu(x) = \min\{\max\{x, 0\}, 1\}$. Suppose $\|\boldsymbol{w}^t\| \ge c_{\boldsymbol{w}}$ for all $t$ with some $c_{\boldsymbol{w}} > 0$. Then if the learning rate $\eta > 0$ is sufficiently small, for any initialization $(\boldsymbol{v}^0, \boldsymbol{w}^0)$, the objective sequence $\{f(\boldsymbol{v}^t, \boldsymbol{w}^t)\}$ is monotonically decreasing, and $\{(\boldsymbol{v}^t, \boldsymbol{w}^t)\}$ converges to a saddle point or a (local) minimizer of the population loss minimization (2). In addition, if $\boldsymbol{1}_m^\top \boldsymbol{v}^* \ne 0$ and $m > 1$, the descent and convergence properties do not hold for Algorithm 1 with the identity function $\mu(x) = x$ near the local minimizers satisfying $\theta(\boldsymbol{w}, \boldsymbol{w}^*) = \pi$ and $\boldsymbol{v} = (\boldsymbol{I}_m + \boldsymbol{1}_m \boldsymbol{1}_m^\top)^{-1}(\boldsymbol{1}_m \boldsymbol{1}_m^\top - \boldsymbol{I}_m)\boldsymbol{v}^*$.*

**Remark 1.** *The convergence guarantee for the coarse gradient descent is established under the assumption that there are infinite training samples. When there are only a few data, in a coarse scale, the empirical loss roughly descends along the direction of negative coarse gradient, as illustrated by Figure 1. As the sample size increases, the empirical loss gains monotonicity and smoothness. This explains why (proper) STE works so well with massive amounts of data as in deep learning.*

**Remark 2.** *The same results hold, if the Gaussian assumption on the input data is weakened to that their rows i.i.d. follow some rotation-invariant distribution. The proof will be substantially similar.*

In the rest of this section, we sketch the mathematical analysis for the main results.

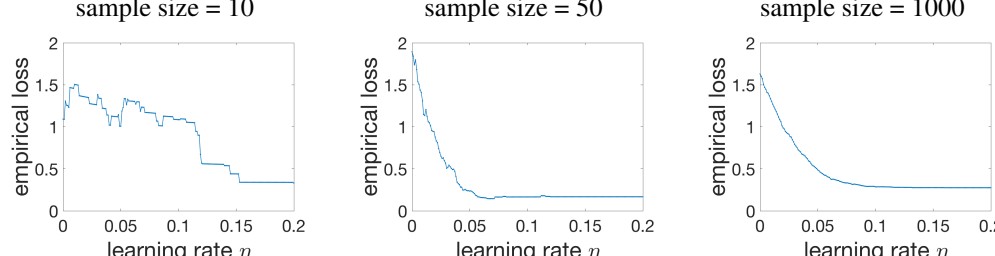

Figure 1: The plots of the empirical loss moving by one step in the direction of negative coarse gradient v.s. the learning rate (step size) $\eta$ for different sample sizes.

### 3.1 DERIVATIVE OF THE VANILLA RELU AS STE

If we choose the derivative of ReLU $\mu(x) = \max\{x, 0\}$ as the STE in (5), it is easy to see $\mu'(x) = \sigma(x)$, and we have the following expressions of $\mathbb{E}_{\mathbf{Z}}\left[\frac{\partial \ell}{\partial \boldsymbol{v}}(\boldsymbol{v}, \boldsymbol{w}; \mathbf{Z})\right]$ and $\mathbb{E}_{\mathbf{Z}}\left[\boldsymbol{g}_{\text{relu}}(\boldsymbol{v}, \boldsymbol{w}; \mathbf{Z})\right]$ for Algorithm 1.

**Lemma 4.** *The expected partial gradient of $\ell(\boldsymbol{v}, \boldsymbol{w}; \mathbf{Z})$ w.r.t. $\boldsymbol{v}$ is*

$$\mathbb{E}_{\mathbf{Z}}\left[\frac{\partial \ell}{\partial \boldsymbol{v}}(\boldsymbol{v}, \boldsymbol{w}; \mathbf{Z})\right] = \frac{\partial f}{\partial \boldsymbol{v}}(\boldsymbol{v}, \boldsymbol{w}). \tag{10}$$

*Let $\mu(x) = \max\{x, 0\}$ in (5). The expected coarse gradient w.r.t. $\boldsymbol{w}$ is*

$$\mathbb{E}_{\mathbf{Z}}\left[\boldsymbol{g}_{\text{relu}}(\boldsymbol{v}, \boldsymbol{w}; \mathbf{Z})\right] = \frac{h(\boldsymbol{v}, \boldsymbol{v}^*)}{2\sqrt{2\pi}}\frac{\boldsymbol{w}}{\|\boldsymbol{w}\|} - \cos\left(\frac{\theta(\boldsymbol{w}, \boldsymbol{w}^*)}{2}\right)\frac{\boldsymbol{v}^\top \boldsymbol{v}^*}{\sqrt{2\pi}}\frac{\frac{\boldsymbol{w}}{\|\boldsymbol{w}\|} + \boldsymbol{w}^*}{\left\|\frac{\boldsymbol{w}}{\|\boldsymbol{w}\|} + \boldsymbol{w}^*\right\|}, {}^1 \tag{11}$$

*where $h(\boldsymbol{v}, \boldsymbol{v}^*) = \|\boldsymbol{v}\|^2 + (\mathbf{1}_m^\top \boldsymbol{v})^2 - (\mathbf{1}_m^\top \boldsymbol{v})(\mathbf{1}_m^\top \boldsymbol{v}^*) + \boldsymbol{v}^\top \boldsymbol{v}^*$.*

As stated in Lemma 5 below, the key observation is that the coarse partial gradient $\mathbb{E}_{\mathbf{Z}}\left[\boldsymbol{g}_{\text{relu}}(\boldsymbol{v}, \boldsymbol{w}; \mathbf{Z})\right]$ has non-negative correlation with the population partial gradient $\frac{\partial f}{\partial \boldsymbol{w}}(\boldsymbol{v}, \boldsymbol{w})$, and $-\mathbb{E}_{\mathbf{Z}}\left[\boldsymbol{g}_{\text{relu}}(\boldsymbol{v}, \boldsymbol{w}; \mathbf{Z})\right]$ together with $-\mathbb{E}_{\mathbf{Z}}\left[\frac{\partial \ell}{\partial \boldsymbol{v}}(\boldsymbol{v}, \boldsymbol{w}; \mathbf{Z})\right]$ form a descent direction for minimizing the population loss.

**Lemma 5.** *If $\boldsymbol{w} \neq \mathbf{0}_n$ and $\theta(\boldsymbol{w}, \boldsymbol{w}^*) \in (0, \pi)$, then the inner product between the expected coarse and population gradients w.r.t. $\boldsymbol{w}$ is*

$$\left\langle \mathbb{E}_{\mathbf{Z}}\left[\boldsymbol{g}_{\text{relu}}(\boldsymbol{v}, \boldsymbol{w}; \mathbf{Z})\right], \frac{\partial f}{\partial \boldsymbol{w}}(\boldsymbol{v}, \boldsymbol{w})\right\rangle = \frac{\sin(\theta(\boldsymbol{w}, \boldsymbol{w}^*))}{2(\sqrt{2\pi})^3 \|\boldsymbol{w}\|}(\boldsymbol{v}^\top \boldsymbol{v}^*)^2 \geq 0.$$

*Moreover, if further $\|\boldsymbol{v}\| \leq C_{\boldsymbol{v}}$ and $\|\boldsymbol{w}\| \geq c_{\boldsymbol{w}}$, there exists a constant $A_{\text{relu}} > 0$ depending on $C_{\boldsymbol{v}}$ and $c_{\boldsymbol{w}}$, such that*

$$\left\|\mathbb{E}_{\mathbf{Z}}\left[\boldsymbol{g}_{\text{relu}}(\boldsymbol{v}, \boldsymbol{w}; \mathbf{Z})\right]\right\|^2 \leq A_{\text{relu}}\left(\left\|\frac{\partial f}{\partial \boldsymbol{v}}(\boldsymbol{v}, \boldsymbol{w})\right\|^2 + \left\langle \mathbb{E}_{\mathbf{Z}}\left[\boldsymbol{g}_{\text{relu}}(\boldsymbol{v}, \boldsymbol{w}; \mathbf{Z})\right], \frac{\partial f}{\partial \boldsymbol{w}}(\boldsymbol{v}, \boldsymbol{w})\right\rangle\right). \tag{12}$$

Clearly, when $\left\langle \mathbb{E}_{\mathbf{Z}}\left[\boldsymbol{g}_{\text{relu}}(\boldsymbol{v}, \boldsymbol{w}; \mathbf{Z})\right], \frac{\partial f}{\partial \boldsymbol{w}}(\boldsymbol{v}, \boldsymbol{w})\right\rangle > 0$, $\mathbb{E}_{\mathbf{Z}}\left[\boldsymbol{g}_{\text{relu}}(\boldsymbol{v}, \boldsymbol{w}; \mathbf{Z})\right]$ is roughly in the same direction as $\frac{\partial f}{\partial \boldsymbol{w}}(\boldsymbol{v}, \boldsymbol{w})$. Moreover, since by Lemma 4, $\mathbb{E}_{\mathbf{Z}}\left[\frac{\partial \ell}{\partial \boldsymbol{v}}(\boldsymbol{v}, \boldsymbol{w}; \mathbf{Z})\right] = \frac{\partial f}{\partial \boldsymbol{v}}(\boldsymbol{v}, \boldsymbol{w})$, we expect

---

[1]We redefine the second term as $\mathbf{0}_n$ in the case $\theta(\boldsymbol{w}, \boldsymbol{w}^*) = \pi$, or equivalently, $\frac{\boldsymbol{w}}{\|\boldsymbol{w}\|} + \boldsymbol{w}^* = \mathbf{0}_n$.

that the coarse gradient descent behaves like the gradient descent directly on $f(\boldsymbol{v}, \boldsymbol{w})$. Here we would like to highlight the significance of the estimate (12) in guaranteeing the descent property of Algorithm 1. By the Lipschitz continuity of $\nabla f$ specified in Lemma 3, it holds that

$$
\begin{aligned}
f(\boldsymbol{v}^{t+1}, \boldsymbol{w}^{t+1}) - f(\boldsymbol{v}^t, \boldsymbol{w}^t) \leq{} & \left\langle \frac{\partial f}{\partial \boldsymbol{v}}(\boldsymbol{v}^t, \boldsymbol{w}^t), \boldsymbol{v}^{t+1} - \boldsymbol{v}^t \right\rangle + \left\langle \frac{\partial f}{\partial \boldsymbol{w}}(\boldsymbol{v}^t, \boldsymbol{w}^t), \boldsymbol{w}^{t+1} - \boldsymbol{w}^t \right\rangle \\
& + \frac{L}{2}(\|\boldsymbol{v}^{t+1} - \boldsymbol{v}^t\|^2 + \|\boldsymbol{w}^{t+1} - \boldsymbol{w}^t\|^2) \\
={} & -\left(\eta - \frac{L\eta^2}{2}\right) \left\| \frac{\partial f}{\partial \boldsymbol{v}}(\boldsymbol{v}^t, \boldsymbol{w}^t) \right\|^2 + \frac{L\eta^2}{2} \left\| \mathbb{E}_{\mathbf{Z}}\left[\boldsymbol{g}_{\mathrm{relu}}(\boldsymbol{v}^t, \boldsymbol{w}^t; \mathbf{Z})\right] \right\|^2 \\
& -\eta \left\langle \frac{\partial f}{\partial \boldsymbol{w}}(\boldsymbol{v}^t, \boldsymbol{w}^t), \mathbb{E}_{\mathbf{Z}}\left[\boldsymbol{g}_{\mathrm{relu}}(\boldsymbol{v}^t, \boldsymbol{w}^t; \mathbf{Z})\right] \right\rangle \\
\overset{a)}{\leq}{} & -\left(\eta - (1 + A_{\mathrm{relu}})\frac{L\eta^2}{2}\right) \left\| \frac{\partial f}{\partial \boldsymbol{v}}(\boldsymbol{v}^t, \boldsymbol{w}^t) \right\|^2 \\
& -\left(\eta - \frac{A_{\mathrm{relu}}L\eta^2}{2}\right) \left\langle \frac{\partial f}{\partial \boldsymbol{w}}(\boldsymbol{v}^t, \boldsymbol{w}^t), \mathbb{E}_{\mathbf{Z}}\left[\boldsymbol{g}_{\mathrm{relu}}(\boldsymbol{v}^t, \boldsymbol{w}^t; \mathbf{Z})\right] \right\rangle,
\end{aligned}
\tag{13}
$$

where a) is due to (12). Therefore, if $\eta$ is small enough, we have monotonically decreasing energy until convergence.

**Lemma 6.** *When Algorithm 1 converges, $\mathbb{E}_{\mathbf{Z}}\left[\frac{\partial \ell}{\partial \boldsymbol{v}}(\boldsymbol{v}, \boldsymbol{w}; \mathbf{Z})\right]$ and $\mathbb{E}_{\mathbf{Z}}\left[\boldsymbol{g}_{\mathrm{relu}}(\boldsymbol{v}, \boldsymbol{w}; \mathbf{Z})\right]$ vanish simultaneously, which only occurs at the*

1. *Saddle points where (8) is satisfied according to Proposition 1.*

2. *Minimizers of (2) where $\boldsymbol{v} = \boldsymbol{v}^*$, $\theta(\boldsymbol{w}, \boldsymbol{w}^*) = 0$, or $\boldsymbol{v} = (\boldsymbol{I}_m + \mathbf{1}_m \mathbf{1}_m^\top)^{-1}(\mathbf{1}_m \mathbf{1}_m^\top - \boldsymbol{I}_m)\boldsymbol{v}^*$, $\theta(\boldsymbol{w}, \boldsymbol{w}^*) = \pi$.*

Lemma 6 states that when Algorithm 1 using ReLU STE converges, it can only converge to a critical point of the population loss function.

## 3.2 DERIVATIVE OF THE CLIPPED RELU AS STE

For the STE using clipped ReLU, $\mu(x) = \min\{\max\{x, 0\}, 1\}$ and $\mu'(x) = 1_{\{0<x<1\}}(x)$. We have results similar to Lemmas 5 and 6. That is, the coarse partial gradient using clipped ReLU STE $\mathbb{E}_{\mathbf{Z}}\left[\boldsymbol{g}_{\mathrm{crelu}}(\boldsymbol{v}, \boldsymbol{w}; \mathbf{Z})\right]$ generally has positive correlation with the true partial gradient of the population loss $\frac{\partial f}{\partial \boldsymbol{w}}(\boldsymbol{v}, \boldsymbol{w})$ (Lemma 7)). Moreover, the coarse gradient vanishes and only vanishes at the critical points (Lemma 8).

**Lemma 7.** *If $\boldsymbol{w} \neq \mathbf{0}_n$ and $\theta(\boldsymbol{w}, \boldsymbol{w}^*) \in (0, \pi)$, then*

$$
\begin{aligned}
\mathbb{E}_{\mathbf{Z}}\left[\boldsymbol{g}_{\mathrm{crelu}}(\boldsymbol{v}, \boldsymbol{w}; \mathbf{Z})\right] ={} & \frac{p(0, \boldsymbol{w})h(\boldsymbol{v}, \boldsymbol{v}^*)}{2} \frac{\boldsymbol{w}}{\|\boldsymbol{w}\|} - (\boldsymbol{v}^\top \boldsymbol{v}^*)\csc(\theta/2) \cdot q(\theta, \boldsymbol{w}) \frac{\frac{\boldsymbol{w}}{\|\boldsymbol{w}\|} + \boldsymbol{w}^*}{\left\|\frac{\boldsymbol{w}}{\|\boldsymbol{w}\|} + \boldsymbol{w}^*\right\|} \\
& - (\boldsymbol{v}^\top \boldsymbol{v}^*)\left(p(\theta, \boldsymbol{w}) - \cot(\theta/2) \cdot q(\theta, \boldsymbol{w})\right) \frac{\boldsymbol{w}}{\|\boldsymbol{w}\|},
\end{aligned}
$$

*where $h(\boldsymbol{v}, \boldsymbol{v}^*) := \|\boldsymbol{v}\|^2 + (\mathbf{1}_m^\top \boldsymbol{v})^2 - (\mathbf{1}_m^\top \boldsymbol{v})(\mathbf{1}_m^\top \boldsymbol{v}^*) + \boldsymbol{v}^\top \boldsymbol{v}^*$ same as in Lemma 5, and*

$$
p(\theta, \boldsymbol{w}) := \frac{1}{2\pi} \int_{-\frac{\pi}{2}+\theta}^{\frac{\pi}{2}} \cos(\phi)\xi\left(\frac{\sec(\phi)}{\|\boldsymbol{w}\|}\right) \mathrm{d}\phi, \quad q(\theta, \boldsymbol{w}) := \frac{1}{2\pi} \int_{-\frac{\pi}{2}+\theta}^{\frac{\pi}{2}} \sin(\phi)\xi\left(\frac{\sec(\phi)}{\|\boldsymbol{w}\|}\right) \mathrm{d}\phi
$$

*with $\xi(x) := \int_0^x r^2 \exp(-\frac{r^2}{2})\mathrm{d}r$. The inner product between the expected coarse and true gradients w.r.t. $\boldsymbol{w}$*

$$
\left\langle \mathbb{E}_{\mathbf{Z}}\left[\boldsymbol{g}_{\mathrm{crelu}}(\boldsymbol{v}, \boldsymbol{w}; \mathbf{Z})\right], \frac{\partial f}{\partial \boldsymbol{w}}(\boldsymbol{v}, \boldsymbol{w}) \right\rangle = \frac{q(\theta, \boldsymbol{w})}{2\pi\|\boldsymbol{w}\|}(\boldsymbol{v}^\top \boldsymbol{v}^*)^2 \geq 0.
$$

*Moreover, if further $\|\boldsymbol{v}\| \leq C_{\boldsymbol{v}}$ and $\|\boldsymbol{w}\| \geq c_{\boldsymbol{w}}$, there exists a constant $A_{\mathrm{crelu}} > 0$ depending on $C_{\boldsymbol{v}}$ and $c_{\boldsymbol{w}}$, such that*

$$\left\| \mathbb{E}_{\mathbf{Z}}\Big[\boldsymbol{g}_{\mathrm{crelu}}(\boldsymbol{v}, \boldsymbol{w}; \mathbf{Z})\Big] \right\|^2 \leq A_{\mathrm{crelu}} \left( \left\| \frac{\partial f}{\partial \boldsymbol{v}}(\boldsymbol{v}, \boldsymbol{w}) \right\|^2 + \left\langle \mathbb{E}_{\mathbf{Z}}\Big[\boldsymbol{g}_{\mathrm{crelu}}(\boldsymbol{v}, \boldsymbol{w}; \mathbf{Z})\Big], \frac{\partial f}{\partial \boldsymbol{w}}(\boldsymbol{v}, \boldsymbol{w}) \right\rangle \right).$$

**Lemma 8.** *When Algorithm 1 converges, $\mathbb{E}_{\mathbf{Z}}\Big[\frac{\partial \ell}{\partial \boldsymbol{v}}(\boldsymbol{v}, \boldsymbol{w}; \mathbf{Z})\Big]$ and $\mathbb{E}_{\mathbf{Z}}\Big[\boldsymbol{g}_{\mathrm{crelu}}(\boldsymbol{v}, \boldsymbol{w}; \mathbf{Z})\Big]$ vanish simultaneously, which only occurs at the*

1. *Saddle points where (8) is satisfied according to Proposition 1.*

2. *Minimizers of (2) where $\boldsymbol{v} = \boldsymbol{v}^*$, $\theta(\boldsymbol{w}, \boldsymbol{w}^*) = 0$, or $\boldsymbol{v} = (\boldsymbol{I}_m + \mathbf{1}_m \mathbf{1}_m^\top)^{-1}(\mathbf{1}_m \mathbf{1}_m^\top - \boldsymbol{I}_m)\boldsymbol{v}^*$, $\theta(\boldsymbol{w}, \boldsymbol{w}^*) = \pi$.*

### 3.3 Derivative of the Identity Function as STE

Now we consider the derivative of identity function. Similar results to Lemmas 5 and 6 are not valid anymore. It happens that the coarse gradient derived from the identity STE does not vanish at local minima, and Algorithm 1 may never converge there.

**Lemma 9.** *Let $\mu(x) = x$ in (5). Then the expected coarse partial gradient w.r.t. $\boldsymbol{w}$ is*

$$\mathbb{E}_{\mathbf{Z}}\Big[\boldsymbol{g}_{\mathrm{id}}(\boldsymbol{v}, \boldsymbol{w}; \mathbf{Z})\Big] = \frac{1}{\sqrt{2\pi}} \left( \|\boldsymbol{v}\|^2 \frac{\boldsymbol{w}}{\|\boldsymbol{w}\|} - (\boldsymbol{v}^\top \boldsymbol{v}^*)\boldsymbol{w}^* \right). \tag{14}$$

*If $\theta(\boldsymbol{w}, \boldsymbol{w}^*) = \pi$ and $\boldsymbol{v} = (\boldsymbol{I}_m + \mathbf{1}_m \mathbf{1}_m^\top)^{-1}(\mathbf{1}_m \mathbf{1}_m^\top - \boldsymbol{I}_m)\boldsymbol{v}^*$,*

$$\left\| \mathbb{E}_{\mathbf{Z}}\Big[\boldsymbol{g}_{\mathrm{id}}(\boldsymbol{v}, \boldsymbol{w}; \mathbf{Z})\Big] \right\| = \frac{2(m-1)}{\sqrt{2\pi}(m+1)^2}(\mathbf{1}_m^\top \boldsymbol{v}^*)^2 \geq 0,$$

*i.e., $\mathbb{E}_{\mathbf{Z}}\Big[\boldsymbol{g}_{\mathrm{id}}(\boldsymbol{v}, \boldsymbol{w}; \mathbf{Z})\Big]$ does not vanish at the local minimizers if $\mathbf{1}_m^\top \boldsymbol{v}^* \neq 0$ and $m > 1$.*

**Lemma 10.** *If $\boldsymbol{w} \neq \mathbf{0}_n$ and $\theta(\boldsymbol{w}, \boldsymbol{w}^*) \in (0, \pi)$, then the inner product between the expected coarse and true gradients w.r.t. $\boldsymbol{w}$ is*

$$\left\langle \mathbb{E}_{\mathbf{Z}}\Big[\boldsymbol{g}_{\mathrm{id}}(\boldsymbol{v}, \boldsymbol{w}; \mathbf{Z})\Big], \frac{\partial f}{\partial \boldsymbol{w}}(\boldsymbol{v}, \boldsymbol{w}) \right\rangle = \frac{\sin(\theta(\boldsymbol{w}, \boldsymbol{w}^*))}{(\sqrt{2\pi})^3 \|\boldsymbol{w}\|}(\boldsymbol{v}^\top \boldsymbol{v}^*)^2 \geq 0. \tag{15}$$

*When $\theta(\boldsymbol{w}, \boldsymbol{w}^*) \to \pi$, $\boldsymbol{v} \to (\boldsymbol{I}_m + \mathbf{1}_m \mathbf{1}_m^\top)^{-1}(\mathbf{1}_m \mathbf{1}_m^\top - \boldsymbol{I}_m)\boldsymbol{v}^*$, if $\mathbf{1}_m^\top \boldsymbol{v}^* \neq 0$ and $m > 1$, we have*

$$\frac{\left\| \mathbb{E}_{\mathbf{Z}}\Big[\boldsymbol{g}_{\mathrm{id}}(\boldsymbol{v}, \boldsymbol{w}; \mathbf{Z})\Big] \right\|^2}{\left\| \frac{\partial f}{\partial \boldsymbol{v}}(\boldsymbol{v}, \boldsymbol{w}) \right\|^2 + \left\langle \mathbb{E}_{\mathbf{Z}}\Big[\boldsymbol{g}_{\mathrm{id}}(\boldsymbol{v}, \boldsymbol{w}; \mathbf{Z})\Big], \frac{\partial f}{\partial \boldsymbol{w}}(\boldsymbol{v}, \boldsymbol{w}) \right\rangle} \to +\infty. \tag{16}$$

Lemma 9 suggests that if $\mathbf{1}_m^\top \boldsymbol{v}^* \neq 0$, the coarse gradient descent will never converge near the spurious minimizers with $\theta(\boldsymbol{w}, \boldsymbol{w}^*) = \pi$ and $\boldsymbol{v} = (\boldsymbol{I}_m + \mathbf{1}_m \mathbf{1}_m^\top)^{-1}(\mathbf{1}_m \mathbf{1}_m^\top - \boldsymbol{I}_m)\boldsymbol{v}^*$, because $\mathbb{E}_{\mathbf{Z}}\Big[\boldsymbol{g}_{\mathrm{id}}(\boldsymbol{v}, \boldsymbol{w}; \mathbf{Z})\Big]$ does not vanish there. By the positive correlation implied by (15) of Lemma 10, for some proper $(\boldsymbol{v}^0, \boldsymbol{w}^0)$, the iterates $\{(\boldsymbol{v}^t, \boldsymbol{w}^t)\}$ may move towards a local minimizer in the beginning. But when $\{(\boldsymbol{v}^t, \boldsymbol{w}^t)\}$ approaches it, the descent property (13) does not hold for $\mathbb{E}_{\mathbf{Z}}[\boldsymbol{g}_{\mathrm{id}}(\boldsymbol{v}, \boldsymbol{w}; \mathbf{Z})]$ because of (16), hence the training loss begins to increase and instability arises.

## 4 Experiments

While our theory implies that both vanilla and clipped ReLUs learn a two-linear-layer CNN, their empirical performances on deeper nets are different. In this section, we compare the performances of the identity, ReLU and clipped ReLU STEs on MNIST (LeCun et al., 1998) and CIFAR-10 (Krizhevsky, 2009) benchmarks for 2-bit or 4-bit quantized activations. As an illustration, we plot

the 2-bit quantized ReLU and its associated clipped ReLU in Figure 3 in the appendix. Intuitively, the clipped ReLU should be the best performer, as it best approximates the original quantized ReLU. We also report the instability issue of the training algorithm when using an improper STE in section 4.2. In all experiments, the weights are kept float.

The resolution $\alpha$ for the quantized ReLU needs to be carefully chosen to maintain the full-precision level accuracy. To this end, we follow (Cai et al., 2017) and resort to a modified batch normalization layer (Ioffe & Szegedy, 2015) without the scale and shift, whose output components approximately follow a unit Gaussian distribution. Then the $\alpha$ that fits the input of activation layer the best can be pre-computed by a variant of Lloyd's algorithm (Lloyd, 1982; Yin et al., 2018a) applied to a set of simulated 1-D half-Gaussian data. After determining the $\alpha$, it will be fixed during the whole training process. Since the original LeNet-5 does not have batch normalization, we add one prior to each activation layer. We emphasize that we are not claiming the superiority of the quantization approach used here, as it is nothing but the HWGQ (Cai et al., 2017), except we consider the uniform quantization.

The optimizer we use is the stochastic (coarse) gradient descent with momentum = 0.9 for all experiments. We train 50 epochs for LeNet-5 (LeCun et al., 1998) on MNIST, and 200 epochs for VGG-11 (Simonyan & Zisserman, 2014) and ResNet-20 (He et al., 2016) on CIFAR-10. The parameters/weights are initialized with those from their pre-trained full-precision counterparts. The schedule of the learning rate is specified in Table 2 in the appendix.

### 4.1 COMPARISON RESULTS

The experimental results are summarized in Table 1, where we record both the training losses and validation accuracies. Among the three STEs, the derivative of clipped ReLU gives the best overall performance, followed by vanilla ReLU and then by the identity function. For deeper networks, clipped ReLU is the best performer. But on the relatively shallow LeNet-5 network, vanilla ReLU exhibits comparable performance to the clipped ReLU, which is somewhat in line with our theoretical finding that ReLU is a great STE for learning the two-linear-layer (shallow) CNN.

|  | Network | BitWidth | Straight-through estimator | | |
|---|---|---|---|---|---|
|  |  |  | identity | vanilla ReLU | clipped ReLU |
| MNIST | LeNet5 | 2 | $2.6 \times 10^{-2}$/98.49 | $5.1 \times 10^{-3}$/99.24 | $5.4 \times 10^{-3}$/99.23 |
|  |  | 4 | $6.0 \times 10^{-3}$/98.98 | $9.0 \times 10^{-4}$/99.32 | $8.8 \times 10^{-4}$/99.24 |
| CIFAR10 | VGG11 | 2 | 0.19/86.58 | 0.10/88.69 | 0.02/90.92 |
|  |  | 4 | $3.1 \times 10^{-2}$/90.19 | $1.5 \times 10^{-3}$/92.01 | $1.3 \times 10^{-3}$/92.08 |
|  | ResNet20 | 2 | 1.56/46.52 | 1.50/48.05 | 0.24/88.39 |
|  |  | 4 | 1.38/54.16 | 0.25/86.59 | 0.04/91.24 |

Table 1: Training loss/validation accuracy (%) on MNIST and CIFAR-10 with quantized activations and float weights, for STEs using derivatives of the identity function, vanilla ReLU and clipped ReLU at bit-widths 2 and 4.

### 4.2 INSTABILITY

We report the phenomenon of being repelled from a good minimum on ResNet-20 with 4-bit activations when using the identity STE, to demonstrate the instability issue as predicted in Theorem 1. By Table 1, the coarse gradient descent algorithms using the vanilla and clipped ReLUs converge to the neighborhoods of the minima with validation accuracies (training losses) of 86.59% (0.25) and 91.24% (0.04), respectively, whereas that using the identity STE gives 54.16% (1.38). Note that the landscape of the empirical loss function does not depend on which STE is used in the training. Then we initialize training with the two improved minima and use the identity STE. To see if the algorithm is stable there, we start the training with a tiny learning rate of $10^{-5}$. For both initializations, the training loss and validation error significantly increase within the first 20 epochs; see Figure 4.2. To speedup training, at epoch 20, we switch to the normal schedule of learning rate specified in Table 2 and run 200 additional epochs. The training using the identity STE ends up with a much worse minimum. This is because the coarse gradient with identity STE does not vanish at the good minima in this case (Lemma 9). Similarly, the poor performance of ReLU STE on 2-bit activated ResNet-20

is also due to the instability of the corresponding training algorithm at good minima, as illustrated by Figure 4 in Appendix C, although it diverges much slower.

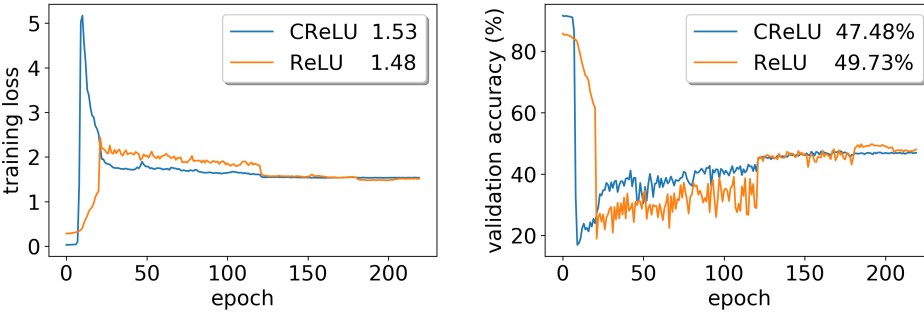

Figure 2: When initialized with weights (good minima) produced by the vanilla (orange) and clipped (blue) ReLUs on ResNet-20 with 4-bit activations, the coarse gradient descent using the identity STE ends up being repelled from there. The learning rate is set to $10^{-5}$ until epoch 20.

## 5 CONCLUDING REMARKS

We provided the first theoretical justification for the concept of STE that it gives rise to descent training algorithm. We considered three STEs: the derivatives of the identity function, vanilla ReLU and clipped ReLU, for learning a two-linear-layer CNN with binary activation. We derived the explicit formulas of the expected coarse gradients corresponding to the STEs, and showed that the negative expected coarse gradients based on vanilla and clipped ReLUs are descent directions for minimizing the population loss, whereas the identity STE is not since it generates a coarse gradient incompatible with the energy landscape. The instability/incompatibility issue was confirmed in CIFAR experiments for improper choices of STE. In the future work, we aim further understanding of coarse gradient descent for large-scale optimization problems with intractable gradients.

### ACKNOWLEDGMENTS

This work was partially supported by NSF grants DMS-1522383, IIS-1632935, ONR grant N00014-18-1-2527, AFOSR grant FA9550-18-0167, DOE grant DE-SC0013839 and STROBE STC NSF grant DMR-1548924.

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

## APPENDIX

### A. THE PLOTS OF QUANTIZED AND CLIPPED RELUS

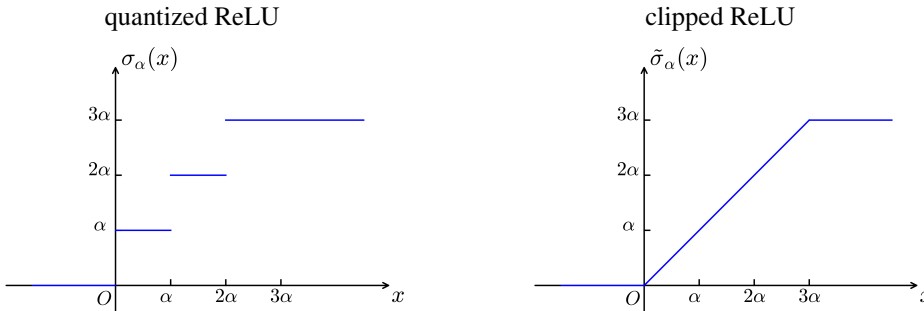

Figure 3: The plots of 2-bit quantized ReLU $\sigma_\alpha(x)$ (with $2^2 = 4$ quantization levels including 0) and the associated clipped ReLU $\tilde{\sigma}_\alpha(x)$. $\alpha$ is the resolution determined in advance of the network training.

### B. THE SCHEDULE OF LEARNING RATE

| Network | # epochs | Batch size | Learning rate | | |
|---|---|---|---|---|---|
| | | | initial | decay rate | milestone |
| LeNet5 | 50 | 64 | 0.1 | 0.1 | [20,40] |
| VGG11 | 200 | 128 | 0.01 | 0.1 | [80,140] |
| ResNet20 | 200 | 128 | 0.01 | 0.1 | [80,140] |

Table 2: The schedule of the learning rate.

### C. INSTABILITY OF RELU STE ON RESNET-20 WITH 2-BIT ACTIVATIONS

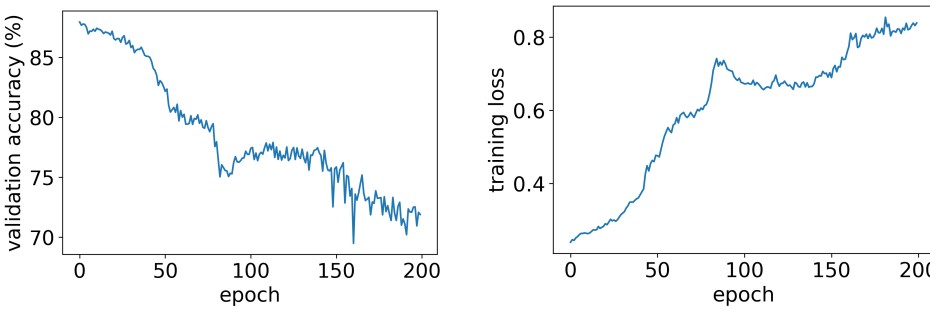

Figure 4: When initialized with the weights produced by the clipped ReLU STE on ResNet-20 with 2-bit activations (88.38% validation accuracy), the coarse gradient descent using the ReLU STE with $10^{-5}$ learning rate is not stable there, and both classification and training errors begin to increase.

### D. ADDITIONAL SUPPORTING LEMMAS

**Lemma 11.** *Let $\mathbf{z} \in \mathbb{R}^n$ be a Gaussian random vector with entries i.i.d. sampled from $\mathcal{N}(0,1)$. Given nonzero vectors $\boldsymbol{w}$, $\tilde{\boldsymbol{w}} \in \mathbb{R}^n$ with the angle $\theta$, we have*

$$\mathbb{E}\left[\mathbf{1}_{\{\mathbf{z}^\top \boldsymbol{w} > 0\}}\right] = \frac{1}{2}, \ \mathbb{E}\left[\mathbf{1}_{\{\mathbf{z}^\top \boldsymbol{w} > 0, \, \mathbf{z}^\top \tilde{\boldsymbol{w}} > 0\}}\right] = \frac{\pi - \theta}{2\pi},$$

*and*

$$\mathbb{E}\left[\mathbf{z}1_{\{\mathbf{z}^\top\boldsymbol{w}>0\}}\right] = \frac{1}{\sqrt{2\pi}}\frac{\boldsymbol{w}}{\|\boldsymbol{w}\|}, \ \mathbb{E}\left[\mathbf{z}1_{\{\mathbf{z}^\top\boldsymbol{w}>0,\,\mathbf{z}^\top\tilde{\boldsymbol{w}}>0\}}\right] = \frac{\cos(\theta/2)}{\sqrt{2\pi}}\frac{\frac{\boldsymbol{w}}{\|\boldsymbol{w}\|}+\frac{\tilde{\boldsymbol{w}}}{\|\tilde{\boldsymbol{w}}\|}}{\left\|\frac{\boldsymbol{w}}{\|\boldsymbol{w}\|}+\frac{\tilde{\boldsymbol{w}}}{\|\tilde{\boldsymbol{w}}\|}\right\|},^{2}$$

**Proof of Lemma 11.** The third identity was proved in Lemma A.1 of (Du et al., 2018). To show the first one, without loss of generality we assume $\boldsymbol{w} = [w_1, \mathbf{0}_{n-1}^\top]^\top$ with $w_1 > 0$, then $\mathbb{E}\left[1_{\{\mathbf{z}^\top\boldsymbol{w}>0\}}\right] = \mathbb{P}(z_1 > 0) = \frac{1}{2}$.

We further assume $\tilde{\boldsymbol{w}} = [\tilde{w}_1, \tilde{w}_2, \mathbf{0}_{n-2}^\top]^\top$. It is easy to see that

$$\mathbb{E}\left[1_{\{\mathbf{z}^\top\boldsymbol{w}>0,\,\mathbf{z}^\top\tilde{\boldsymbol{w}}>0\}}\right] = \mathbb{P}(\mathbf{z}^\top\boldsymbol{w} > 0, \mathbf{z}^\top\tilde{\boldsymbol{w}} > 0) = \frac{\pi-\theta}{2\pi}.$$

To prove the last identity, we use polar representation of two-dimensional Gaussian random variables, where $r$ is the radius and $\phi$ is the angle with $\mathrm{d}\mathbb{P}_r = r\exp(-r^2/2)\mathrm{d}r$ and $\mathrm{d}\mathbb{P}_\phi = \frac{1}{2\pi}\mathrm{d}\phi$. Then $\mathbb{E}\left[z_i 1_{\{\mathbf{z}^\top\boldsymbol{w}>0,\,\mathbf{z}^\top\tilde{\boldsymbol{w}}>0\}}\right] = 0$ for $i \geq 3$. Moreover,

$$\mathbb{E}\left[z_1 1_{\{\mathbf{z}^\top\boldsymbol{w}>0,\,\mathbf{z}^\top\tilde{\boldsymbol{w}}>0\}}\right] = \frac{1}{2\pi}\int_0^\infty r^2\exp\left(-\frac{r^2}{2}\right)\mathrm{d}r\int_{-\frac{\pi}{2}+\theta}^{\frac{\pi}{2}}\cos(\phi)\mathrm{d}\phi = \frac{1+\cos(\theta)}{2\sqrt{2\pi}}$$

and

$$\mathbb{E}\left[z_2 1_{\{\mathbf{z}^\top\boldsymbol{w}>0,\,\mathbf{z}^\top\tilde{\boldsymbol{w}}>0\}}\right] = \frac{1}{2\pi}\int_0^\infty r^2\exp\left(-\frac{r^2}{2}\right)\mathrm{d}r\int_{-\frac{\pi}{2}+\theta}^{\frac{\pi}{2}}\sin(\phi)\mathrm{d}\phi = \frac{\sin(\theta)}{2\sqrt{2\pi}}.$$

Therefore,

$$\mathbb{E}\left[\mathbf{z}1_{\{\mathbf{z}^\top\boldsymbol{w}>0,\,\mathbf{z}^\top\tilde{\boldsymbol{w}}>0\}}\right] = \frac{\cos(\theta/2)}{\sqrt{2\pi}}[\cos(\theta/2),\sin(\theta/2),\mathbf{0}_{n-2}^\top]^\top = \frac{\cos(\theta/2)}{\sqrt{2\pi}}\frac{\frac{\boldsymbol{w}}{\|\boldsymbol{w}\|}+\frac{\tilde{\boldsymbol{w}}}{\|\tilde{\boldsymbol{w}}\|}}{\left\|\frac{\boldsymbol{w}}{\|\boldsymbol{w}\|}+\frac{\tilde{\boldsymbol{w}}}{\|\tilde{\boldsymbol{w}}\|}\right\|},$$

where the last equality holds because $\frac{\boldsymbol{w}}{\|\boldsymbol{w}\|}$ and $\frac{\frac{\boldsymbol{w}}{\|\boldsymbol{w}\|}+\frac{\tilde{\boldsymbol{w}}}{\|\tilde{\boldsymbol{w}}\|}}{\left\|\frac{\boldsymbol{w}}{\|\boldsymbol{w}\|}+\frac{\tilde{\boldsymbol{w}}}{\|\tilde{\boldsymbol{w}}\|}\right\|}$ are two unit-normed vectors with angle $\theta/2$. $\qquad\square$

**Lemma 12.** *Let* $\mathbf{z} \in \mathbb{R}^n$ *be a Gaussian random vector with entries i.i.d. sampled from* $\mathcal{N}(0,1)$. *Given nonzero vectors* $\boldsymbol{w}, \tilde{\boldsymbol{w}} \in \mathbb{R}^n$ *with the angle* $\theta$, *we have*

$$\mathbb{E}\left[\mathbf{z}1_{\{0<\mathbf{z}^\top\boldsymbol{w}<1\}}\right] = p(0,\boldsymbol{w})\frac{\boldsymbol{w}}{\|\boldsymbol{w}\|},$$

$$\mathbb{E}\left[\mathbf{z}1_{\{0<\mathbf{z}^\top\boldsymbol{w}<1,\,\mathbf{z}^\top\tilde{\boldsymbol{w}}>0\}}\right] = ((p(\theta,\boldsymbol{w})) - \cot(\theta/2)\cdot q(\theta,\boldsymbol{w}))\frac{\boldsymbol{w}}{\|\boldsymbol{w}\|}$$
$$+ \csc(\theta/2)\cdot q(\theta,\boldsymbol{w})\frac{\frac{\boldsymbol{w}}{\|\boldsymbol{w}\|}+\frac{\tilde{\boldsymbol{w}}}{\|\tilde{\boldsymbol{w}}\|}}{\left\|\frac{\boldsymbol{w}}{\|\boldsymbol{w}\|}+\frac{\tilde{\boldsymbol{w}}}{\|\tilde{\boldsymbol{w}}\|}\right\|}.$$

*Here*

$$p(\theta,\boldsymbol{w}) = \frac{1}{2\pi}\int_{-\frac{\pi}{2}+\theta}^{\frac{\pi}{2}}\cos(\phi)\xi\left(\frac{\sec(\phi)}{\|\boldsymbol{w}\|}\right)\mathrm{d}\phi, \ q(\theta,\boldsymbol{w}) = \frac{1}{2\pi}\int_{-\frac{\pi}{2}+\theta}^{\frac{\pi}{2}}\sin(\phi)\xi\left(\frac{\sec(\phi)}{\|\boldsymbol{w}\|}\right)\mathrm{d}\phi$$

*with* $\xi(x) = \int_0^x r^2\exp(-\frac{r^2}{2})\mathrm{d}r$ *satisfying* $\xi(+\infty) = \sqrt{\frac{\pi}{2}}$. *In addition,*

$$p(\pi,\boldsymbol{w}) = q(0,\boldsymbol{w}) = q(\pi,\boldsymbol{w}) = 0.$$

---

[2]Same as in Lemma 4, we redefine $\mathbb{E}\left[\mathbf{z}1_{\{\mathbf{z}^\top\boldsymbol{w}>0,\,\mathbf{z}^\top\tilde{\boldsymbol{w}}>0\}}\right] = \mathbf{0}_n$ in the case $\theta(\boldsymbol{w},\tilde{\boldsymbol{w}}) = \pi$.

**Proof of Lemma 12.** Following the proof of Lemma 11, we let $\boldsymbol{w} = [w_1, \boldsymbol{0}_{n-1}^\top]^\top$ with $w_1 > 0$, $\tilde{\boldsymbol{w}} = [\tilde{w}_1, \tilde{w}_2, \boldsymbol{0}_{n-2}^\top]^\top$, and use the polar representation of two-dimensional Gaussian distribution. Then

$$\mathbb{E}\left[z_1 1_{\{0 < \boldsymbol{z}^\top \boldsymbol{w} < 1\}}\right] = \frac{1}{2\pi} \int_{-\frac{\pi}{2}}^{\frac{\pi}{2}} \cos(\phi) \int_0^{\frac{\sec(\phi)}{\|\boldsymbol{w}\|}} r^2 \exp\left(-\frac{r^2}{2}\right) \mathrm{d}r\mathrm{d}\phi = p(0, \boldsymbol{w})$$

and

$$\mathbb{E}\left[z_2 1_{\{0 < \boldsymbol{z}^\top \boldsymbol{w} < 1\}}\right] = \frac{1}{2\pi} \int_{-\frac{\pi}{2}}^{\frac{\pi}{2}} \sin(\phi) \int_0^{\frac{\sec(\phi)}{\|\boldsymbol{w}\|}} r^2 \exp\left(-\frac{r^2}{2}\right) \mathrm{d}r\mathrm{d}\phi = q(0, \boldsymbol{w}) = 0,$$

since the integrand above is an odd function in $\phi$. Moreover, $\mathbb{E}\left[z_i 1_{\{0 < \boldsymbol{z}^\top \boldsymbol{w} < 1\}}\right] = 0$ for $i \geq 3$. So the first identity holds. For the second one, we have

$$\mathbb{E}\left[z_1 1_{\{0 < \boldsymbol{z}^\top \boldsymbol{w} < 1, \boldsymbol{z}^\top \tilde{\boldsymbol{w}} > 0\}}\right] = \frac{1}{2\pi} \int_{-\frac{\pi}{2}+\theta}^{\frac{\pi}{2}} \cos(\phi) \int_0^{\frac{\sec(\phi)}{\|\boldsymbol{w}\|}} r^2 \exp\left(-\frac{r^2}{2}\right) \mathrm{d}r\mathrm{d}\phi = p(\theta, \boldsymbol{w}),$$

and similarly, $\mathbb{E}\left[z_2 1_{\{0 < \boldsymbol{z}^\top \boldsymbol{w} < 1, \boldsymbol{z}^\top \tilde{\boldsymbol{w}} > 0\}}\right] = q(\theta, \boldsymbol{w})$. Therefore,

$$\mathbb{E}[\boldsymbol{z} 1_{\{0 < \boldsymbol{z}^\top \boldsymbol{w} < 1, \boldsymbol{z}^\top \tilde{\boldsymbol{w}} > 0\}}] = [p(\theta, \boldsymbol{w}), q(\theta, \boldsymbol{w}), \boldsymbol{0}_{n-2}^\top]^\top.$$

Since $\frac{\boldsymbol{w}}{\|\boldsymbol{w}\|} = [1, \boldsymbol{0}_{n-1}^\top]^\top$ and $\frac{\frac{\boldsymbol{w}}{\|\boldsymbol{w}\|} + \frac{\tilde{\boldsymbol{w}}}{\|\tilde{\boldsymbol{w}}\|}}{\|\frac{\boldsymbol{w}}{\|\boldsymbol{w}\|} + \frac{\tilde{\boldsymbol{w}}}{\|\tilde{\boldsymbol{w}}\|}\|} = [\cos(\theta/2), \sin(\theta/2), \boldsymbol{0}_{n-2}^\top]^\top$, it is easy to check the second identity is also true. $\square$

**Lemma 13.** *Let $p(\theta, w)$ and $q(\theta, \boldsymbol{w})$ be defined in Lemma 12. Then for $\theta \in [\frac{\pi}{2}, \pi]$, we have*

1. $p(\theta, \boldsymbol{w}) \leq q(\theta, \boldsymbol{w})$.

2. $\left(1 - \frac{\theta}{\pi}\right) p(0, \boldsymbol{w}) \leq q(\theta, \boldsymbol{w})$.

**Proof of Lemma 13.** 1. Let $\theta \in [\frac{\pi}{2}, \pi]$, since $\xi \geq 0$,

$$p(\theta, \boldsymbol{w}) = \frac{1}{2\pi} \int_{\theta-\frac{\pi}{2}}^{\frac{\pi}{2}} \cos(\phi)\xi\left(\frac{\sec(\phi)}{\|\boldsymbol{w}\|}\right) \mathrm{d}\phi = \frac{1}{2\pi} \int_{\theta-\frac{\pi}{2}}^{\frac{\pi}{2}} \sin\left(\frac{\pi}{2} - \phi\right) \xi\left(\frac{\sec(\phi)}{\|\boldsymbol{w}\|}\right) \mathrm{d}\phi$$

$$\leq \frac{1}{2\pi} \int_{\theta-\frac{\pi}{2}}^{\frac{\pi}{2}} \sin(\theta - \phi)\xi\left(\frac{\sec(\phi)}{\|\boldsymbol{w}\|}\right) \mathrm{d}\phi \leq \frac{1}{2\pi} \int_{\theta-\frac{\pi}{2}}^{\frac{\pi}{2}} \sin(\phi)\xi\left(\frac{\sec(\phi)}{\|\boldsymbol{w}\|}\right) \mathrm{d}\phi = q(\theta, \boldsymbol{w}),$$

where the last inequality is due to the rearrangement inequality since both $\sin(\phi)$ and $\xi\left(\frac{\sec(\phi)}{\|\boldsymbol{w}\|}\right)$ are increasing in $\phi$ on $[0, \frac{\pi}{2}]$.

2. Since $\cos(\phi)\xi\left(\frac{\sec(\phi)}{\|\boldsymbol{w}\|}\right)$ is even, we have

$$\frac{1}{\pi} \int_{-\frac{\pi}{2}}^{\frac{\pi}{2}} \cos(\phi)\xi\left(\frac{\sec(\phi)}{\|\boldsymbol{w}\|}\right) \mathrm{d}\phi = \frac{2}{\pi} \int_0^{\frac{\pi}{2}} \cos(\phi)\xi\left(\frac{\sec(\phi)}{\|\boldsymbol{w}\|}\right) \mathrm{d}\phi$$

$$\leq \frac{2}{\pi} \int_0^{\frac{\pi}{2}} \sin(\phi)\xi\left(\frac{\sec(\phi)}{\|\boldsymbol{w}\|}\right) \mathrm{d}\phi \leq \frac{1}{\pi - \theta} \int_{\theta-\frac{\pi}{2}}^{\frac{\pi}{2}} \sin(\phi)\xi\left(\frac{\sec(\phi)}{\|\boldsymbol{w}\|}\right) \mathrm{d}\phi.$$

The first inequality is due to part 1 which gives $p(\pi/2, \boldsymbol{w}) \leq q(\pi/2, \boldsymbol{w})$, whereas the second one holds because $\sin(\phi)\xi\left(\frac{\sec(\phi)}{\|\boldsymbol{w}\|}\right)$ is increasing on $[0, \frac{\pi}{2}]$. $\square$

**Lemma 14.** *For any nonzero vectors $\boldsymbol{w}$ and $\tilde{\boldsymbol{w}}$ with $\|\tilde{\boldsymbol{w}}\| \geq \|\boldsymbol{w}\| = c > 0$, we have*

1. $|\theta(\boldsymbol{w}, \boldsymbol{w}^*) - \theta(\tilde{\boldsymbol{w}}, \boldsymbol{w}^*)| \le \frac{\pi}{2c}\|\boldsymbol{w} - \tilde{\boldsymbol{w}}\|.$

2. $\left\| \frac{1}{\|\boldsymbol{w}\|} \frac{\left(\boldsymbol{I}_n - \frac{\boldsymbol{w}\boldsymbol{w}^\top}{\|\boldsymbol{w}\|^2}\right)\boldsymbol{w}^*}{\left\|\left(\boldsymbol{I}_n - \frac{\boldsymbol{w}\boldsymbol{w}^\top}{\|\boldsymbol{w}\|^2}\right)\boldsymbol{w}^*\right\|} - \frac{1}{\|\tilde{\boldsymbol{w}}\|} \frac{\left(\boldsymbol{I}_n - \frac{\tilde{\boldsymbol{w}}\tilde{\boldsymbol{w}}^\top}{\|\tilde{\boldsymbol{w}}\|^2}\right)\boldsymbol{w}^*}{\left\|\left(\boldsymbol{I}_n - \frac{\tilde{\boldsymbol{w}}\tilde{\boldsymbol{w}}^\top}{\|\tilde{\boldsymbol{w}}\|^2}\right)\boldsymbol{w}^*\right\|} \right\| \le \frac{1}{c^2}\|\boldsymbol{w} - \tilde{\boldsymbol{w}}\|.$

**Proof of Lemma 14.** 1. Since by Cauchy-Schwarz inequality,

$$\left\langle \tilde{\boldsymbol{w}}, \boldsymbol{w} - \frac{c\tilde{\boldsymbol{w}}}{\|\tilde{\boldsymbol{w}}\|} \right\rangle = \tilde{\boldsymbol{w}}^\top \boldsymbol{w} - c\|\tilde{\boldsymbol{w}}\| \le 0,$$

we have

$$\|\tilde{\boldsymbol{w}} - \boldsymbol{w}\|^2 = \left\|\left(1 - \frac{c}{\|\tilde{\boldsymbol{w}}\|}\right)\tilde{\boldsymbol{w}} - \left(\boldsymbol{w} - \frac{c\tilde{\boldsymbol{w}}}{\|\tilde{\boldsymbol{w}}\|}\right)\right\|^2 \ge \left\|\left(1 - \frac{c}{\|\tilde{\boldsymbol{w}}\|}\right)\tilde{\boldsymbol{w}}\right\|^2 + \left\|\boldsymbol{w} - \frac{c\tilde{\boldsymbol{w}}}{\|\tilde{\boldsymbol{w}}\|}\right\|^2$$

$$\ge \left\|\boldsymbol{w} - \frac{c\tilde{\boldsymbol{w}}}{\|\tilde{\boldsymbol{w}}\|}\right\|^2 = c^2 \left\|\frac{\boldsymbol{w}}{\|\boldsymbol{w}\|} - \frac{\tilde{\boldsymbol{w}}}{\|\tilde{\boldsymbol{w}}\|}\right\|^2. \tag{17}$$

Therefore,

$$|\theta(\boldsymbol{w}, \boldsymbol{w}^*) - \theta(\tilde{\boldsymbol{w}}, \boldsymbol{w}^*)| \le \theta(\boldsymbol{w}, \tilde{\boldsymbol{w}}) = \theta\left(\frac{\boldsymbol{w}}{\|\boldsymbol{w}\|}, \frac{\tilde{\boldsymbol{w}}}{\|\tilde{\boldsymbol{w}}\|}\right)$$

$$\le \pi \sin\left(\frac{\theta\left(\frac{\boldsymbol{w}}{\|\boldsymbol{w}\|}, \frac{\tilde{\boldsymbol{w}}}{\|\tilde{\boldsymbol{w}}\|}\right)}{2}\right) = \frac{\pi}{2}\left\|\frac{\boldsymbol{w}}{\|\boldsymbol{w}\|} - \frac{\tilde{\boldsymbol{w}}}{\|\tilde{\boldsymbol{w}}\|}\right\| \le \frac{\pi}{2c}\|\boldsymbol{w} - \tilde{\boldsymbol{w}}\|,$$

where we used the fact $\sin(x) \ge \frac{2x}{\pi}$ for $x \in [0, \frac{\pi}{2}]$ and the estimate in (17).

2. Since $\left(\boldsymbol{I}_n - \frac{\boldsymbol{w}\boldsymbol{w}^\top}{\|\boldsymbol{w}\|^2}\right)\boldsymbol{w}^*$ is the projection of $\boldsymbol{w}^*$ onto the complement space of $\boldsymbol{w}$, and likewise for $\left(\boldsymbol{I}_n - \frac{\tilde{\boldsymbol{w}}\tilde{\boldsymbol{w}}^\top}{\|\tilde{\boldsymbol{w}}\|^2}\right)\boldsymbol{w}^*$, the angle between $\left(\boldsymbol{I}_n - \frac{\boldsymbol{w}\boldsymbol{w}^\top}{\|\boldsymbol{w}\|^2}\right)\boldsymbol{w}^*$ and $\left(\boldsymbol{I}_n - \frac{\tilde{\boldsymbol{w}}\tilde{\boldsymbol{w}}^\top}{\|\tilde{\boldsymbol{w}}\|^2}\right)\boldsymbol{w}^*$ is equal to the angle between $\boldsymbol{w}$ and $\tilde{\boldsymbol{w}}$. Therefore,

$$\left\langle \frac{\left(\boldsymbol{I}_n - \frac{\boldsymbol{w}\boldsymbol{w}^\top}{\|\boldsymbol{w}\|^2}\right)\boldsymbol{w}^*}{\left\|\left(\boldsymbol{I}_n - \frac{\boldsymbol{w}\boldsymbol{w}^\top}{\|\boldsymbol{w}\|^2}\right)\boldsymbol{w}^*\right\|}, \frac{\left(\boldsymbol{I}_n - \frac{\tilde{\boldsymbol{w}}\tilde{\boldsymbol{w}}^\top}{\|\tilde{\boldsymbol{w}}\|^2}\right)\boldsymbol{w}^*}{\left\|\left(\boldsymbol{I}_n - \frac{\tilde{\boldsymbol{w}}\tilde{\boldsymbol{w}}^\top}{\|\tilde{\boldsymbol{w}}\|^2}\right)\boldsymbol{w}^*\right\|} \right\rangle = \left\langle \frac{\boldsymbol{w}}{\|\boldsymbol{w}\|}, \frac{\tilde{\boldsymbol{w}}}{\|\tilde{\boldsymbol{w}}\|} \right\rangle,$$

and thus

$$\left\| \frac{1}{\|\boldsymbol{w}\|} \frac{\left(\boldsymbol{I}_n - \frac{\boldsymbol{w}\boldsymbol{w}^\top}{\|\boldsymbol{w}\|^2}\right)\boldsymbol{w}^*}{\left\|\left(\boldsymbol{I}_n - \frac{\boldsymbol{w}\boldsymbol{w}^\top}{\|\boldsymbol{w}\|^2}\right)\boldsymbol{w}^*\right\|} - \frac{1}{\|\tilde{\boldsymbol{w}}\|} \frac{\left(\boldsymbol{I}_n - \frac{\tilde{\boldsymbol{w}}\tilde{\boldsymbol{w}}^\top}{\|\tilde{\boldsymbol{w}}\|^2}\right)\boldsymbol{w}^*}{\left\|\left(\boldsymbol{I}_n - \frac{\tilde{\boldsymbol{w}}\tilde{\boldsymbol{w}}^\top}{\|\tilde{\boldsymbol{w}}\|^2}\right)\boldsymbol{w}^*\right\|} \right\|$$

$$= \left\|\frac{\boldsymbol{w}}{\|\boldsymbol{w}\|^2} - \frac{\tilde{\boldsymbol{w}}}{\|\tilde{\boldsymbol{w}}\|^2}\right\| = \frac{\|\boldsymbol{w} - \tilde{\boldsymbol{w}}\|}{\|\boldsymbol{w}\|\|\tilde{\boldsymbol{w}}\|} \le \frac{1}{c^2}\|\boldsymbol{w} - \tilde{\boldsymbol{w}}\|.$$

The second equality above holds because

$$\left\|\frac{\boldsymbol{w}}{\|\boldsymbol{w}\|^2} - \frac{\tilde{\boldsymbol{w}}}{\|\tilde{\boldsymbol{w}}\|^2}\right\|^2 = \frac{1}{\|\boldsymbol{w}\|^2} + \frac{1}{\|\tilde{\boldsymbol{w}}\|^2} - \frac{2\langle\boldsymbol{w}, \tilde{\boldsymbol{w}}\rangle}{\|\boldsymbol{w}\|^2\|\tilde{\boldsymbol{w}}\|^2} = \frac{\|\boldsymbol{w} - \tilde{\boldsymbol{w}}\|^2}{\|\boldsymbol{w}\|^2\|\tilde{\boldsymbol{w}}\|^2}.$$

$\square$

## E. MAIN PROOFS

**Lemma 1.** *If $\boldsymbol{w} \ne \boldsymbol{0}_n$, the population loss $f(\boldsymbol{v}, \boldsymbol{w})$ is given by*

$$\frac{1}{8}\left[\boldsymbol{v}^\top(\boldsymbol{I}_m + \boldsymbol{1}_m\boldsymbol{1}_m^\top)\boldsymbol{v} - 2\boldsymbol{v}^\top\left(\left(1 - \frac{2}{\pi}\theta(\boldsymbol{w}, \boldsymbol{w}^*)\right)\boldsymbol{I}_m + \boldsymbol{1}_m\boldsymbol{1}_m^\top\right)\boldsymbol{v}^* + (\boldsymbol{v}^*)^\top(\boldsymbol{I}_m + \boldsymbol{1}_m\boldsymbol{1}_m^\top)\boldsymbol{v}^*\right].$$

*In addition, $f(\boldsymbol{v}, \boldsymbol{w}) = \frac{1}{8}(\boldsymbol{v}^*)^\top(\boldsymbol{I}_m + \boldsymbol{1}_m\boldsymbol{1}_m^\top)\boldsymbol{v}^*$ for $\boldsymbol{w} = \boldsymbol{0}_n$.*

**Proof of Lemma 1.** We notice that

$$f(\boldsymbol{v}, \boldsymbol{w}) = \frac{1}{2} \big( \boldsymbol{v}^\top \mathbb{E}_{\mathbf{Z}} [\sigma(\mathbf{Z}\boldsymbol{w})\sigma(\mathbf{Z}\boldsymbol{w})^\top] \boldsymbol{v} - 2\boldsymbol{v}^\top \mathbb{E}_{\mathbf{Z}} [\sigma(\mathbf{Z}\boldsymbol{w})\sigma(\mathbf{Z}\boldsymbol{w}^*)^\top] \boldsymbol{v}^*$$
$$+ (\boldsymbol{v}^*)^\top \mathbb{E}_{\mathbf{Z}} [\sigma(\mathbf{Z}\boldsymbol{w}^*)\sigma(\mathbf{Z}\boldsymbol{w}^*)^\top] \boldsymbol{v}^* \big).$$

Let $\mathbf{Z}_i^\top$ be the $i^{\text{th}}$ row vector of $\mathbf{Z}$. Since $\boldsymbol{w} \neq \mathbf{0}_n$, using Lemma 11, we have

$$\mathbb{E}_{\mathbf{Z}} \big[ \sigma(\mathbf{Z}\boldsymbol{w})\sigma(\mathbf{Z}\boldsymbol{w})^\top \big]_{ii} = \mathbb{E} \big[ \sigma(\mathbf{Z}_i^\top \boldsymbol{w})\sigma(\mathbf{Z}_i^\top \boldsymbol{w}) \big] = \mathbb{E} \Big[ 1_{\{\mathbf{Z}_i^\top \boldsymbol{w} > 0\}} \Big] = \frac{1}{2},$$

and for $i \neq j$,

$$\mathbb{E}_{\mathbf{Z}} \big[ \sigma(\mathbf{Z}\boldsymbol{w})\sigma(\mathbf{Z}\boldsymbol{w})^\top \big]_{ij} = \mathbb{E} \big[ \sigma(\mathbf{Z}_i^\top \boldsymbol{w})\sigma(\mathbf{Z}_j^\top \boldsymbol{w}) \big] = \mathbb{E} \Big[ 1_{\{\mathbf{Z}_i^\top \boldsymbol{w} > 0\}} \Big] \mathbb{E} \Big[ 1_{\{\mathbf{Z}_j^\top \boldsymbol{w} > 0\}} \Big] = \frac{1}{4}.$$

Therefore, $\mathbb{E}_{\mathbf{Z}} \big[ \sigma(\mathbf{Z}\boldsymbol{w})\sigma(\mathbf{Z}\boldsymbol{w})^\top \big] = \mathbb{E}_{\mathbf{Z}} \big[ \sigma(\mathbf{Z}\boldsymbol{w}^*)\sigma(\mathbf{Z}\boldsymbol{w}^*)^\top \big] = \frac{1}{4} \big( \boldsymbol{I}_m + \mathbf{1}_m \mathbf{1}_m^\top \big)$. Furthermore,

$$\mathbb{E}_{\mathbf{Z}} \big[ \sigma(\mathbf{Z}\boldsymbol{w})\sigma(\mathbf{Z}\boldsymbol{w}^*)^\top \big]_{ii} = \mathbb{E} \Big[ 1_{\{\mathbf{Z}_i^\top \boldsymbol{w} > 0, \mathbf{Z}_i^\top \boldsymbol{w}^* > 0\}} \Big] = \frac{\pi - \theta(\boldsymbol{w}, \boldsymbol{w}^*)}{2\pi},$$

and $\mathbb{E}_{\mathbf{Z}} \big[ \sigma(\mathbf{Z}\boldsymbol{w})\sigma(\mathbf{Z}\boldsymbol{w}^*)^\top \big]_{ij} = \frac{1}{4}$. So,

$$\mathbb{E}_{\mathbf{Z}} \big[ \sigma(\mathbf{Z}\boldsymbol{w})\sigma(\mathbf{Z}\boldsymbol{w}^*)^\top \big] = \frac{1}{4} \left( \left( 1 - \frac{2\theta(\boldsymbol{w}, \boldsymbol{w}^*)}{\pi} \right) \boldsymbol{I}_m + \mathbf{1}_m \mathbf{1}_m^\top \right).$$

Then it is easy to validate the first claim. Moreover, if $\boldsymbol{w} = \mathbf{0}_n$, then

$$f(\boldsymbol{v}, \boldsymbol{w}) = \frac{1}{2} (\boldsymbol{v}^*)^\top \mathbb{E}_{\mathbf{Z}} [\sigma(\mathbf{Z}\boldsymbol{w}^*)\sigma(\mathbf{Z}\boldsymbol{w}^*)^\top] \boldsymbol{v}^* = \frac{1}{8} (\boldsymbol{v}^*)^\top \big( \boldsymbol{I}_m + \mathbf{1}_m \mathbf{1}_m^\top \big) \boldsymbol{v}^*.$$

$\square$

**Lemma 2.** *If $\boldsymbol{w} \neq \mathbf{0}_n$ and $\theta(\boldsymbol{w}, \boldsymbol{w}^*) \in (0, \pi)$, the partial gradients of $f(\boldsymbol{v}, \boldsymbol{w})$ w.r.t. $\boldsymbol{v}$ and $\boldsymbol{w}$ are*

$$\frac{\partial f}{\partial \boldsymbol{v}}(\boldsymbol{v}, \boldsymbol{w}) = \frac{1}{4} \big( \boldsymbol{I}_m + \mathbf{1}_m \mathbf{1}_m^\top \big) \boldsymbol{v} - \frac{1}{4} \left( \left( 1 - \frac{2}{\pi} \theta(\boldsymbol{w}, \boldsymbol{w}^*) \right) \boldsymbol{I}_m + \mathbf{1}_m \mathbf{1}_m^\top \right) \boldsymbol{v}^*$$

*and*

$$\frac{\partial f}{\partial \boldsymbol{w}}(\boldsymbol{v}, \boldsymbol{w}) = -\frac{\boldsymbol{v}^\top \boldsymbol{v}^*}{2\pi \|\boldsymbol{w}\|} \frac{\left( \boldsymbol{I}_n - \frac{\boldsymbol{w}\boldsymbol{w}^\top}{\|\boldsymbol{w}\|^2} \right) \boldsymbol{w}^*}{\left\| \left( \boldsymbol{I}_n - \frac{\boldsymbol{w}\boldsymbol{w}^\top}{\|\boldsymbol{w}\|^2} \right) \boldsymbol{w}^* \right\|},$$

*respectively.*

**Proof of Lemma 2.** The first claim is trivial, and we only show the second one. Since $\theta(\boldsymbol{w}, \boldsymbol{w}^*) = \arccos \left( \frac{\boldsymbol{w}^\top \boldsymbol{w}^*}{\|\boldsymbol{w}\|} \right)$ is differentiable w.r.t. $\boldsymbol{w}$ at $\theta(\boldsymbol{w}, \boldsymbol{w}^*) \in (0, \pi)$, we have

$$\frac{\partial f}{\partial \boldsymbol{w}}(\boldsymbol{v}, \boldsymbol{w}) = \frac{\boldsymbol{v}^\top \boldsymbol{v}^*}{2\pi} \frac{\partial \theta}{\partial \boldsymbol{w}}(\boldsymbol{w}, \boldsymbol{w}^*) = -\frac{\boldsymbol{v}^\top \boldsymbol{v}^*}{2\pi} \frac{\|\boldsymbol{w}\|^2 \boldsymbol{w}^* - (\boldsymbol{w}^\top \boldsymbol{w}^*) \boldsymbol{w}}{\|\boldsymbol{w}\|^3 \sqrt{1 - \frac{(\boldsymbol{w}^\top \boldsymbol{w}^*)^2}{\|\boldsymbol{w}\|^2}}}$$

$$= -\frac{\boldsymbol{v}^\top \boldsymbol{v}^*}{2\pi \|\boldsymbol{w}\|} \frac{\left( \boldsymbol{I}_n - \frac{\boldsymbol{w}\boldsymbol{w}^\top}{\|\boldsymbol{w}\|^2} \right) \boldsymbol{w}^*}{\left\| \left( \boldsymbol{I}_n - \frac{\boldsymbol{w}\boldsymbol{w}^\top}{\|\boldsymbol{w}\|^2} \right) \boldsymbol{w}^* \right\|}.$$

$\square$

**Proposition 1.** *If the true parameter $\boldsymbol{v}^*$ satisfies $(\mathbf{1}_m^\top \boldsymbol{v}^*)^2 < \frac{m+1}{2}\|\boldsymbol{v}^*\|^2$, then*

$$\left\{ (\boldsymbol{v}, \boldsymbol{w}) : \boldsymbol{v} = (\boldsymbol{I}_m + \mathbf{1}_m \mathbf{1}_m^\top)^{-1} \left( \frac{-(\mathbf{1}_m^\top \boldsymbol{v}^*)^2}{(m+1)\|\boldsymbol{v}^*\|^2 - (\mathbf{1}_m^\top \boldsymbol{v}^*)^2} \boldsymbol{I}_m + \mathbf{1}_m \mathbf{1}_m^\top \right) \boldsymbol{v}^*, \right.$$

$$\left. \theta(\boldsymbol{w}, \boldsymbol{w}^*) = \frac{\pi}{2} \frac{(m+1)\|\boldsymbol{v}^*\|^2}{(m+1)\|\boldsymbol{v}^*\|^2 - (\mathbf{1}_m^\top \boldsymbol{v}^*)^2} \right\}$$

*give the saddle points obeying (8), and $\{(\boldsymbol{v}, \boldsymbol{w}) : \theta(\boldsymbol{w}, \boldsymbol{w}^*) = \pi, \ \boldsymbol{v} = (\boldsymbol{I}_m + \mathbf{1}_m \mathbf{1}_m^\top)^{-1}(\mathbf{1}_m \mathbf{1}_m^\top - \boldsymbol{I}_m)\boldsymbol{v}^*\}$ are the spurious local minimizers. Otherwise, the model (2) has no saddle points or spurious local minimizers.*

**Proof of Proposition 1.** Suppose $\boldsymbol{v}^\top \boldsymbol{v}^* = 0$ and $\frac{\partial f}{\partial \boldsymbol{v}}(\boldsymbol{v}, \boldsymbol{w}) = \mathbf{0}$, then by Lemma 1,

$$0 = \boldsymbol{v}^\top \boldsymbol{v}^* = (\boldsymbol{v}^*)^\top (\boldsymbol{I}_m + \mathbf{1}_m \mathbf{1}_m^\top)^{-1} \left( \left(1 - \frac{2}{\pi}\theta(\boldsymbol{w}, \boldsymbol{w}^*)\right) \boldsymbol{I}_m + \mathbf{1}_m \mathbf{1}_m^\top \right) \boldsymbol{v}^*. \tag{18}$$

From (18) it follows that

$$\frac{2}{\pi}\theta(\boldsymbol{w}, \boldsymbol{w}^*)(\boldsymbol{v}^*)^\top (\boldsymbol{I}_m + \mathbf{1}_m \mathbf{1}_m^\top)^{-1}\boldsymbol{v}^* = (\boldsymbol{v}^*)^\top (\boldsymbol{I}_m + \mathbf{1}_m \mathbf{1}_m^\top)^{-1} \left( \boldsymbol{I}_m + \mathbf{1}_m \mathbf{1}_m^\top \right) \boldsymbol{v}^* = \|\boldsymbol{v}^*\|^2. \tag{19}$$

On the other hand, from (18) it also follows that

$$\left( \frac{2}{\pi}\theta(\boldsymbol{w}, \boldsymbol{w}^*) - 1 \right) (\boldsymbol{v}^*)^\top (\boldsymbol{I}_m + \mathbf{1}_m \mathbf{1}_m^\top)^{-1}\boldsymbol{v}^* = (\boldsymbol{v}^*)^\top (\boldsymbol{I}_m + \mathbf{1}_m \mathbf{1}_m^\top)^{-1}\mathbf{1}_m(\mathbf{1}_m^\top \boldsymbol{v}^*) = \frac{(\mathbf{1}^\top \boldsymbol{v}^*)^2}{m+1},$$

where we used $(\boldsymbol{I}_m + \mathbf{1}_m \mathbf{1}_m^\top)\mathbf{1}_m = (m+1)\mathbf{1}_m$. Taking the difference of the two equalities above gives

$$(\boldsymbol{v}^*)^\top (\boldsymbol{I}_m + \mathbf{1}_m \mathbf{1}_m^\top)^{-1}\boldsymbol{v}^* = \|\boldsymbol{v}^*\|^2 - \frac{(\mathbf{1}_m^\top \boldsymbol{v}^*)^2}{m+1}. \tag{20}$$

By (19), we have $\theta(\boldsymbol{w}, \boldsymbol{w}^*) = \frac{\pi}{2} \frac{(m+1)\|\boldsymbol{v}^*\|^2}{(m+1)\|\boldsymbol{v}^*\|^2 - (\mathbf{1}_m^\top \boldsymbol{v}^*)^2}$, which requires

$$\frac{\pi}{2} \frac{(m+1)\|\boldsymbol{v}^*\|^2}{(m+1)\|\boldsymbol{v}^*\|^2 - (\mathbf{1}_m^\top \boldsymbol{v}^*)^2} < \pi, \text{ or equivalently, } (\mathbf{1}_m^\top \boldsymbol{v}^*)^2 < \frac{m+1}{2}\|\boldsymbol{v}^*\|^2.$$

Furthermore, since $\frac{\partial f}{\partial \boldsymbol{v}}(\boldsymbol{v}, \boldsymbol{w}) = 0$, we have

$$\boldsymbol{v} = (\boldsymbol{I}_m + \mathbf{1}_m \mathbf{1}_m^\top)^{-1} \left( \left(1 - \frac{2}{\pi}\theta(\boldsymbol{w}, \boldsymbol{w}^*)\right) \boldsymbol{I}_m + \mathbf{1}_m \mathbf{1}_m^\top \right) \boldsymbol{v}^*$$

$$= (\boldsymbol{I}_m + \mathbf{1}_m \mathbf{1}_m^\top)^{-1} \left( \frac{-(\mathbf{1}_m^\top \boldsymbol{v}^*)^2}{(m+1)\|\boldsymbol{v}^*\|^2 - (\mathbf{1}_m^\top \boldsymbol{v}^*)^2} \boldsymbol{I}_m + \mathbf{1}_m \mathbf{1}_m^\top \right) \boldsymbol{v}^*.$$

Next, we check the local optimality of the stationary points. By ignoring the scaling and constant terms, we rewrite the objective function as

$$\tilde{f}(\boldsymbol{v}, \theta) := \boldsymbol{v}^\top \left( \boldsymbol{I}_m + \mathbf{1}_m \mathbf{1}_m^\top \right)\boldsymbol{v} - 2\boldsymbol{v}^\top \left( \left(1 - \frac{2}{\pi}\theta\right) \boldsymbol{I}_m + \mathbf{1}_m \mathbf{1}_m^\top \right) \boldsymbol{v}^*, \text{ for } \theta \in [0, \pi].$$

It is easy to check that its Hessian matrix

$$\nabla^2 \tilde{f}(\boldsymbol{v}, \theta) = \begin{bmatrix} 2(\boldsymbol{I}_m + \mathbf{1}_m \mathbf{1}_m^\top) & \frac{4}{\pi}\boldsymbol{v}^* \\ \frac{4}{\pi}(\boldsymbol{v}^*)^\top & 0 \end{bmatrix}$$

is indefinite. Therefore, the stationary points are saddle points.

Moreover, if $(\mathbf{1}_m^\top \boldsymbol{v}^*)^2 < \frac{m+1}{2}\|\boldsymbol{v}^*\|^2$, at the point $(\boldsymbol{v}, \theta) = ((\boldsymbol{I}_m + \mathbf{1}_m \mathbf{1}_m^\top)^{-1}(\mathbf{1}_m \mathbf{1}_m^\top - \boldsymbol{I}_m)\boldsymbol{v}^*, \pi)$, we have

$$\boldsymbol{v}^\top \boldsymbol{v}^* = (\boldsymbol{v}^*)^\top (\boldsymbol{I}_m + \mathbf{1}_m \mathbf{1}_m^\top)^{-1}(\mathbf{1}_m \mathbf{1}_m^\top - \boldsymbol{I}_m)\boldsymbol{v}^*$$

$$= \|\boldsymbol{v}^*\|^2 - 2(\boldsymbol{v}^*)^\top (\boldsymbol{I}_m + \mathbf{1}_m \mathbf{1}_m^\top)^{-1}\boldsymbol{v}^* = \frac{2(\mathbf{1}_m^\top \boldsymbol{v}^*)^2}{m+1} - \|\boldsymbol{v}^*\|^2 < 0, \tag{21}$$

where we used (20) in the last identity. We consider an arbitrary point $(\boldsymbol{v} + \Delta\boldsymbol{v}, \pi + \Delta\theta)$ in the neighborhood of $(\boldsymbol{v}, \pi)$ with $\Delta\theta \leq 0$. The perturbed objective value is

$$\tilde{f}(\boldsymbol{v} + \Delta\boldsymbol{v}, \pi + \Delta\theta) = (\boldsymbol{v} + \Delta\boldsymbol{v})^\top \left( \boldsymbol{I}_m + \boldsymbol{1}_m \boldsymbol{1}_m^\top \right)(\boldsymbol{v} + \Delta\boldsymbol{v}) - 2(\boldsymbol{v} + \Delta\boldsymbol{v})^\top \left( \boldsymbol{1}_m \boldsymbol{1}_m^\top - \boldsymbol{I}_m \right) \boldsymbol{v}^*$$
$$+ \frac{2\Delta\theta}{\pi}(\boldsymbol{v} + \Delta\boldsymbol{v})^\top \boldsymbol{v}^*.$$

On the right hand side, since $\boldsymbol{v} = (\boldsymbol{I}_m + \boldsymbol{1}_m \boldsymbol{1}_m^\top)^{-1}(\boldsymbol{1}_m \boldsymbol{1}_m^\top - \boldsymbol{I}_m)\boldsymbol{v}^*$ is the unique minimizer to the quadratic function $\tilde{f}(\boldsymbol{v}, \pi)$, we have if $\Delta\boldsymbol{v} \neq \boldsymbol{0}_m$,

$$(\boldsymbol{v} + \Delta\boldsymbol{v})^\top \left( \boldsymbol{I}_m + \boldsymbol{1}_m \boldsymbol{1}_m^\top \right)(\boldsymbol{v} + \Delta\boldsymbol{v}) - 2(\boldsymbol{v} + \Delta\boldsymbol{v})^\top \left( \boldsymbol{1}_m \boldsymbol{1}_m^\top - \boldsymbol{I}_m \right) \boldsymbol{v}^* > \tilde{f}(\boldsymbol{v}, \pi).$$

Moreover, for sufficiently small $\|\Delta\boldsymbol{v}\|$, it holds that $\Delta\theta \cdot (\boldsymbol{v} + \Delta\boldsymbol{v})^\top \boldsymbol{v}^* > 0$ for $\Delta\theta < 0$ because of (21). Therefore, $\tilde{f}(\boldsymbol{v} + \Delta\boldsymbol{v}, \pi + \Delta\theta) > \tilde{f}(\boldsymbol{v}, \pi)$ whenever $(\Delta\boldsymbol{v}, \Delta\theta)$ is small and non-zero, and $((\boldsymbol{I}_m + \boldsymbol{1}_m \boldsymbol{1}_m^\top)^{-1}(\boldsymbol{1}_m \boldsymbol{1}_m^\top - \boldsymbol{I}_m)\boldsymbol{v}^*, \pi)$ is a local minimizer of $\tilde{f}$.

To prove the second claim, suppose $(\boldsymbol{1}_m^\top \boldsymbol{v}^*)^2 \geq \frac{m+1}{2}\|\boldsymbol{v}^*\|^2$, then either $\frac{\partial f}{\partial \boldsymbol{w}}(\boldsymbol{v}, \boldsymbol{w})$ does not exist, or $\frac{\partial f}{\partial \boldsymbol{v}}(\boldsymbol{v}, \boldsymbol{w})$ and $\frac{\partial f}{\partial \boldsymbol{w}}(\boldsymbol{v}, \boldsymbol{w})$ do not vanish simultaneously, and thus there is no stationary point.

At the point $(\boldsymbol{v}, \theta) = ((\boldsymbol{I}_m + \boldsymbol{1}_m \boldsymbol{1}_m^\top)^{-1}(\boldsymbol{1}_m \boldsymbol{1}_m^\top - \boldsymbol{I}_m)\boldsymbol{v}^*, \pi)$, we have

$$\boldsymbol{v}^\top \boldsymbol{v}^* = \frac{2(\boldsymbol{1}_m^\top \boldsymbol{v}^*)^2}{m+1} - \|\boldsymbol{v}^*\|^2 \geq 0.$$

If $\boldsymbol{v}^\top \boldsymbol{v}^* > 0$, since $\nabla\tilde{f}(\boldsymbol{v}, \theta) = \frac{1}{4}[\boldsymbol{0}_m^\top, \frac{2}{\pi}\boldsymbol{v}^\top \boldsymbol{v}^*]^\top$, a small perturbation $[\boldsymbol{0}_m^\top, \Delta\theta]^\top$ with $\Delta\theta < 0$ will give a strictly decreased objective value, so $(\boldsymbol{v}, \theta) = \left((\boldsymbol{I}_m + \boldsymbol{1}_m \boldsymbol{1}_m^\top)^{-1}(\boldsymbol{1}_m \boldsymbol{1}_m^\top - \boldsymbol{I}_m)\boldsymbol{v}^*, \pi\right)$ is not a local minimizer. If $\boldsymbol{v}^\top \boldsymbol{v}^* = 0$, then $\nabla\tilde{f}(\boldsymbol{v}, \theta) = \boldsymbol{0}_{n+1}$, the same conclusion can be reached by examining the second order necessary condition. $\square$

**Lemma 3.** *For any differentiable points $(\boldsymbol{v}, \boldsymbol{w})$ and $(\tilde{\boldsymbol{v}}, \tilde{\boldsymbol{w}})$ with $\min\{\|\boldsymbol{w}\|, \|\tilde{\boldsymbol{w}}\|\} = c_{\boldsymbol{w}} > 0$ and $\max\{\|\boldsymbol{v}\|, \|\tilde{\boldsymbol{v}}\|\} = C_{\boldsymbol{v}}$, there exists a Lipschitz constant $L > 0$ depending on $C_{\boldsymbol{v}}$ and $c_{\boldsymbol{w}}$, such that*

$$\|\nabla f(\boldsymbol{v}, \boldsymbol{w}) - \nabla f(\tilde{\boldsymbol{v}}, \tilde{\boldsymbol{w}})\| \leq L\|(\boldsymbol{v}, \boldsymbol{w}) - (\tilde{\boldsymbol{v}}, \tilde{\boldsymbol{w}})\|.$$

**Proof of Lemma 3.** It is easy to check that $\|\boldsymbol{I}_m + \boldsymbol{1}_m \boldsymbol{1}_m^\top\| = m + 1$. Then

$$\left\| \frac{\partial f}{\partial \boldsymbol{v}}(\boldsymbol{v}, \boldsymbol{w}) - \frac{\partial f}{\partial \boldsymbol{v}}(\tilde{\boldsymbol{v}}, \tilde{\boldsymbol{w}}) \right\| = \frac{1}{4} \left\| (\boldsymbol{I}_m + \boldsymbol{1}_m \boldsymbol{1}_m^\top)(\boldsymbol{v} - \tilde{\boldsymbol{v}}) + \frac{2}{\pi}(\theta(\boldsymbol{w}, \boldsymbol{w}^*) - \theta(\tilde{\boldsymbol{w}}, \boldsymbol{w}^*))\boldsymbol{v}^* \right\|$$
$$\leq \frac{m+1}{4}\|\boldsymbol{v} - \tilde{\boldsymbol{v}}\| + \frac{\|\boldsymbol{v}^*\|}{2\pi}|\theta(\boldsymbol{w}, \boldsymbol{w}^*) - \theta(\tilde{\boldsymbol{w}}, \boldsymbol{w}^*)|$$
$$\leq \frac{m+1}{4}\|\boldsymbol{v} - \tilde{\boldsymbol{v}}\| + \frac{\|\boldsymbol{v}^*\|}{4c_{\boldsymbol{w}}}\|\boldsymbol{w} - \tilde{\boldsymbol{w}}\|$$
$$\leq \frac{1}{4}\left( m + 1 + \frac{\|\boldsymbol{v}^*\|}{c_{\boldsymbol{w}}} \right)\|(\boldsymbol{v}, \boldsymbol{w}) - (\tilde{\boldsymbol{v}}, \tilde{\boldsymbol{w}})\|,$$

where the last inequality is due to Lemma 14.1.

We further have

$$
\begin{aligned}
\left\| \frac{\partial f}{\partial \boldsymbol{w}}(\boldsymbol{v}, \boldsymbol{w}) - \frac{\partial f}{\partial \boldsymbol{w}}(\tilde{\boldsymbol{v}}, \tilde{\boldsymbol{w}}) \right\| &= \left\| \frac{\boldsymbol{v}^\top \boldsymbol{v}^*}{2\pi \|\boldsymbol{w}\|} \frac{\left(\boldsymbol{I}_n - \frac{\boldsymbol{w}\boldsymbol{w}^\top}{\|\boldsymbol{w}\|^2}\right)\boldsymbol{w}^*}{\left\|\left(\boldsymbol{I}_n - \frac{\boldsymbol{w}\boldsymbol{w}^\top}{\|\boldsymbol{w}\|^2}\right)\boldsymbol{w}^*\right\|} - \frac{\tilde{\boldsymbol{v}}^\top \boldsymbol{v}^*}{2\pi \|\tilde{\boldsymbol{w}}\|} \frac{\left(\boldsymbol{I}_n - \frac{\tilde{\boldsymbol{w}}\tilde{\boldsymbol{w}}^\top}{\|\tilde{\boldsymbol{w}}\|^2}\right)\boldsymbol{w}^*}{\left\|\left(\boldsymbol{I}_n - \frac{\tilde{\boldsymbol{w}}\tilde{\boldsymbol{w}}^\top}{\|\tilde{\boldsymbol{w}}\|^2}\right)\boldsymbol{w}^*\right\|} \right\| \\
&\leq \left\| \frac{\boldsymbol{v}^\top \boldsymbol{v}^*}{2\pi \|\boldsymbol{w}\|} \frac{\left(\boldsymbol{I}_n - \frac{\boldsymbol{w}\boldsymbol{w}^\top}{\|\boldsymbol{w}\|^2}\right)\boldsymbol{w}^*}{\left\|\left(\boldsymbol{I}_n - \frac{\boldsymbol{w}\boldsymbol{w}^\top}{\|\boldsymbol{w}\|^2}\right)\boldsymbol{w}^*\right\|} - \frac{\boldsymbol{v}^\top \boldsymbol{v}^*}{2\pi \|\tilde{\boldsymbol{w}}\|} \frac{\left(\boldsymbol{I}_n - \frac{\tilde{\boldsymbol{w}}\tilde{\boldsymbol{w}}^\top}{\|\tilde{\boldsymbol{w}}\|^2}\right)\boldsymbol{w}^*}{\left\|\left(\boldsymbol{I}_n - \frac{\tilde{\boldsymbol{w}}\tilde{\boldsymbol{w}}^\top}{\|\tilde{\boldsymbol{w}}\|^2}\right)\boldsymbol{w}^*\right\|} \right\| \\
&\quad + \left\| \frac{\boldsymbol{v}^\top \boldsymbol{v}^*}{2\pi \|\tilde{\boldsymbol{w}}\|} \frac{\left(\boldsymbol{I}_n - \frac{\tilde{\boldsymbol{w}}\tilde{\boldsymbol{w}}^\top}{\|\tilde{\boldsymbol{w}}\|^2}\right)\boldsymbol{w}^*}{\left\|\left(\boldsymbol{I}_n - \frac{\tilde{\boldsymbol{w}}\tilde{\boldsymbol{w}}^\top}{\|\tilde{\boldsymbol{w}}\|^2}\right)\boldsymbol{w}^*\right\|} - \frac{\tilde{\boldsymbol{v}}^\top \boldsymbol{v}^*}{2\pi \|\tilde{\boldsymbol{w}}\|} \frac{\left(\boldsymbol{I}_n - \frac{\tilde{\boldsymbol{w}}\tilde{\boldsymbol{w}}^\top}{\|\tilde{\boldsymbol{w}}\|^2}\right)\boldsymbol{w}^*}{\left\|\left(\boldsymbol{I}_n - \frac{\tilde{\boldsymbol{w}}\tilde{\boldsymbol{w}}^\top}{\|\tilde{\boldsymbol{w}}\|^2}\right)\boldsymbol{w}^*\right\|} \right\| \\
&\leq \frac{|\boldsymbol{v}^\top \boldsymbol{v}^*|}{2\pi c_{\boldsymbol{w}}^2} \|\boldsymbol{w} - \tilde{\boldsymbol{w}}\| + \frac{\|\boldsymbol{v}^*\|}{2\pi c_{\boldsymbol{w}}} \|\boldsymbol{v} - \tilde{\boldsymbol{v}}\| \\
&\leq \frac{(C_{\boldsymbol{v}} + c_{\boldsymbol{w}})\|\boldsymbol{v}^*\|}{2\pi c_{\boldsymbol{w}}^2} \|(\boldsymbol{v}, \boldsymbol{w}) - (\tilde{\boldsymbol{v}}, \tilde{\boldsymbol{w}})\|,
\end{aligned}
$$

where the second last inequality is to due to Lemma 14.2. Combining the two inequalities above validates the claim. □

**Lemma 4.** *The expected partial gradient of $\ell(\boldsymbol{v}, \boldsymbol{w}; \mathbf{Z})$ w.r.t. $\boldsymbol{v}$ is*

$$
\mathbb{E}_{\mathbf{Z}} \left[ \frac{\partial \ell}{\partial \boldsymbol{v}}(\boldsymbol{v}, \boldsymbol{w}; \mathbf{Z}) \right] = \frac{\partial f}{\partial \boldsymbol{v}}(\boldsymbol{v}, \boldsymbol{w}).
$$

*Let $\mu(x) = \max\{x, 0\}$ in (5). The expected coarse gradient w.r.t. $\boldsymbol{w}$ is*

$$
\mathbb{E}_{\mathbf{Z}} \left[ \boldsymbol{g}_{\mathrm{relu}}(\boldsymbol{v}, \boldsymbol{w}; \mathbf{Z}) \right] = \frac{h(\boldsymbol{v}, \boldsymbol{v}^*)}{2\sqrt{2\pi}} \frac{\boldsymbol{w}}{\|\boldsymbol{w}\|} - \cos\left(\frac{\theta(\boldsymbol{w}, \boldsymbol{w}^*)}{2}\right) \frac{\boldsymbol{v}^\top \boldsymbol{v}^*}{\sqrt{2\pi}} \frac{\frac{\boldsymbol{w}}{\|\boldsymbol{w}\|} + \boldsymbol{w}^*}{\left\| \frac{\boldsymbol{w}}{\|\boldsymbol{w}\|} + \boldsymbol{w}^* \right\|},^3
$$

*where $h(\boldsymbol{v}, \boldsymbol{v}^*) = \|\boldsymbol{v}\|^2 + (\mathbf{1}_m^\top \boldsymbol{v})^2 - (\mathbf{1}_m^\top \boldsymbol{v})(\mathbf{1}_m^\top \boldsymbol{v}^*) + \boldsymbol{v}^\top \boldsymbol{v}^*$.*

**Proof of Lemma 4.** The first claim is true because $\frac{\partial \ell}{\partial \boldsymbol{v}}(\boldsymbol{v}, \boldsymbol{w}; \mathbf{Z})$ is linear in $\boldsymbol{v}$. By (5),

$$
\boldsymbol{g}_\mu(\boldsymbol{v}, \boldsymbol{w}; \mathbf{Z}) = \mathbf{Z}^\top \left( \mu'(\mathbf{Z}\boldsymbol{w}) \odot \boldsymbol{v} \right) \left( \boldsymbol{v}^\top \sigma(\mathbf{Z}\boldsymbol{w}) - (\boldsymbol{v}^*)^\top \sigma(\mathbf{Z}\boldsymbol{w}^*) \right).
$$

Using the fact that $\mu' = \sigma = 1_{\{x > 0\}}$, we have

$$
\begin{aligned}
\mathbb{E}_{\mathbf{Z}} \left[ \boldsymbol{g}_{\mathrm{relu}}(\boldsymbol{v}, \boldsymbol{w}; \mathbf{Z}) \right] &= \mathbb{E}_{\mathbf{Z}} \left[ \left( \sum_{i=1}^m v_i \sigma(\mathbf{Z}_i^\top \boldsymbol{w}) - \sum_{i=1}^m v_i^* \sigma(\mathbf{Z}_i^\top \boldsymbol{w}^*) \right) \left( \sum_{i=1}^m \mathbf{Z}_i v_i \sigma(\mathbf{Z}_i^\top \boldsymbol{w}) \right) \right] \\
&= \mathbb{E}_{\mathbf{Z}} \left[ \left( \sum_{i=1}^m v_i 1_{\{\mathbf{Z}_i^\top \boldsymbol{w} > 0\}} - \sum_{i=1}^m v_i^* 1_{\{\mathbf{Z}_i^\top \boldsymbol{w}^* > 0\}} \right) \left( \sum_{i=1}^m 1_{\{\mathbf{Z}_i^\top \boldsymbol{w} > 0\}} v_i \mathbf{Z}_i \right) \right].
\end{aligned}
$$

Invoking Lemma 11, we have

$$
\mathbb{E} \left[ \mathbf{Z}_i 1_{\{\mathbf{Z}_i^\top \boldsymbol{w} > 0, \mathbf{Z}_j^\top \boldsymbol{w} > 0\}} \right] = \begin{cases} \frac{1}{\sqrt{2\pi}} \frac{\boldsymbol{w}}{\|\boldsymbol{w}\|} & \text{if } i = j, \\ \frac{1}{2\sqrt{2\pi}} \frac{\boldsymbol{w}}{\|\boldsymbol{w}\|} & \text{if } i \neq j, \end{cases}
$$

and

$$
\mathbb{E} \left[ \mathbf{Z}_i 1_{\{\mathbf{Z}_i^\top \boldsymbol{w} > 0, \mathbf{Z}_j^\top \boldsymbol{w}^* > 0\}} \right] = \begin{cases} \frac{\cos(\theta(\boldsymbol{w}, \boldsymbol{w}^*)/2)}{\sqrt{2\pi}} \frac{\frac{\boldsymbol{w}}{\|\boldsymbol{w}\|} + \boldsymbol{w}^*}{\left\| \frac{\boldsymbol{w}}{\|\boldsymbol{w}\|} + \boldsymbol{w}^* \right\|} & \text{if } i = j, \\ \frac{1}{2\sqrt{2\pi}} \frac{\boldsymbol{w}}{\|\boldsymbol{w}\|} & \text{if } i \neq j. \end{cases}
$$

___
[3] We redefine the second term as $\mathbf{0}_n$ in the case $\theta(\boldsymbol{w}, \boldsymbol{w}^*) = \pi$, or equivalently, $\frac{\boldsymbol{w}}{\|\boldsymbol{w}\|} + \boldsymbol{w}^* = \mathbf{0}_n$.

Therefore,

$$
\begin{aligned}
\mathbb{E}_{\mathbf{Z}}\left[\boldsymbol{g}_{\mathrm{relu}}(\boldsymbol{v},\boldsymbol{w};\mathbf{Z})\right] = {} & \sum_{i=1}^{m} v_i^2 \mathbb{E}\left[\mathbf{Z}_i \mathbf{1}_{\{\mathbf{Z}_i^\top \boldsymbol{w}>0\}}\right] + \sum_{i=1}^{m}\sum_{\substack{j=1\\j\neq i}}^{m} v_i v_j \mathbb{E}\left[\mathbf{Z}_i \mathbf{1}_{\{\mathbf{Z}_i^\top \boldsymbol{w}>0,\mathbf{Z}_j^\top \boldsymbol{w}>0\}}\right] \\
& - \sum_{i=1}^{m} v_i v_i^* \mathbb{E}\left[\mathbf{Z}_i \mathbf{1}_{\{\mathbf{Z}_i^\top \boldsymbol{w}>0,\mathbf{Z}_i^\top \boldsymbol{w}^*>0\}}\right] \\
& - \sum_{i=1}^{m}\sum_{\substack{j=1\\j\neq i}}^{m} v_i v_j^* \mathbb{E}\left[\mathbf{Z}_i \mathbf{1}_{\{\mathbf{Z}_i^\top \boldsymbol{w}>0,\mathbf{Z}_j^\top \boldsymbol{w}^*>0\}}\right] \\
= {} & \frac{1}{2\sqrt{2\pi}}\left(\|\boldsymbol{v}\|^2 + (\mathbf{1}_m^\top \boldsymbol{v})^2\right)\frac{\boldsymbol{w}}{\|\boldsymbol{w}\|} - \cos\left(\frac{\theta(\boldsymbol{w},\boldsymbol{w}^*)}{2}\right)\frac{\boldsymbol{v}^\top \boldsymbol{v}^*}{\sqrt{2\pi}}\frac{\frac{\boldsymbol{w}}{\|\boldsymbol{w}\|}+\boldsymbol{w}^*}{\left\|\frac{\boldsymbol{w}}{\|\boldsymbol{w}\|}+\boldsymbol{w}^*\right\|} \\
& - \frac{1}{2\sqrt{2\pi}}\left((\mathbf{1}_m^\top \boldsymbol{v})(\mathbf{1}_m^\top \boldsymbol{v}^*) - \boldsymbol{v}^\top \boldsymbol{v}^*\right)\frac{\boldsymbol{w}}{\|\boldsymbol{w}\|},
\end{aligned}
$$

and the result follows. $\qquad\square$

**Lemma 5.** *If $\boldsymbol{w} \neq \mathbf{0}_n$ and $\theta(\boldsymbol{w},\boldsymbol{w}^*) \in (0,\pi)$, then the inner product between the expected coarse and true gradients w.r.t. $\boldsymbol{w}$ is*

$$
\left\langle \mathbb{E}_{\mathbf{Z}}\left[\boldsymbol{g}_{\mathrm{relu}}(\boldsymbol{v},\boldsymbol{w};\mathbf{Z})\right], \frac{\partial f}{\partial \boldsymbol{w}}(\boldsymbol{v},\boldsymbol{w})\right\rangle = \frac{\sin\left(\theta(\boldsymbol{w},\boldsymbol{w}^*)\right)}{2(\sqrt{2\pi})^3\|\boldsymbol{w}\|}(\boldsymbol{v}^\top \boldsymbol{v}^*)^2 \geq 0.
$$

*Moreover, if further $\|\boldsymbol{v}\| \leq C_{\boldsymbol{v}}$ and $\|\boldsymbol{w}\| \geq c_{\boldsymbol{w}}$, there exists a constant $A_{\mathrm{relu}} > 0$ depending on $C_{\boldsymbol{v}}$ and $c_{\boldsymbol{w}}$, such that*

$$
\left\|\mathbb{E}_{\mathbf{Z}}\left[\boldsymbol{g}_{\mathrm{relu}}(\boldsymbol{v},\boldsymbol{w};\mathbf{Z})\right]\right\|^2 \leq A_{\mathrm{relu}}\left(\left\|\frac{\partial f}{\partial \boldsymbol{v}}(\boldsymbol{v},\boldsymbol{w})\right\|^2 + \left\langle \mathbb{E}_{\mathbf{Z}}\left[\boldsymbol{g}_{\mathrm{relu}}(\boldsymbol{v},\boldsymbol{w};\mathbf{Z})\right], \frac{\partial f}{\partial \boldsymbol{w}}(\boldsymbol{v},\boldsymbol{w})\right\rangle\right).
$$

**Proof of Lemma 5.** By Lemmas 2 and 4, we have

$$
\frac{\partial f}{\partial \boldsymbol{w}}(\boldsymbol{v},\boldsymbol{w}) = -\frac{\boldsymbol{v}^\top \boldsymbol{v}^*}{2\pi\|\boldsymbol{w}\|}\frac{\left(\boldsymbol{I}_n - \frac{\boldsymbol{w}\boldsymbol{w}^\top}{\|\boldsymbol{w}\|^2}\right)\boldsymbol{w}^*}{\left\|\left(\boldsymbol{I}_n - \frac{\boldsymbol{w}\boldsymbol{w}^\top}{\|\boldsymbol{w}\|^2}\right)\boldsymbol{w}^*\right\|}
$$

and

$$
\mathbb{E}_{\mathbf{Z}}\left[\boldsymbol{g}_{\mathrm{relu}}(\boldsymbol{v},\boldsymbol{w};\mathbf{Z})\right] = \frac{h(\boldsymbol{v},\boldsymbol{v}^*)}{2\sqrt{2\pi}}\frac{\boldsymbol{w}}{\|\boldsymbol{w}\|} - \cos\left(\frac{\theta(\boldsymbol{w},\boldsymbol{w}^*)}{2}\right)\frac{\boldsymbol{v}^\top \boldsymbol{v}^*}{\sqrt{2\pi}}\frac{\frac{\boldsymbol{w}}{\|\boldsymbol{w}\|}+\boldsymbol{w}^*}{\left\|\frac{\boldsymbol{w}}{\|\boldsymbol{w}\|}+\boldsymbol{w}^*\right\|}.
$$

Notice that $\left(\boldsymbol{I}_n - \frac{\boldsymbol{w}\boldsymbol{w}^\top}{\|\boldsymbol{w}\|^2}\right)\boldsymbol{w} = \boldsymbol{0}_n$ and $\|\boldsymbol{w}^*\| = 1$, if $\theta(\boldsymbol{w}, \boldsymbol{w}_*) \neq 0, \pi$, then we have

$$
\begin{aligned}
&\left\langle \mathbb{E}_{\mathbf{Z}}\left[\boldsymbol{g}_{\mathrm{relu}}(\boldsymbol{v}, \boldsymbol{w}; \mathbf{Z})\right], \frac{\partial f}{\partial \boldsymbol{w}}(\boldsymbol{v}, \boldsymbol{w})\right\rangle \\
&= \cos\left(\frac{\theta(\boldsymbol{w}, \boldsymbol{w}^*)}{2}\right)\frac{(\boldsymbol{v}^\top\boldsymbol{v}^*)^2}{(\sqrt{2\pi})^3}\left\langle \frac{1}{\|\boldsymbol{w}\|}\frac{\left(\boldsymbol{I}_n - \frac{\boldsymbol{w}\boldsymbol{w}^\top}{\|\boldsymbol{w}\|^2}\right)\boldsymbol{w}^*}{\left\|\left(\boldsymbol{I}_n - \frac{\boldsymbol{w}\boldsymbol{w}^\top}{\|\boldsymbol{w}\|^2}\right)\boldsymbol{w}^*\right\|}, \frac{\boldsymbol{w}^*}{\left\|\frac{\boldsymbol{w}}{\|\boldsymbol{w}\|} + \boldsymbol{w}^*\right\|}\right\rangle \\
&= \cos\left(\frac{\theta(\boldsymbol{w}, \boldsymbol{w}^*)}{2}\right)\frac{(\boldsymbol{v}^\top\boldsymbol{v}^*)^2}{(\sqrt{2\pi})^3}\frac{\|\boldsymbol{w}\|^2 - (\boldsymbol{w}^\top\boldsymbol{w}^*)^2}{\|\|\boldsymbol{w}\|^2\boldsymbol{w}^* - \boldsymbol{w}(\boldsymbol{w}^\top\boldsymbol{w}^*)\|\,\|\boldsymbol{w} + \|\boldsymbol{w}\|\boldsymbol{w}^*\|} \\
&= \cos\left(\frac{\theta(\boldsymbol{w}, \boldsymbol{w}^*)}{2}\right)\frac{(\boldsymbol{v}^\top\boldsymbol{v}^*)^2}{(\sqrt{2\pi})^3}\frac{\|\boldsymbol{w}\|^2 - (\boldsymbol{w}^\top\boldsymbol{w}^*)^2}{\sqrt{\|\boldsymbol{w}\|^4 - \|\boldsymbol{w}\|^2(\boldsymbol{w}^\top\boldsymbol{w}^*)^2}\sqrt{2(\|\boldsymbol{w}\|^2 + \|\boldsymbol{w}\|(\boldsymbol{w}^\top\boldsymbol{w}^*))}} \\
&= \cos\left(\frac{\theta(\boldsymbol{w}, \boldsymbol{w}^*)}{2}\right)\frac{(\boldsymbol{v}^\top\boldsymbol{v}^*)^2}{4(\sqrt{\pi}\|\boldsymbol{w}\|)^3}\frac{\|\boldsymbol{w}\|^2 - (\boldsymbol{w}^\top\boldsymbol{w}^*)^2}{\sqrt{\|\boldsymbol{w}\|^2 - (\boldsymbol{w}^\top\boldsymbol{w}^*)^2}\sqrt{\|\boldsymbol{w}\| + (\boldsymbol{w}^\top\boldsymbol{w}^*)}} \\
&= \cos\left(\frac{\theta(\boldsymbol{w}, \boldsymbol{w}^*)}{2}\right)\frac{(\boldsymbol{v}^\top\boldsymbol{v}^*)^2\sqrt{1 - \frac{\boldsymbol{w}^\top\boldsymbol{w}^*}{\|\boldsymbol{w}\|}}}{4(\sqrt{\pi})^3\|\boldsymbol{w}\|} \\
&= \cos\left(\frac{\theta(\boldsymbol{w}, \boldsymbol{w}^*)}{2}\right)\frac{(\boldsymbol{v}^\top\boldsymbol{v}^*)^2\sqrt{1 - \cos(\theta(\boldsymbol{w}, \boldsymbol{w}^*))}}{4(\sqrt{\pi})^3\|\boldsymbol{w}\|} \\
&= \frac{\sin(\theta(\boldsymbol{w}, \boldsymbol{w}^*))}{2(\sqrt{2\pi})^3\|\boldsymbol{w}\|}(\boldsymbol{v}^\top\boldsymbol{v}^*)^2.
\end{aligned}
$$

To show the second claim, without loss of generality, we assume $\|\boldsymbol{w}\| = 1$. Denote $\theta := \theta(\boldsymbol{w}, \boldsymbol{w}^*)$. By Lemma 1, we have

$$
\frac{\partial f}{\partial \boldsymbol{v}}(\boldsymbol{v}, \boldsymbol{w}) = \frac{1}{4}\left(\boldsymbol{I}_m + \boldsymbol{1}_m\boldsymbol{1}_m^\top\right)\boldsymbol{v} - \frac{1}{4}\left(\left(1 - \frac{2\theta}{\pi}\right)\boldsymbol{I}_m + \boldsymbol{1}_m\boldsymbol{1}_m^\top\right)\boldsymbol{v}^*.
$$

By Lemma 4,

$$
\mathbb{E}_{\mathbf{Z}}\left[\boldsymbol{g}_{\mathrm{relu}}(\boldsymbol{v}, \boldsymbol{w}; \mathbf{Z})\right] = \frac{h(\boldsymbol{v}, \boldsymbol{v}^*)}{2\sqrt{2\pi}}\frac{\boldsymbol{w}}{\|\boldsymbol{w}\|} - \cos\left(\frac{\theta}{2}\right)\frac{\boldsymbol{v}^\top\boldsymbol{v}^*}{\sqrt{2\pi}}\frac{\frac{\boldsymbol{w}}{\|\boldsymbol{w}\|} + \boldsymbol{w}^*}{\left\|\frac{\boldsymbol{w}}{\|\boldsymbol{w}\|} + \boldsymbol{w}^*\right\|}, \tag{22}
$$

where

$$
\begin{aligned}
h(\boldsymbol{v}, \boldsymbol{v}^*) &= \|\boldsymbol{v}\|^2 + (\boldsymbol{1}_m^\top\boldsymbol{v})^2 - (\boldsymbol{1}_m^\top\boldsymbol{v})(\boldsymbol{1}_m^\top\boldsymbol{v}^*) + \boldsymbol{v}^\top\boldsymbol{v}^* \\
&= \boldsymbol{v}^\top\left(\boldsymbol{I}_m + \boldsymbol{1}_m\boldsymbol{1}_m^\top\right)\boldsymbol{v} - \boldsymbol{v}^\top(\boldsymbol{1}_m\boldsymbol{1}_m^\top - \boldsymbol{I}_m)\boldsymbol{v}^* \\
&= \boldsymbol{v}^\top\left(\boldsymbol{I}_m + \boldsymbol{1}_m\boldsymbol{1}_m^\top\right)\boldsymbol{v} - \boldsymbol{v}^\top\left(\boldsymbol{1}_m\boldsymbol{1}_m^\top + \left(1 - \frac{2\theta}{\pi}\right)\boldsymbol{I}_m\right)\boldsymbol{v}^* + 2\left(1 - \frac{\theta}{\pi}\right)\boldsymbol{v}^\top\boldsymbol{v}^* \\
&= 4\boldsymbol{v}^\top\frac{\partial f}{\partial \boldsymbol{v}}(\boldsymbol{v}, \boldsymbol{w}) + 2\left(1 - \frac{\theta}{\pi}\right)\boldsymbol{v}^\top\boldsymbol{v}^*, \tag{23}
\end{aligned}
$$

and by the first claim,

$$
\left\langle \mathbb{E}_{\mathbf{Z}}\left[\boldsymbol{g}_{\mathrm{relu}}(\boldsymbol{v}, \boldsymbol{w}; \mathbf{Z})\right], \frac{\partial f}{\partial \boldsymbol{w}}(\boldsymbol{v}, \boldsymbol{w})\right\rangle = \frac{\sin(\theta)}{2(\sqrt{2\pi})^3\|\boldsymbol{w}\|}(\boldsymbol{v}^\top\boldsymbol{v}^*)^2.
$$

Hence, for some $A_{\mathrm{relu}}$ depending only on $C_{\boldsymbol{v}}$ and $c_{\boldsymbol{w}}$, we have

$$
\left\| \mathbb{E}_{\mathbf{Z}}\Big[\boldsymbol{g}_{\mathrm{relu}}(\boldsymbol{v},\boldsymbol{w};\mathbf{Z})\Big] \right\|^2
$$

$$
= \left\| \frac{2\boldsymbol{v}^\top \frac{\partial f}{\partial \boldsymbol{v}}(\boldsymbol{v},\boldsymbol{w})}{\sqrt{2\pi}}\frac{\boldsymbol{w}}{\|\boldsymbol{w}\|} + \cos\left(\frac{\theta}{2}\right)\frac{\boldsymbol{v}^\top \boldsymbol{v}^*}{\sqrt{2\pi}}\left(\frac{\boldsymbol{w}}{\|\boldsymbol{w}\|} - \frac{\frac{\boldsymbol{w}}{\|\boldsymbol{w}\|}+\boldsymbol{w}^*}{\left\|\frac{\boldsymbol{w}}{\|\boldsymbol{w}\|}+\boldsymbol{w}^*\right\|}\right) \right.
$$

$$
\left. + \left(1 - \frac{\theta}{\pi} - \cos\left(\frac{\theta}{2}\right)\right)\frac{\boldsymbol{v}^\top \boldsymbol{v}^*}{\sqrt{2\pi}}\frac{\boldsymbol{w}}{\|\boldsymbol{w}\|} \right\|^2
$$

$$
\leq \frac{6C_{\boldsymbol{v}}^2}{\pi}\left\|\frac{\partial f}{\partial \boldsymbol{v}}(\boldsymbol{v},\boldsymbol{w})\right\|^2 + \cos^2\left(\frac{\theta}{2}\right)\frac{3(\boldsymbol{v}^\top \boldsymbol{v}^*)^2}{2\pi}\left\|\frac{\boldsymbol{w}}{\|\boldsymbol{w}\|} - \frac{\frac{\boldsymbol{w}}{\|\boldsymbol{w}\|}+\boldsymbol{w}^*}{\left\|\frac{\boldsymbol{w}}{\|\boldsymbol{w}\|}+\boldsymbol{w}^*\right\|}\right\|^2
$$

$$
+ \left(1 - \frac{\theta}{\pi} - \cos\left(\frac{\theta}{2}\right)\right)^2\frac{3(\boldsymbol{v}^\top \boldsymbol{v}^*)^2}{2\pi}
$$

$$
\leq \frac{6C_{\boldsymbol{v}}^2}{\pi}\left\|\frac{\partial f}{\partial \boldsymbol{v}}(\boldsymbol{v},\boldsymbol{w})\right\|^2 + \cos^2\left(\frac{\theta}{2}\right)\frac{3\theta^2}{8\pi}(\boldsymbol{v}^\top \boldsymbol{v}^*)^2 + \left(1 - \frac{\theta}{\pi} - \cos\left(\frac{\theta}{2}\right)\right)^2\frac{3(\boldsymbol{v}^\top \boldsymbol{v}^*)^2}{2\pi}
$$

$$
\leq \frac{6C_{\boldsymbol{v}}^2}{\pi}\left\|\frac{\partial f}{\partial \boldsymbol{v}}(\boldsymbol{v},\boldsymbol{w})\right\|^2 + \frac{3\pi}{8}\cos^2\left(\frac{\theta}{2}\right)\sin^2\left(\frac{\theta}{2}\right)(\boldsymbol{v}^\top \boldsymbol{v}^*)^2 + \frac{3\sin(\theta)}{2\pi}(\boldsymbol{v}^\top \boldsymbol{v}^*)^2
$$

$$
\leq A_{\mathrm{relu}}\left(\left\|\frac{\partial f}{\partial \boldsymbol{v}}(\boldsymbol{v},\boldsymbol{w})\right\|^2 + \left\langle \mathbb{E}_{\mathbf{Z}}\Big[\boldsymbol{g}_{\mathrm{relu}}(\boldsymbol{v},\boldsymbol{w};\mathbf{Z})\Big], \frac{\partial f}{\partial \boldsymbol{w}}(\boldsymbol{v},\boldsymbol{w})\right\rangle\right),
$$

where the equality is due to (22) and (23), the first inequality is due to Cauchy-Schwarz inequality, the second inequality holds because the angle between $\frac{\boldsymbol{w}}{\|\boldsymbol{w}\|}$ and $\frac{\frac{\boldsymbol{w}}{\|\boldsymbol{w}\|}+\boldsymbol{w}^*}{\left\|\frac{\boldsymbol{w}}{\|\boldsymbol{w}\|}+\boldsymbol{w}^*\right\|}$ is $\frac{\theta}{2}$ and $\left\|\frac{\boldsymbol{w}}{\|\boldsymbol{w}\|} - \frac{\frac{\boldsymbol{w}}{\|\boldsymbol{w}\|}+\boldsymbol{w}^*}{\left\|\frac{\boldsymbol{w}}{\|\boldsymbol{w}\|}+\boldsymbol{w}^*\right\|}\right\| \leq \frac{\theta}{2}$, whereas the third inequality is due to $\sin(x) \geq \frac{2x}{\pi}$, $\cos(x) \geq 1 - \frac{2x}{\pi}$, and

$$
\left(1 - \frac{2x}{\pi} - \cos(x)\right)^2 \leq \left(\cos(x) - 1 + \frac{2x}{\pi}\right)\left(\cos(x) + 1 - \frac{2x}{\pi}\right) \leq \sin(x)(2\cos(x)) = \sin(2x)
$$

for all $x \in [0, \frac{\pi}{2}]$. $\qquad \square$

**Lemma 6.** *When Algorithm 1 converges, $\mathbb{E}_{\mathbf{Z}}\left[\frac{\partial \ell}{\partial \boldsymbol{v}}(\boldsymbol{v},\boldsymbol{w};\mathbf{Z})\right]$ and $\mathbb{E}_{\mathbf{Z}}\left[\boldsymbol{g}_{\mathrm{relu}}(\boldsymbol{v},\boldsymbol{w};\mathbf{Z})\right]$ vanish simultaneously, which only occurs at the*

1. *Saddle points where (8) is satisfied according to Proposition 1.*

2. *Minimizers of (2) where $\boldsymbol{v} = \boldsymbol{v}^*$, $\theta(\boldsymbol{w},\boldsymbol{w}^*) = 0$, or $\boldsymbol{v} = (\boldsymbol{I}_m + \mathbf{1}_m \mathbf{1}_m^\top)^{-1}(\mathbf{1}_m \mathbf{1}_m^\top - \boldsymbol{I}_m)\boldsymbol{v}^*$, $\theta(\boldsymbol{w},\boldsymbol{w}^*) = \pi$.*

**Proof of Lemma 6.** By Lemma 4, suppose we have

$$
\mathbb{E}_{\mathbf{Z}}\left[\frac{\partial \ell}{\partial \boldsymbol{v}}(\boldsymbol{v},\boldsymbol{w};\mathbf{Z})\right] = \frac{1}{4}\left(\boldsymbol{I}_m + \mathbf{1}_m \mathbf{1}_m^\top\right)\boldsymbol{v} - \frac{1}{4}\left(\left(1 - \frac{2}{\pi}\theta(\boldsymbol{w},\boldsymbol{w}^*)\right)\boldsymbol{I}_m + \mathbf{1}_m \mathbf{1}_m^\top\right)\boldsymbol{v}^* = \mathbf{0}_m \quad (24)
$$

and

$$
\mathbb{E}_{\mathbf{Z}}\left[\boldsymbol{g}_{\mathrm{relu}}(\boldsymbol{v},\boldsymbol{w};\mathbf{Z})\right] = \frac{h(\boldsymbol{v},\boldsymbol{v}^*)}{2\sqrt{2\pi}}\frac{\boldsymbol{w}}{\|\boldsymbol{w}\|} - \cos\left(\frac{\theta(\boldsymbol{w},\boldsymbol{w}^*)}{2}\right)\frac{\boldsymbol{v}^\top \boldsymbol{v}^*}{\sqrt{2\pi}}\frac{\frac{\boldsymbol{w}}{\|\boldsymbol{w}\|} + \boldsymbol{w}^*}{\left\|\frac{\boldsymbol{w}}{\|\boldsymbol{w}\|} + \boldsymbol{w}^*\right\|} = \mathbf{0}_n, \quad (25)
$$

where $h(\boldsymbol{v},\boldsymbol{v}^*) = \|\boldsymbol{v}\|^2 + (\mathbf{1}_m^\top \boldsymbol{v})^2 - (\mathbf{1}_m^\top \boldsymbol{v})(\mathbf{1}_m^\top \boldsymbol{v}^*) + \boldsymbol{v}^\top \boldsymbol{v}^*$. By (25), we must have $\theta(\boldsymbol{w},\boldsymbol{w}^*) = 0$ or $\theta(\boldsymbol{w},\boldsymbol{w}^*) = \pi$ or $\boldsymbol{v}^\top \boldsymbol{v}^* = 0$.

If $\theta(\boldsymbol{w}, \boldsymbol{w}^*) = 0$, then by (24), $\boldsymbol{v} = \boldsymbol{v}^*$, and (25) is satisfied.

If $\theta(\boldsymbol{w}, \boldsymbol{w}^*) = \pi$, then by (24), $\boldsymbol{v} = (\boldsymbol{I}_m + \boldsymbol{1}_m \boldsymbol{1}_m^\top)^{-1}(\boldsymbol{1}_m \boldsymbol{1}_m^\top - \boldsymbol{I}_m)\boldsymbol{v}^*$, and (25) is satisfied.

If $\boldsymbol{v}^\top \boldsymbol{v}^* = 0$, then by (24), we have the expressions for $\boldsymbol{v}$ and $\theta(\boldsymbol{w}, \boldsymbol{w}^*)$ from Proposition 1, and (25) is satisfied. $\qquad\square$

**Lemma 7.** *If $\boldsymbol{w} \neq \boldsymbol{0}_n$ and $\theta(\boldsymbol{w}, \boldsymbol{w}^*) \in (0, \pi)$, then*

$$
\mathbb{E}_{\mathbf{Z}}\left[\boldsymbol{g}_{\mathrm{crelu}}(\boldsymbol{v}, \boldsymbol{w}; \mathbf{Z})\right] = \frac{p(0, \boldsymbol{w})h(\boldsymbol{v}, \boldsymbol{v}^*)}{2} \frac{\boldsymbol{w}}{\|\boldsymbol{w}\|} - (\boldsymbol{v}^\top \boldsymbol{v}^*)\csc(\theta/2) \cdot q(\theta, \boldsymbol{w}) \frac{\frac{\boldsymbol{w}}{\|\boldsymbol{w}\|} + \boldsymbol{w}^*}{\left\|\frac{\boldsymbol{w}}{\|\boldsymbol{w}\|} + \boldsymbol{w}^*\right\|}
$$
$$
- (\boldsymbol{v}^\top \boldsymbol{v}^*)\left(p(\theta, \boldsymbol{w}) - \cot(\theta/2) \cdot q(\theta, \boldsymbol{w})\right) \frac{\boldsymbol{w}}{\|\boldsymbol{w}\|}, \tag{26}
$$

*where $h(\boldsymbol{v}, \boldsymbol{v}^*) := \|\boldsymbol{v}\|^2 + (\boldsymbol{1}_m^\top \boldsymbol{v})^2 - (\boldsymbol{1}_m^\top \boldsymbol{v})(\boldsymbol{1}_m^\top \boldsymbol{v}^*) + \boldsymbol{v}^\top \boldsymbol{v}^*$ same as in Lemma 5, and*

$$
p(\theta, \boldsymbol{w}) := \frac{1}{2\pi}\int_{-\frac{\pi}{2}+\theta}^{\frac{\pi}{2}} \cos(\phi)\xi\left(\frac{\sec(\phi)}{\|\boldsymbol{w}\|}\right) \mathrm{d}\phi, \quad q(\theta, \boldsymbol{w}) := \frac{1}{2\pi}\int_{-\frac{\pi}{2}+\theta}^{\frac{\pi}{2}} \sin(\phi)\xi\left(\frac{\sec(\phi)}{\|\boldsymbol{w}\|}\right) \mathrm{d}\phi
$$

*with $\xi(x) := \int_0^x r^2 \exp(-\frac{r^2}{2})\mathrm{d}r$. The inner product between the expected coarse and true gradients w.r.t. $\boldsymbol{w}$*

$$
\left\langle \mathbb{E}_{\mathbf{Z}}\left[\boldsymbol{g}_{\mathrm{crelu}}(\boldsymbol{v}, \boldsymbol{w}; \mathbf{Z})\right], \frac{\partial f}{\partial \boldsymbol{w}}(\boldsymbol{v}, \boldsymbol{w})\right\rangle = \frac{q(\theta, \boldsymbol{w})}{2\pi\|\boldsymbol{w}\|}(\boldsymbol{v}^\top \boldsymbol{v}^*)^2 \geq 0.
$$

*Moreover, if further $\|\boldsymbol{v}\| \leq C_{\boldsymbol{v}}$ and $\|\boldsymbol{w}\| \geq c_{\boldsymbol{w}}$, there exists a constant $A_{\mathrm{crelu}} > 0$ depending on $C_{\boldsymbol{v}}$ and $c_{\boldsymbol{w}}$, such that*

$$
\left\|\mathbb{E}_{\mathbf{Z}}\left[\boldsymbol{g}_{\mathrm{crelu}}(\boldsymbol{v}, \boldsymbol{w}; \mathbf{Z})\right]\right\|^2 \leq A_{\mathrm{crelu}}\left(\left\|\frac{\partial f}{\partial \boldsymbol{v}}(\boldsymbol{v}, \boldsymbol{w})\right\|^2 + \left\langle \mathbb{E}_{\mathbf{Z}}\left[\boldsymbol{g}_{\mathrm{crelu}}(\boldsymbol{v}, \boldsymbol{w}; \mathbf{Z})\right], \frac{\partial f}{\partial \boldsymbol{w}}(\boldsymbol{v}, \boldsymbol{w})\right\rangle\right).
$$

**Proof of Lemma 7.** Denote $\theta := \theta(\boldsymbol{w}, \boldsymbol{w}^*)$. We first compute $\mathbb{E}_{\mathbf{Z}}\left[\boldsymbol{g}_{\mathrm{crelu}}(\boldsymbol{v}, \boldsymbol{w}; \mathbf{Z})\right]$. By (5),

$$
\boldsymbol{g}_\mu(\boldsymbol{v}, \boldsymbol{w}; \mathbf{Z}) = \mathbf{Z}^\top\left(\mu'(\mathbf{Z}\boldsymbol{w}) \odot \boldsymbol{v}\right)\left(\boldsymbol{v}^\top \sigma(\mathbf{Z}\boldsymbol{w}) - (\boldsymbol{v}^*)^\top \sigma(\mathbf{Z}\boldsymbol{w}^*)\right).
$$

Since $\mu' = \mathbb{1}_{\{0 < x < 1\}}$ and $\sigma = \mathbb{1}_{\{x > 0\}}$, we have

$$
\mathbb{E}_{\mathbf{Z}}\left[\boldsymbol{g}_{\mathrm{crelu}}(\boldsymbol{v}, \boldsymbol{w}; \mathbf{Z})\right] = \mathbb{E}_{\mathbf{Z}}\left[\left(\sum_{i=1}^m v_i \mu'(\mathbf{Z}_i^\top \boldsymbol{w}) - \sum_{i=1}^m v_i^* \mu'(\mathbf{Z}_i^\top \boldsymbol{w}^*)\right)\left(\sum_{i=1}^m \mathbf{Z}_i v_i \sigma(\mathbf{Z}_i^\top \boldsymbol{w})\right)\right]
$$
$$
= \mathbb{E}_{\mathbf{Z}}\left[\left(\sum_{i=1}^m v_i \mathbb{1}_{\{0 < \mathbf{Z}_i^\top \boldsymbol{w} < 1\}} - \sum_{i=1}^m v_i^* \mathbb{1}_{\{0 < \mathbf{Z}_i^\top \boldsymbol{w}^* < 1\}}\right)\left(\sum_{i=1}^m \mathbb{1}_{\{\mathbf{Z}_i^\top \boldsymbol{w} > 0\}} v_i \mathbf{Z}_i\right)\right]
$$
$$
= \frac{p(0, \boldsymbol{w})h(\boldsymbol{v}, \boldsymbol{v}^*)}{2} \frac{\boldsymbol{w}}{\|\boldsymbol{w}\|} - (\boldsymbol{v}^\top \boldsymbol{v}^*)\csc(\theta/2) \cdot q(\theta, \boldsymbol{w}) \frac{\frac{\boldsymbol{w}}{\|\boldsymbol{w}\|} + \boldsymbol{w}^*}{\left\|\frac{\boldsymbol{w}}{\|\boldsymbol{w}\|} + \boldsymbol{w}^*\right\|}
$$
$$
- (\boldsymbol{v}^\top \boldsymbol{v}^*)\left(p(\theta, \|\boldsymbol{w}\|) - \cot(\theta/2) \cdot q(\theta, \boldsymbol{w})\right) \frac{\boldsymbol{w}}{\|\boldsymbol{w}\|}.
$$

In the last equality above, we called Lemma 12.

Notice that $\left(\boldsymbol{I}_n - \frac{\boldsymbol{w}\boldsymbol{w}^\top}{\|\boldsymbol{w}\|^2}\right)\boldsymbol{w} = \boldsymbol{0}_n$ and $\|\boldsymbol{w}^*\| = 1$. If $\theta(\boldsymbol{w}, \boldsymbol{w}_*) \neq 0, \pi$, then the inner product between $\mathbb{E}_{\mathbf{Z}}\left[\boldsymbol{g}_{\mathrm{crelu}}(\boldsymbol{v}, \boldsymbol{w}; \mathbf{Z})\right]$ and $\frac{\partial f}{\partial \boldsymbol{w}}(\boldsymbol{v}, \boldsymbol{w})$ is given by

$$
\left\langle \mathbb{E}_{\mathbf{Z}}\left[\boldsymbol{g}_{\mathrm{crelu}}(\boldsymbol{v}, \boldsymbol{w}; \mathbf{Z})\right], \frac{\partial f}{\partial \boldsymbol{w}}(\boldsymbol{v}, \boldsymbol{w})\right\rangle
$$
$$
= \csc\left(\frac{\theta}{2}\right)\frac{q(\theta, \boldsymbol{w})}{2\pi}(\boldsymbol{v}^\top \boldsymbol{v}^*)^2 \left\langle \frac{1}{\|\boldsymbol{w}\|}\frac{\left(\boldsymbol{I}_n - \frac{\boldsymbol{w}\boldsymbol{w}^\top}{\|\boldsymbol{w}\|^2}\right)\boldsymbol{w}^*}{\left\|\left(\boldsymbol{I}_n - \frac{\boldsymbol{w}\boldsymbol{w}^\top}{\|\boldsymbol{w}\|^2}\right)\boldsymbol{w}^*\right\|}, \frac{\boldsymbol{w}^*}{\left\|\frac{\boldsymbol{w}}{\|\boldsymbol{w}\|} + \boldsymbol{w}^*\right\|}\right\rangle
$$
$$
= \frac{q(\theta, \boldsymbol{w})}{2\pi\|\boldsymbol{w}\|}(\boldsymbol{v}^\top \boldsymbol{v}^*)^2 \geq 0.
$$

In the last line, $q(\theta, \boldsymbol{w}) \geq 0$ because $\sin(\phi)\xi\left(\frac{\sec(\phi)}{\|\boldsymbol{w}\|}\right)$ is odd in $\phi$ and positive for $\phi \in (0, \frac{\pi}{2}]$.

Next, we bound $\left\|\mathbb{E}_{\mathbf{Z}}\left[\boldsymbol{g}_{\mathrm{crelu}}(\boldsymbol{v}, \boldsymbol{w}; \mathbf{Z})\right]\right\|^2$. Since (23) gives

$$h(\boldsymbol{v}, \boldsymbol{v}^*) = 4\boldsymbol{v}^\top \frac{\partial f}{\partial \boldsymbol{v}}(\boldsymbol{v}, \boldsymbol{w}) + 2\left(1 - \frac{\theta}{\pi}\right)\boldsymbol{v}^\top \boldsymbol{v}^*,$$

where according to Lemma 1,

$$\frac{\partial f}{\partial \boldsymbol{v}}(\boldsymbol{v}, \boldsymbol{w}) = \frac{1}{4}\left(\boldsymbol{I}_m + \mathbf{1}_m \mathbf{1}_m^\top\right)\boldsymbol{v} - \frac{1}{4}\left(\left(1 - \frac{2\theta}{\pi}\right)\boldsymbol{I}_m + \mathbf{1}_m \mathbf{1}_m^\top\right)\boldsymbol{v}^*.$$

We rewrite the coarse partial gradient w.r.t. $\boldsymbol{w}$ in (26) as

$$\mathbb{E}_{\mathbf{Z}}\left[\boldsymbol{g}_{\mathrm{crelu}}(\boldsymbol{v}, \boldsymbol{w}; \mathbf{Z})\right] = p(0, \boldsymbol{w})\left(2\boldsymbol{v}^\top \frac{\partial f}{\partial \boldsymbol{v}}(\boldsymbol{v}, \boldsymbol{w}) + \left(1 - \frac{\theta}{\pi}\right)\boldsymbol{v}^\top \boldsymbol{v}^*\right)\frac{\boldsymbol{w}}{\|\boldsymbol{w}\|}$$

$$- (\boldsymbol{v}^\top \boldsymbol{v}^*)\csc(\theta/2) \cdot q(\theta, \boldsymbol{w})\frac{\frac{\boldsymbol{w}}{\|\boldsymbol{w}\|} + \boldsymbol{w}^*}{\left\|\frac{\boldsymbol{w}}{\|\boldsymbol{w}\|} + \boldsymbol{w}^*\right\|}$$

$$- (\boldsymbol{v}^\top \boldsymbol{v}^*)\left(p(\theta, \boldsymbol{w}) - \cot(\theta/2) \cdot q(\theta, \boldsymbol{w})\right)\frac{\boldsymbol{w}}{\|\boldsymbol{w}\|}.$$

$$= 2p(0, \boldsymbol{w})\boldsymbol{v}^\top \frac{\partial f}{\partial \boldsymbol{v}}(\boldsymbol{v}, \boldsymbol{w})\frac{\boldsymbol{w}}{\|\boldsymbol{w}\|}$$

$$+ (\boldsymbol{v}^\top \boldsymbol{v}^*)\left(\left(1 - \frac{\theta}{\pi}\right)p(0, \boldsymbol{w}) - p(\theta, \boldsymbol{w})\right)\frac{\boldsymbol{w}}{\|\boldsymbol{w}\|}$$

$$+ (\boldsymbol{v}^\top \boldsymbol{v}^*)\csc(\theta/2) \cdot q(\theta, \boldsymbol{w})\left(\frac{\boldsymbol{w}}{\|\boldsymbol{w}\|} - \frac{\frac{\boldsymbol{w}}{\|\boldsymbol{w}\|} + \boldsymbol{w}^*}{\left\|\frac{\boldsymbol{w}}{\|\boldsymbol{w}\|} + \boldsymbol{w}^*\right\|}\right)$$

$$- (\boldsymbol{v}^\top \boldsymbol{v}^*)(\csc(\theta/2) - \cot(\theta/2))q(\theta, \boldsymbol{w})\frac{\boldsymbol{w}}{\|\boldsymbol{w}\|}$$

To prove the last claim, we notice that in the above equality,

$$\left\|\boldsymbol{v}^\top \frac{\partial f}{\partial \boldsymbol{v}}(\boldsymbol{v}, \boldsymbol{w})\frac{\boldsymbol{w}}{\|\boldsymbol{w}\|}\right\|^2 \leq C_{\boldsymbol{v}}^2 \left\|\frac{\partial f}{\partial \boldsymbol{v}}(\boldsymbol{v}, \boldsymbol{w})\right\|^2, \tag{27}$$

$$\left\|(\csc(\theta/2) - \cot(\theta/2))q(\theta, \boldsymbol{w})\frac{\boldsymbol{w}}{\|\boldsymbol{w}\|}\right\|^2 = (\csc(\theta/2) - \cot(\theta/2))^2 q(\theta, \boldsymbol{w})^2 \leq q(\theta, \boldsymbol{w})^2, \tag{28}$$

$$\left\|\csc(\theta/2) \cdot q(\theta, \boldsymbol{w})\left(\frac{\boldsymbol{w}}{\|\boldsymbol{w}\|} - \frac{\frac{\boldsymbol{w}}{\|\boldsymbol{w}\|} + \boldsymbol{w}^*}{\left\|\frac{\boldsymbol{w}}{\|\boldsymbol{w}\|} + \boldsymbol{w}^*\right\|}\right)\right\|^2 \leq \left(\frac{\theta/2}{\sin(\theta/2)}\right)^2 q(\theta, \boldsymbol{w})^2 \leq \frac{\pi^2}{4}q(\theta, \boldsymbol{w})^2. \tag{29}$$

Now, what is left is to bound $\left(\left(1 - \frac{\theta}{\pi}\right)p(0, \boldsymbol{w}) - p(\theta, \boldsymbol{w})\right)^2$, using a multiple of $q(\theta, \boldsymbol{w})$. Recall that

$$p(\theta, \boldsymbol{w}) = \frac{1}{2\pi}\int_{-\frac{\pi}{2}+\theta}^{\frac{\pi}{2}}\cos(\phi)\xi\left(\frac{\sec(\phi)}{\|\boldsymbol{w}\|}\right)\mathrm{d}\phi, \quad q(\theta, \boldsymbol{w}) = \frac{1}{2\pi}\int_{-\frac{\pi}{2}+\theta}^{\frac{\pi}{2}}\sin(\phi)\xi\left(\frac{\sec(\phi)}{\|\boldsymbol{w}\|}\right)\mathrm{d}\phi.$$

We first show that both $\left(\left(1 - \frac{\theta}{\pi}\right)p(0, \boldsymbol{w}) - p(\theta, \boldsymbol{w})\right)^2$ and $q(\theta, \boldsymbol{w})$ are symmetric with respect to $\theta = \frac{\pi}{2}$ on $[0, \pi]$. This is because

$$q(\pi - \theta, \boldsymbol{w}) = \frac{1}{2\pi}\int_{\frac{\pi}{2}-\theta}^{\frac{\pi}{2}}\sin(\phi)\xi\left(\frac{\sec(\phi)}{\|\boldsymbol{w}\|}\right)\mathrm{d}\phi$$

$$= q(\theta, \boldsymbol{w}) + \frac{1}{2\pi}\int_{\frac{\pi}{2}-\theta}^{-\frac{\pi}{2}+\theta}\sin(\phi)\xi\left(\frac{\sec(\phi)}{\|\boldsymbol{w}\|}\right)\mathrm{d}\phi = q(\theta, \boldsymbol{w}).$$

and

$$\left(1 - \frac{\pi - \theta}{\pi}\right) p(0, \boldsymbol{w}) - p(\pi - \theta, \boldsymbol{w})$$

$$= \frac{\theta}{\pi} \frac{1}{2\pi} \int_{-\frac{\pi}{2}}^{\frac{\pi}{2}} \cos(\phi)\xi\left(\frac{\sec(\phi)}{\|\boldsymbol{w}\|}\right) \mathrm{d}\phi - \frac{1}{2\pi} \int_{\frac{\pi}{2}-\theta}^{\frac{\pi}{2}} \cos(\phi)\xi\left(\frac{\sec(\phi)}{\|\boldsymbol{w}\|}\right) \mathrm{d}\phi$$

$$= \left(\frac{\theta}{\pi} - 1\right) \frac{1}{2\pi} \int_{-\frac{\pi}{2}}^{\frac{\pi}{2}} \cos(\phi)\xi\left(\frac{\sec(\phi)}{\|\boldsymbol{w}\|}\right) \mathrm{d}\phi + \frac{1}{2\pi} \int_{-\frac{\pi}{2}}^{\frac{\pi}{2}-\theta} \cos(\phi)\xi\left(\frac{\sec(\phi)}{\|\boldsymbol{w}\|}\right) \mathrm{d}\phi$$

$$= -\left(\left(1 - \frac{\theta}{\pi}\right) p(0, \boldsymbol{w}) - p(\theta, \boldsymbol{w})\right).$$

Therefore, it suffices to consider $\theta \in [\frac{\pi}{2}, \pi]$ only. Then calling Lemma 13 for $\theta \in [\frac{\pi}{2}, \pi]$, we have $p(\theta, \boldsymbol{w}) \leq q(\theta, \boldsymbol{w})$ and $\left(1 - \frac{\theta}{\pi}\right) p(0, \boldsymbol{w}) \leq q(\theta, \boldsymbol{w})$. Therefore,

$$\left(p(\theta, \boldsymbol{w}) - \left(1 - \frac{\theta}{\pi}\right) p(0, \boldsymbol{w})\right)^2 \leq q(\theta, \boldsymbol{w})^2.$$

Combining the above estimate together with (27), (28) and (29), and using Cauchy-Schwarz inequality, we have

$$\|\mathbb{E}_{\mathbf{Z}}[g_{\mathrm{crelu}}(\boldsymbol{v}, \boldsymbol{w}; \mathbf{Z})]\|^2 \leq 4\left(4p(0, \boldsymbol{w})C_{\boldsymbol{v}}^2 \left\|\frac{\partial f}{\partial \boldsymbol{v}}(\boldsymbol{v}, \boldsymbol{w})\right\|^2 + \left(2 + \frac{\pi^2}{4}\right) q(\theta, \boldsymbol{w})^2 (\boldsymbol{v}^\top \boldsymbol{v}^*)^2\right),$$

where $p(0, \boldsymbol{w})$ and $q(\theta, \boldsymbol{w})$ are uniformly bounded. This completes the proof.

$\square$

**Lemma 8.** *When Algorithm 1 converges, $\mathbb{E}_{\mathbf{Z}}\left[\frac{\partial \ell}{\partial \boldsymbol{v}}(\boldsymbol{v}, \boldsymbol{w}; \mathbf{Z})\right]$ and $\mathbb{E}_{\mathbf{Z}}\left[g_{\mathrm{crelu}}(\boldsymbol{v}, \boldsymbol{w}; \mathbf{Z})\right]$ vanish simultaneously, which only occurs at the*

1. *Saddle points where (8) is satisfied according to Proposition 1.*

2. *Minimizers of (2) where $\boldsymbol{v} = \boldsymbol{v}^*$, $\theta(\boldsymbol{w}, \boldsymbol{w}^*) = 0$, or $\boldsymbol{v} = (\boldsymbol{I}_m + \mathbf{1}_m \mathbf{1}_m^\top)^{-1}(\mathbf{1}_m \mathbf{1}_m^\top - \boldsymbol{I}_m)\boldsymbol{v}^*$, $\theta(\boldsymbol{w}, \boldsymbol{w}^*) = \pi$.*

**Proof of Lemma 8.** The proof of Lemma 8 is similar to that of Lemma 6, and we omit it here. The core part is that $q(\theta, \boldsymbol{w})$ defined in Lemma 12 is non-negative and equals 0 only at $\theta = 0, \pi$, as well as $p(0, \boldsymbol{w}) \geq p(\theta, \boldsymbol{w}) \geq p(\pi, \boldsymbol{w}) = 0$. $\square$

**Lemma 9.** *Let $\mu(x) = x$ in (5). Then the expected coarse partial gradient w.r.t. $\boldsymbol{w}$ is*

$$\mathbb{E}_{\mathbf{Z}}\left[g_{\mathrm{id}}(\boldsymbol{v}, \boldsymbol{w}; \mathbf{Z})\right] = \frac{1}{\sqrt{2\pi}}\left(\|\boldsymbol{v}\|^2 \frac{\boldsymbol{w}}{\|\boldsymbol{w}\|} - (\boldsymbol{v}^\top \boldsymbol{v}^*)\boldsymbol{w}^*\right).$$

*If $\theta(\boldsymbol{w}, \boldsymbol{w}^*) = \pi$ and $\boldsymbol{v} = (\boldsymbol{I}_m + \mathbf{1}_m \mathbf{1}_m^\top)^{-1}(\mathbf{1}_m \mathbf{1}_m^\top - \boldsymbol{I}_m)\boldsymbol{v}^*$,*

$$\left\|\mathbb{E}_{\mathbf{Z}}\left[g_{\mathrm{id}}(\boldsymbol{v}, \boldsymbol{w}; \mathbf{Z})\right]\right\| = \frac{2(m-1)}{\sqrt{2\pi}(m+1)^2}(\mathbf{1}_m^\top \boldsymbol{v}^*)^2 \geq 0,$$

*i.e., $\mathbb{E}_{\mathbf{Z}}\left[g_{\mathrm{id}}(\boldsymbol{v}, \boldsymbol{w}; \mathbf{Z})\right]$ does not vanish at the local minimizers if $\mathbf{1}_m^\top \boldsymbol{v}^* \neq 0$ and $m > 1$,.*

**Proof of Lemma 9.** By (5),

$$\boldsymbol{g}_\mu(\boldsymbol{v}, \boldsymbol{w}; \mathbf{Z}) = \mathbf{Z}^\top\left(\mu'(\mathbf{Z}\boldsymbol{w}) \odot \boldsymbol{v}\right)\left(\boldsymbol{v}^\top \sigma(\mathbf{Z}\boldsymbol{w}) - (\boldsymbol{v}^*)^\top \sigma(\mathbf{Z}\boldsymbol{w}^*)\right).$$

Using the facts that $\mu' = 1$ and $\sigma = 1_{\{x>0\}}$, we have

$$
\begin{aligned}
\mathbb{E}_{\mathbf{Z}}\left[\boldsymbol{g}_{\mathrm{id}}(\boldsymbol{v}, \boldsymbol{w}; \mathbf{Z})\right] &= \mathbb{E}_{\mathbf{Z}}\left[\left(\sum_{i=1}^{m} v_i 1_{\{\mathbf{Z}_i^\top \boldsymbol{w}>0\}} - \sum_{i=1}^{m} v_i^* 1_{\{\mathbf{Z}_i^\top \boldsymbol{w}^*>0\}}\right)\left(\sum_{i=1}^{m} v_i \mathbf{Z}_i\right)\right] \\
&= \sum_{i=1}^{m}\sum_{j=1}^{m} v_i v_j \mathbb{E}\left[\mathbf{Z}_i 1_{\{\mathbf{Z}_j^\top \boldsymbol{w}>0\}}\right] - \sum_{i=1}^{m}\sum_{j=1}^{m} v_i^* v_j \mathbb{E}\left[\mathbf{Z}_i 1_{\{\mathbf{Z}_j^\top \boldsymbol{w}^*>0\}}\right] \\
&= \frac{1}{\sqrt{2\pi}}\left(\|\boldsymbol{v}\|^2 \frac{\boldsymbol{w}}{\|\boldsymbol{w}\|} - (\boldsymbol{v}^\top \boldsymbol{v}^*)\boldsymbol{w}^*\right).
\end{aligned}
$$

In the last equality above, we called the third identity in Lemma 11. If $\theta(\boldsymbol{w}, \boldsymbol{w}^*) = \pi$ and $\boldsymbol{v} = \left(\boldsymbol{I}_m + \mathbf{1}_m \mathbf{1}_m^\top\right)^{-1}(\mathbf{1}_m \mathbf{1}_m^\top - \boldsymbol{I}_m)\boldsymbol{v}^*$, then

$$
\begin{aligned}
\left\|\mathbb{E}_{\mathbf{Z}}\left[\boldsymbol{g}_{\mathrm{id}}(\boldsymbol{v}, \boldsymbol{w}; \mathbf{Z})\right]\right\| &= \frac{1}{\sqrt{2\pi}}|\boldsymbol{v}^\top(\boldsymbol{v} + \boldsymbol{v}^*)| \\
&= \frac{1}{\sqrt{2\pi}}\left|(\boldsymbol{v}^*)^\top (\mathbf{1}_m \mathbf{1}_m^\top - \boldsymbol{I}_m)(\boldsymbol{I}_m + \mathbf{1}_m \mathbf{1}_m^\top)^{-1}\left(\left(\boldsymbol{I}_m + \mathbf{1}_m \mathbf{1}_m^\top\right)^{-1}(\mathbf{1}_m \mathbf{1}_m^\top - \boldsymbol{I}_m) + \boldsymbol{I}_m\right)\boldsymbol{v}^*\right| \\
&= \frac{2}{\sqrt{2\pi}}\left|(\boldsymbol{v}^*)^\top (\mathbf{1}_m \mathbf{1}_m^\top - \boldsymbol{I}_m)(\boldsymbol{I}_m + \mathbf{1}_m \mathbf{1}_m^\top)^{-1}(\boldsymbol{I}_m + \mathbf{1}_m \mathbf{1}_m^\top)^{-1}\mathbf{1}_m(\mathbf{1}_m^\top \boldsymbol{v}^*)\right| \\
&= \frac{2}{\sqrt{2\pi}(m+1)^2}\left|(\boldsymbol{v}^*)^\top (\mathbf{1}_m \mathbf{1}_m^\top - \boldsymbol{I}_m)\mathbf{1}_m(\mathbf{1}_m^\top \boldsymbol{v}^*)\right| \\
&= \frac{2(m-1)}{\sqrt{2\pi}(m+1)^2}(\mathbf{1}_m^\top \boldsymbol{v}^*)^2.
\end{aligned}
$$

In the third equality, we used the identity $(\boldsymbol{I}_m + \mathbf{1}_m \mathbf{1}_m^\top)\mathbf{1}_m = (m+1)\mathbf{1}_m$ twice. $\qquad\square$

**Lemma 10.** *If $\boldsymbol{w} \neq \mathbf{0}_n$ and $\theta(\boldsymbol{w}, \boldsymbol{w}^*) \in (0, \pi)$, then the inner product between the expected coarse and true gradients w.r.t. $\boldsymbol{w}$ is*

$$
\left\langle \mathbb{E}_{\mathbf{Z}}\left[\boldsymbol{g}_{\mathrm{id}}(\boldsymbol{v}, \boldsymbol{w}; \mathbf{Z})\right], \frac{\partial f}{\partial \boldsymbol{w}}(\boldsymbol{v}, \boldsymbol{w})\right\rangle = \frac{\sin\left(\theta(\boldsymbol{w}, \boldsymbol{w}^*)\right)}{(\sqrt{2\pi})^3\|\boldsymbol{w}\|}(\boldsymbol{v}^\top \boldsymbol{v}^*)^2 \geq 0.
$$

*When $\theta(\boldsymbol{w}, \boldsymbol{w}^*) \to \pi$, $\boldsymbol{v} \to \left(\boldsymbol{I}_m + \mathbf{1}_m \mathbf{1}_m^\top\right)^{-1}(\mathbf{1}_m \mathbf{1}_m^\top - \boldsymbol{I}_m)\boldsymbol{v}^*$, if $\mathbf{1}_m^\top \boldsymbol{v}^* \neq 0$ and $m > 1$, we have*

$$
\frac{\left\|\mathbb{E}_{\mathbf{Z}}\left[\boldsymbol{g}_{\mathrm{id}}(\boldsymbol{v}, \boldsymbol{w}; \mathbf{Z})\right]\right\|^2}{\left\|\frac{\partial f}{\partial \boldsymbol{v}}(\boldsymbol{v}, \boldsymbol{w})\right\|^2 + \left\langle \mathbb{E}_{\mathbf{Z}}\left[\boldsymbol{g}_{\mathrm{id}}(\boldsymbol{v}, \boldsymbol{w}; \mathbf{Z})\right], \frac{\partial f}{\partial \boldsymbol{w}}(\boldsymbol{v}, \boldsymbol{w})\right\rangle} \to +\infty.
$$

**Proof of Lemma 10.** By Lemmas 2 and 4, we have

$$
\frac{\partial f}{\partial \boldsymbol{w}}(\boldsymbol{v}, \boldsymbol{w}) = -\frac{\boldsymbol{v}^\top \boldsymbol{v}^*}{2\pi\|\boldsymbol{w}\|}\frac{\left(\boldsymbol{I}_n - \frac{\boldsymbol{w}\boldsymbol{w}^\top}{\|\boldsymbol{w}\|^2}\right)\boldsymbol{w}^*}{\left\|\left(\boldsymbol{I}_n - \frac{\boldsymbol{w}\boldsymbol{w}^\top}{\|\boldsymbol{w}\|^2}\right)\boldsymbol{w}^*\right\|}
$$

and

$$
\mathbb{E}_{\mathbf{Z}}\left[\boldsymbol{g}_{\mathrm{id}}(\boldsymbol{v}, \boldsymbol{w}; \mathbf{Z})\right] = \frac{1}{\sqrt{2\pi}}\left(\|\boldsymbol{v}\|^2 \frac{\boldsymbol{w}}{\|\boldsymbol{w}\|} - (\boldsymbol{v}^\top \boldsymbol{v}^*)\boldsymbol{w}^*\right).
$$

Since $\left(\boldsymbol{I}_n - \frac{\boldsymbol{w}\boldsymbol{w}^\top}{\|\boldsymbol{w}\|^2}\right)\boldsymbol{w} = \mathbf{0}_n$ and $\|\boldsymbol{w}^*\| = 1$, if $\theta(\boldsymbol{w}, \boldsymbol{w}_*) \neq 0, \pi$, then we have

$$
\begin{aligned}
\left\langle \mathbb{E}_{\mathbf{Z}}\left[\boldsymbol{g}_{\mathrm{id}}(\boldsymbol{v}, \boldsymbol{w}; \mathbf{Z})\right], \frac{\partial f}{\partial \boldsymbol{w}}(\boldsymbol{v}, \boldsymbol{w})\right\rangle &= \frac{(\boldsymbol{v}^\top \boldsymbol{v}^*)^2}{(\sqrt{2\pi})^3\|\boldsymbol{w}\|}\frac{(\boldsymbol{w}^*)^\top\left(\boldsymbol{I}_n - \frac{\boldsymbol{w}\boldsymbol{w}^\top}{\|\boldsymbol{w}\|^2}\right)\boldsymbol{w}^*}{\left\|\left(\boldsymbol{I}_n - \frac{\boldsymbol{w}\boldsymbol{w}^\top}{\|\boldsymbol{w}\|^2}\right)\boldsymbol{w}^*\right\|} \\
&= \frac{(\boldsymbol{v}^\top \boldsymbol{v}^*)^2}{(\sqrt{2\pi})^3\|\boldsymbol{w}\|}\frac{1 - \frac{(\boldsymbol{w}^\top \boldsymbol{w}^*)^2}{\|\boldsymbol{w}\|^2}}{\sqrt{1 - \frac{(\boldsymbol{w}^\top \boldsymbol{w}^*)^2}{\|\boldsymbol{w}\|^2}}} = \frac{(\boldsymbol{v}^\top \boldsymbol{v}^*)^2}{(\sqrt{2\pi})^3\|\boldsymbol{w}\|}\sqrt{1 - \frac{(\boldsymbol{w}^\top \boldsymbol{w}^*)^2}{\|\boldsymbol{w}\|^2}} \\
&= \frac{(\boldsymbol{v}^\top \boldsymbol{v}^*)^2}{(\sqrt{2\pi})^3\|\boldsymbol{w}\|}\sin(\theta(\boldsymbol{w}, \boldsymbol{w}^*)).
\end{aligned}
$$

When $\theta(\boldsymbol{w}, \boldsymbol{w}^*) \rightarrow \pi$, $\boldsymbol{v} \rightarrow (\boldsymbol{I}_m + \boldsymbol{1}_m\boldsymbol{1}_m^\top)^{-1}(\boldsymbol{1}_m\boldsymbol{1}_m^\top - \boldsymbol{I}_m)\boldsymbol{v}^*$, both $\left\|\frac{\partial f}{\partial \boldsymbol{v}}(\boldsymbol{v}, \boldsymbol{w})\right\|$ and $\left\langle \mathbb{E}_{\mathbf{Z}}\left[\boldsymbol{g}_{\mathrm{id}}(\boldsymbol{v}, \boldsymbol{w}; \mathbf{Z})\right], \frac{\partial f}{\partial \boldsymbol{w}}(\boldsymbol{v}, \boldsymbol{w})\right\rangle$ converge to 0. But if $\boldsymbol{1}_m^\top\boldsymbol{v}^* \neq 0$ and $m > 1$, $\left\|\mathbb{E}_{\mathbf{Z}}\left[\boldsymbol{g}_{\mathrm{id}}(\boldsymbol{v}, \boldsymbol{w}; \mathbf{Z})\right]\right\| \rightarrow \frac{2(m-1)}{\sqrt{2\pi}(m+1)^2}(\boldsymbol{1}_m^\top\boldsymbol{v}^*)^2 > 0$, which completes the proof. $\qquad\square$

**Theorem 1.** *Let $\{(\boldsymbol{v}^t, \boldsymbol{w}^t)\}$ be the sequence generated by Algorithm 1 with ReLU $\mu(x) = \max\{x, 0\}$ or clipped ReLU $\mu(x) = \min\{\max\{x, 0\}, 1\}$. Suppose $\|\boldsymbol{w}^t\| \geq c_{\boldsymbol{w}}$ for all $t$ with some $c_{\boldsymbol{w}} > 0$. Then if the learning rate $\eta > 0$ is sufficiently small, for any initialization $(\boldsymbol{v}^0, \boldsymbol{w}^0)$, the objective sequence $\{f(\boldsymbol{v}^t, \boldsymbol{w}^t)\}$ is monotonically decreasing, and $\{(\boldsymbol{v}^t, \boldsymbol{w}^t)\}$ converges to a saddle point or a (local) minimizer of the population loss minimization (2). In addition, if $\boldsymbol{1}_m^\top\boldsymbol{v}^* \neq 0$ and $m > 1$, the descent and convergence properties do not hold for Algorithm 1 with the identity function $\mu(x) = x$ near the local minimizers satisfying $\theta(\boldsymbol{w}, \boldsymbol{w}^*) = \pi$ and $\boldsymbol{v} = (\boldsymbol{I}_m + \boldsymbol{1}_m\boldsymbol{1}_m^\top)^{-1}(\boldsymbol{1}_m\boldsymbol{1}_m^\top - \boldsymbol{I}_m)\boldsymbol{v}^*$.*

**Proof of Theorem 1**. We first prove the upper boundedness of $\{\boldsymbol{v}^t\}$. Due to the coerciveness of $f(\boldsymbol{v}, \boldsymbol{w})$ w.r.t $\boldsymbol{v}$, there exists $C_{\boldsymbol{v}} > 0$, such that $\|\boldsymbol{v}\| \leq C_{\boldsymbol{v}}$ for any $\boldsymbol{v} \in \{\boldsymbol{v} \in \mathbb{R}^m : f(\boldsymbol{v}, \boldsymbol{w}) \leq f(\boldsymbol{v}^0, \boldsymbol{w}^0)$ for some $\boldsymbol{w}\}$. In particular, $\|\boldsymbol{v}^0\| \leq C_{\boldsymbol{v}}$. Using induction, suppose we already have $f(\boldsymbol{v}^t, \boldsymbol{w}^t) \leq f(\boldsymbol{v}^0, \boldsymbol{w}^0)$ and $\|\boldsymbol{v}^t\| \leq C_{\boldsymbol{v}}$. If $\theta(\boldsymbol{w}^t, \boldsymbol{w}^*) = 0$ or $\pi$, then $\theta(\boldsymbol{w}^T, \boldsymbol{w}^*) = 0$ or $\pi$ for all $T \geq t$, and the original problem reduces to a quadratic program in terms of $\boldsymbol{v}$. So $\{\boldsymbol{v}^t\}$ will converge to $\boldsymbol{v}^*$ or $(\boldsymbol{I}_m + \boldsymbol{1}_m\boldsymbol{1}_m^\top)^{-1}(\boldsymbol{1}_m\boldsymbol{1}_m^\top - \boldsymbol{I}_m)\boldsymbol{v}^*$ by choosing a suitable step size $\eta$. In either case, we have $\left\|\mathbb{E}_{\mathbf{Z}}\left[\frac{\partial \ell}{\partial \boldsymbol{v}}(\boldsymbol{v}^t, \boldsymbol{w}^t; \mathbf{Z})\right]\right\|$ and $\left\|\mathbb{E}_{\mathbf{Z}}\left[\boldsymbol{g}_{\mathrm{relu}}(\boldsymbol{v}^t, \boldsymbol{w}^t; \mathbf{Z})\right]\right\|$ both converge to 0. Else if $\theta(\boldsymbol{w}^t, \boldsymbol{w}^*) \in (0, \pi)$, we define for any $a \in [0, 1]$ that

$$\boldsymbol{v}^t(a) := \boldsymbol{v}^t - a(\boldsymbol{v}^{t+1} - \boldsymbol{v}^t) = \boldsymbol{v}^t - a\eta\mathbb{E}_{\mathbf{Z}}\left[\frac{\partial \ell}{\partial \boldsymbol{v}}(\boldsymbol{v}^t, \boldsymbol{w}^t; \mathbf{Z})\right]$$

and

$$\boldsymbol{w}^t(a) := \boldsymbol{w}^t - a(\boldsymbol{w}^{t+1} - \boldsymbol{w}^t) = \boldsymbol{w}^t - a\eta\mathbb{E}_{\mathbf{Z}}\left[\boldsymbol{g}_{\mathrm{relu}}(\boldsymbol{v}^t, \boldsymbol{w}^t; \mathbf{Z})\right],$$

which satisfy

$$\boldsymbol{v}^t(0) = \boldsymbol{v}^t, \ \boldsymbol{v}^t(1) = \boldsymbol{v}^{t+1}, \ \boldsymbol{w}^t(0) = \boldsymbol{w}^t, \ \boldsymbol{w}^t(1) = \boldsymbol{w}^{t+1}.$$

Let us fix $0 < c \leq c_{\boldsymbol{w}}$ and $C \geq C_{\boldsymbol{v}}$. By the expressions of $\mathbb{E}_{\mathbf{Z}}\left[\frac{\partial \ell}{\partial \boldsymbol{v}}(\boldsymbol{v}^t, \boldsymbol{w}^t; \mathbf{Z})\right]$ and $\mathbb{E}_{\mathbf{Z}}\left[\boldsymbol{g}_{\mathrm{relu}}(\boldsymbol{v}^t, \boldsymbol{w}^t; \mathbf{Z})\right]$ given in Lemma 4, and since $\|\boldsymbol{w}^t\| = 1$, for sufficiently small $\tilde{\eta}$ depending on $C_0$ and $c_{\boldsymbol{w}}$, with $\eta \leq \tilde{\eta}$, it holds that $\|\boldsymbol{v}^t(a)\| \leq C$ and $\|\boldsymbol{w}^t(a)\| \geq c$ for all $a \in [0, 1]$. Possibly at some point $a_0$ where $\theta(\boldsymbol{w}^t(a_0), \boldsymbol{w}^*) = 0$ or $\pi$, the partial gradient $\frac{\partial f}{\partial \boldsymbol{w}}(\boldsymbol{v}^t(a_0), \boldsymbol{w}^t(a_0))$ does not exist. Otherwise, $\left\|\frac{\partial f}{\partial \boldsymbol{w}}(\boldsymbol{v}^t(a), \boldsymbol{w}^t(a))\right\|$ is uniformly bounded for all $a \in [0, 1]/\{a_0\}$, which makes it integrable over the interval $[0, 1]$. Then for some constants $L$ and $A_{\mathrm{relu}}$ depending on $C$ and $c$, we

have

$$f(\boldsymbol{v}^{t+1}, \boldsymbol{w}^{t+1}) = f(\boldsymbol{v}^t + (\boldsymbol{v}^{t+1} - \boldsymbol{v}^t), \boldsymbol{w}^t + (\boldsymbol{w}^{t+1} - \boldsymbol{w}^t))$$

$$= f(\boldsymbol{v}^t, \boldsymbol{w}^t) + \int_0^1 \left\langle \frac{\partial f}{\partial \boldsymbol{v}}(\boldsymbol{v}^t(a), \boldsymbol{w}^t(a)), \boldsymbol{v}^{t+1} - \boldsymbol{v}^t \right\rangle \mathrm{d}a$$

$$+ \int_0^1 \left\langle \frac{\partial f}{\partial \boldsymbol{w}}(\boldsymbol{v}^t(a), \boldsymbol{w}^t(a)), \boldsymbol{w}^{t+1} - \boldsymbol{w}^t \right\rangle \mathrm{d}a$$

$$= f(\boldsymbol{v}^t, \boldsymbol{w}^t) + \left\langle \frac{\partial f}{\partial \boldsymbol{v}}(\boldsymbol{v}^t, \boldsymbol{w}^t), \boldsymbol{v}^{t+1} - \boldsymbol{v}^t \right\rangle + \left\langle \frac{\partial f}{\partial \boldsymbol{w}}(\boldsymbol{v}^t, \boldsymbol{w}^t), \boldsymbol{w}^{t+1} - \boldsymbol{w}^t \right\rangle$$

$$+ \int_0^1 \left\langle \frac{\partial f}{\partial \boldsymbol{v}}(\boldsymbol{v}^t(a), \boldsymbol{w}^t(a)) - \frac{\partial f}{\partial \boldsymbol{v}}(\boldsymbol{v}^t, \boldsymbol{w}^t), \boldsymbol{v}^{t+1} - \boldsymbol{v}^t \right\rangle \mathrm{d}a$$

$$+ \int_0^1 \left\langle \frac{\partial f}{\partial \boldsymbol{w}}(\boldsymbol{v}^t(a), \boldsymbol{w}^t(a)) - \frac{\partial f}{\partial \boldsymbol{w}}(\boldsymbol{v}^t, \boldsymbol{w}^t), \boldsymbol{w}^{t+1} - \boldsymbol{w}^t \right\rangle \mathrm{d}a$$

$$\leq f(\boldsymbol{v}^t, \boldsymbol{w}^t) - \left(\eta - \frac{L\eta^2}{2}\right) \left\| \frac{\partial f}{\partial \boldsymbol{v}}(\boldsymbol{v}^t, \boldsymbol{w}^t) \right\|^2$$

$$- \eta \left\langle \frac{\partial f}{\partial \boldsymbol{w}}(\boldsymbol{v}^t, \boldsymbol{w}^t), \mathbb{E}_{\mathbf{Z}}\Big[\boldsymbol{g}_{\mathrm{relu}}(\boldsymbol{v}^t, \boldsymbol{w}^t; \mathbf{Z})\Big] \right\rangle + \frac{L\eta^2}{2} \left\| \mathbb{E}_{\mathbf{Z}}\Big[\boldsymbol{g}_{\mathrm{relu}}(\boldsymbol{v}^t, \boldsymbol{w}^t; \mathbf{Z})\Big] \right\|^2$$

$$\leq f(\boldsymbol{v}^t, \boldsymbol{w}^t) - \left(\eta - (1 + A_{\mathrm{relu}})\frac{L\eta^2}{2}\right) \left\| \frac{\partial f}{\partial \boldsymbol{v}}(\boldsymbol{v}^t, \boldsymbol{w}^t) \right\|^2$$

$$- \left(\eta - \frac{A_{\mathrm{relu}}L\eta^2}{2}\right) \left\langle \frac{\partial f}{\partial \boldsymbol{w}}(\boldsymbol{v}^t, \boldsymbol{w}^t), \mathbb{E}_{\mathbf{Z}}\Big[\boldsymbol{g}_{\mathrm{relu}}(\boldsymbol{v}^t, \boldsymbol{w}^t; \mathbf{Z})\Big] \right\rangle. \tag{30}$$

The third equality is due to the fundamental theorem of calculus. In the first inequality, we called Lemma 3 for $(\boldsymbol{v}^t, \boldsymbol{w}^t)$ and $(\boldsymbol{v}^t(a), \boldsymbol{w}^t(a))$ with $a \in [0,1]/\{a_0\}$. In the last inequality, we used Lemma 5. So when $\eta < \eta_0 := \min\left\{\frac{2}{(1+A_{\mathrm{relu}})L}, \tilde{\eta}\right\}$, we have $f(\boldsymbol{v}^{t+1}, \boldsymbol{w}^{t+1}) \leq f(\boldsymbol{v}^t, \boldsymbol{w}^t) \leq f(\boldsymbol{v}^0, \boldsymbol{w}^0)$, and thus $\|\boldsymbol{v}^{t+1}\| \leq C_{\boldsymbol{v}}$.

Summing up the inequality (30) over $t$ from 0 to $\infty$ and using $f \geq 0$, we have

$$\sum_{t=0}^{\infty} \left(1 - (1 + A_{\mathrm{relu}})\frac{L\eta}{2}\right) \left\| \frac{\partial f}{\partial \boldsymbol{v}}(\boldsymbol{v}^t, \boldsymbol{w}^t) \right\|^2 + \left(1 - \frac{A_{\mathrm{relu}}L\eta}{2}\right) \left\langle \frac{\partial f}{\partial \boldsymbol{w}}(\boldsymbol{v}^t, \boldsymbol{w}^t), \mathbb{E}_{\mathbf{Z}}\Big[\boldsymbol{g}_{\mathrm{relu}}(\boldsymbol{v}^t, \boldsymbol{w}^t; \mathbf{Z})\Big] \right\rangle$$

$$\leq f(\boldsymbol{v}^0, \boldsymbol{w}^0)/\eta < \infty.$$

Hence,

$$\lim_{t \to \infty} \left\| \frac{\partial f}{\partial \boldsymbol{v}}(\boldsymbol{v}^t, \boldsymbol{w}^t) \right\| = 0$$

and

$$\lim_{t \to \infty} \left\langle \frac{\partial f}{\partial \boldsymbol{w}}(\boldsymbol{v}^t, \boldsymbol{w}^t), \mathbb{E}_{\mathbf{Z}}\Big[\boldsymbol{g}_{\mathrm{relu}}(\boldsymbol{v}^t, \boldsymbol{w}^t; \mathbf{Z})\Big] \right\rangle = 0.$$

Invoking Lemma 5 again, we further have

$$\lim_{t \to \infty} \left\| \mathbb{E}_{\mathbf{Z}}\Big[\boldsymbol{g}_{\mathrm{relu}}(\boldsymbol{v}^t, \boldsymbol{w}^t; \mathbf{Z})\Big] \right\| = 0.$$

Invoking Lemma 6, we have that coarse gradient descent with ReLU $\mu(x)$ (subsequentially) converges to a saddle point or a minimizer.

Using Lemmas 7, 8 and similar arguments, we can prove the convergence of coarse gradient descent with clipped ReLU STE.

The second claim follows from Lemmas 9 and 10. $\qquad\square$

## F. CONVERGENCE TO GLOBAL MINIMIZERS

We prove that if the initialization weights $(\boldsymbol{v}^0, \boldsymbol{w}^0)$ satisfy $(\boldsymbol{v}^0)^\top \boldsymbol{v}^* > 0$, $\theta(\boldsymbol{w}^0, \boldsymbol{w}^*) < \frac{\pi}{2}$ and $(\mathbf{1}_m^\top \boldsymbol{v}^*)(\mathbf{1}_m^\top \boldsymbol{v}^0) \leq (\mathbf{1}_m^\top \boldsymbol{v}^*)^2$, then we have convergence guarantee to global optima by using the vanilla or clipped ReLU STE.

**Theorem 2.** *Under the assumptions of Theorem 1, if further the initialization $(\boldsymbol{v}^0, \boldsymbol{w}^0)$ satisfies $(\boldsymbol{v}^0)^\top \boldsymbol{v}^* > 0$, $\theta(\boldsymbol{w}^0, \boldsymbol{w}^*) < \frac{\pi}{2}$ and $(\mathbf{1}_m^\top \boldsymbol{v}^*)(\mathbf{1}_m^\top \boldsymbol{v}^0) \le (\mathbf{1}_m^\top \boldsymbol{v}^*)^2$, then by using the vanila or clipped ReLU STE for sufficiently learning rate $\eta > 0$, we have $(\boldsymbol{v}^t)^\top \boldsymbol{v}^* > 0$ and $\theta(\boldsymbol{w}^t, \boldsymbol{w}^*) < \frac{\pi}{2}$ for all $t > 0$, and $\{(\boldsymbol{v}^t, \boldsymbol{w}^t)\}$ converges to a global minimizer.*

**Proof of Theorem 2.** Proof by induction. Suppose $(\boldsymbol{v}^t)^\top \boldsymbol{v}^* > 0$, $\theta(\boldsymbol{w}^t, \boldsymbol{w}^*) < \frac{\pi}{2}$ and $(\mathbf{1}_m^\top \boldsymbol{v}^*)(\mathbf{1}_m^\top \boldsymbol{v}^t) \le (\mathbf{1}_m^\top \boldsymbol{v}^*)^2$. Then for small enough $\eta$, we have

$$
\begin{aligned}
(\boldsymbol{v}^{t+1})^\top \boldsymbol{v}^* &= \left( \boldsymbol{v}^t - \frac{\eta}{4} \left( (\boldsymbol{I}_m + \mathbf{1}_m \mathbf{1}_m^\top) \boldsymbol{v}^t - \left( \left(1 - \frac{2}{\pi} \theta(\boldsymbol{w}^t, \boldsymbol{w}^*)\right) \boldsymbol{I}_m + \mathbf{1}_m \mathbf{1}_m^\top \right) \boldsymbol{v}^* \right) \right)^\top \boldsymbol{v}^* \\
&= \left(1 - \frac{\eta}{4}\right) (\boldsymbol{v}^t)^\top \boldsymbol{v}^* + \frac{\eta}{4} \left( (\mathbf{1}_m^\top \boldsymbol{v}^*)^2 - (\mathbf{1}_m^\top \boldsymbol{v}^*)(\mathbf{1}_m^\top \boldsymbol{v}^\top) \right) \\
&\quad + \frac{\eta}{4} \left(1 - \frac{2}{\pi} \theta(\boldsymbol{w}^t, \boldsymbol{w}^*)\right) \|\boldsymbol{v}^*\|^2 > 0.
\end{aligned}
$$

and

$$
\begin{aligned}
(\mathbf{1}_m^\top \boldsymbol{v}^{t+1})(\mathbf{1}_m^\top \boldsymbol{v}^*) &= \left(1 - \frac{(m+1)\eta}{4}\right) (\mathbf{1}_m^\top \boldsymbol{v}^t)(\mathbf{1}_m^\top \boldsymbol{v}^*) + \frac{\eta}{4} \left(m + 1 - \frac{2}{\pi} \theta(\boldsymbol{w}, \boldsymbol{w}^*)\right) (\mathbf{1}_m^\top \boldsymbol{v}^*)^2 \\
&\le \left(1 - \frac{\eta}{2\pi} \theta(\boldsymbol{w}, \boldsymbol{w}^*)\right) (\mathbf{1}_m^\top \boldsymbol{v}^*)^2 \le (\mathbf{1}_m^\top \boldsymbol{v}^*)^2.
\end{aligned}
$$

Moreover, by Lemmas 4, 5 and 7, both $\mathbb{E}_{\mathbf{Z}}[\boldsymbol{g}_{\mathrm{relu}}(\boldsymbol{v}, \boldsymbol{w}; \mathbf{Z})]$ and $\mathbb{E}_{\mathbf{Z}}[\boldsymbol{g}_{\mathrm{crelu}}(\boldsymbol{v}, \boldsymbol{w}; \mathbf{Z})]$ can be written in the form of $\alpha_1 \left( \boldsymbol{I}_m - \frac{\boldsymbol{w}\boldsymbol{w}^\top}{\|\boldsymbol{w}\|^2} \right) \boldsymbol{w}^* + \alpha_2 \boldsymbol{w}$, where $\alpha_1 \le 0$ and $\alpha_2$ is bounded by a constant depending on $C_{\boldsymbol{v}}$ and $c_{\boldsymbol{w}}$. Therefore,

$$
(\boldsymbol{w}^{t+1})^\top \boldsymbol{w}^* = (1 - \eta \alpha_2)(\boldsymbol{w}^t)^\top \boldsymbol{w}^* - \alpha_1 (\boldsymbol{w}^*)^\top \left( \boldsymbol{I}_m - \frac{\boldsymbol{w}\boldsymbol{w}^\top}{\|\boldsymbol{w}\|^2} \right) \boldsymbol{w}^* \ge (1 - \eta \alpha_2)(\boldsymbol{w}^t)^\top \boldsymbol{w}^* > 0,
$$

and thus $\theta(\boldsymbol{w}^{t+1}, \boldsymbol{w}^*) < \frac{\pi}{2}$. Finally, since $\{(\boldsymbol{v}^t, \boldsymbol{w}^t)\}$ converges, it can only converge to a global minimizer. $\qquad \square$

