# OpenReview forum: "Understanding Straight-Through Estimator in Training Activation Quantized Neural Nets"
_ICLR.cc/2019/Conference_

### Official Review · AnonReviewer1 · 2018-10-29
**Interesting analysis of STE used in activation bianrized networks but not well written.**

**Rating:** 6
**Confidence:** 3

**Review:**

This paper provides theoretical analysis for two kinds of straight-through estimation (STE) for activation bianrized neural networks. It is theoretically shown that the ReLU STE has better convergence properties the identity STE,  by studying the properties of the orientation and norm of the course gradients for STE.

While the paper presents many theoretical results which might be useful for the community, they are not organized very well.  It is a bit hard for readers to quickly find the most important theoretical results.  Moreover, some symbols are used without definition, e.g. g_{relu} is used before being defined in sec 3.1. The discussions for most theoretical results are very short or not organized well, making the whole paper hard to follow,  e.g., "the key observation ..." after Lemma 4 is actually not about the Lemma 4 above, but Lemma 5 in the next Lemma.  Another major concern is that activation quantization is usually used in combination with weight quantization.  It would be more useful if weight and activation quantizations can be analyzed together.

Clarity in the experiment part can also be further improved. From Table 1, the clipped ReLU STE has the best performance, however, there is no theoretical analysis for it. For ResNet-20 with 2-bit activation, the training loss/accuracy results of vanilla ReLU is much worse than clipped ReLU, is there any explanation for this?  For the discussion in sec 4.2, what information does it want to convey?  What is the "normal schedule of learning rate"? What if the small learning rate 1e-5 is kept after 20 epochs?

Typo: The last sentence on page 3, the definition of y*.

------------------------

The author response have addressed most of my concerns. Thus I have increased my score.

---

> ### Author Response · Authors · 2018-11-18
> **Response to Reviewer 1**
>
> We thank the reviewer for the time and insightful comments.
>
>
> 1. While the paper presents many theoretical results which might be useful for the community, they are not organized very well. It is a bit hard for readers to quickly find the most important theoretical results. Moreover, some symbols are used without definition, e.g. g_{relu} is used before being defined in sec 3.1. The discussions for most theoretical results are very short or not organized well, making the whole paper hard to follow, e.g., "the key observation ..." after Lemma 4 is actually not about the Lemma 4 above, but Lemma 5 in the next Lemma.
>
> Reply: Thanks for your suggestion. In the revision, we improved the presentation of the paper, and we re-summarized our main contributions to make it clearer to the readers.
>
> 2. Another major concern is that activation quantization is usually used in combination with weight quantization. It would be more useful if weight and activation quantizations can be analyzed together.
>
> Reply: We did not use weight quantization because our main interest is study training with quantized activations, and because recent work has shown that weights can be quantized with little effect on performance (Hubara et al., 2016; Rastegari et al., 2016; Zhou et al., 2016).
>
> 3. Clarity in the experiment part can also be further improved. From Table 1, the clipped ReLU STE has the best performance, however, there is no theoretical analysis for it
> Reply: Thanks for your suggestion. We improved the presentation of the experiment section, and added the theoretical analysis of clipped RELU STE in the revision.
>
> 4. For ResNet-20 with 2-bit activation, the training loss/accuracy results of vanilla ReLU is much worse than clipped ReLU, is there any explanation for this?
> Reply: The explanation is that ReLU STE suffers the same instability issue (at good minima) as the identity STE does for 4-bit quantization. We added Figure 4 to demonstrate this point in Appendix C.
>
> 5. For the discussion in sec 4.2, what information does it want to convey?
> Reply: The discussion in sec 4.2 explains why the identity STE works poorly for ResNet-20 with 4-bit. This is because the training algorithm using identity STE simply can not converge to a good minimum. If it could converge well, then when we initialize the weights from the good minima achieved by vanilla ReLU STE or clipped ReLU STE and train the neural networks using a tiny learning rate of 1e-5, the algorithm should be stable there. But we observe that it is not stable and escapes from the good minima.
>
> 6. What is the "normal schedule of learning rate”?
> Reply: The normal schedule of learning rate is specified in Table 2 in appendix B.
>
> 7. What if the small learning rate 1e-5 is kept after 20 epochs?
> Reply: The behavior of the training algorithm using identity STE in the first 20 epochs already demonstrates its instability at good minima (both the classification error and training loss increases). After leaving the minima, keep using learning rate of 1e-5 will lead to extremely slow convergence (to a different point).
>
> 8. Typo: The last sentence on page 3, the definition of y*.
> Reply: This is not a typo. y* was defined separately from y at the beginning of section 2.2.

---

> > ### Comment · AnonReviewer1 · 2018-11-29
> > **Thanks for the response**
> >
> > Thanks for your detailed response. They have addressed most of my concerns, so I raised my score to 6. Still I am not sure how much gain we can benefit from quantizing activation only without weight quantization. Further work on  combination with weight quantization may make it even better.

---

> > > ### Author Response · Authors · 2018-11-29
> > > **Thank you**
> > >
> > > We thank the reviewer for the response and suggestion. We'll further work on  combination with weight quantization in the future.

---

### Official Review · AnonReviewer2 · 2018-11-01
**Interesting paper with some serious but fixable flaws**

**Rating:** 7
**Confidence:** 4

**Review:**

Summary:
The paper presents an analysis of training single-layer hard-threshold (binary activation) networks for regression with a mean-squared loss function using two different straight-through estimators: the original identity-function STE and a ReLU-based STE. The paper demonstrates that training with the latter against the population loss is guaranteed to converge to a critical point, whereas using the former can cause instability in the training.

    Pros:
        - Interesting analysis that provides a novel method for determining which gradient estimators are effective for training single-layer binarized networks and which are not.
        - The paper is fairly clear, despite being quite technical; however, I did find myself jumping around a lot to refer back to previous results or definitions so the ordering and layout could definitely be improved.

    Cons:
        - Related work is missing and some claims in the paper are wrong as a result.
        - A single-layer binarized network is essentially just a perceptron, which we know how to learn already, so it’s not clear how this analysis will benefit analysis of multi-layer binarized networks (however, since it seems like a novel analysis approach, it’s possible that it can be extended). This connection is not made in the paper.
        - The paper does not analyze the most common and successful straight-through estimator: the saturated straight-through estimator, which uses the derivative of the hard_tanh activation (e.g., see [2]) and is a shifted and scaled version of the clipped ReLU STE.

Overall, I like the paper but it has too many issues currently for me to give it a high score. However, if my questions and comments are addressed sufficiently, I would be happy to improve my score.


Detailed questions and comments:

1.	The claim that “we make the first theoretical justification for the concept of STE” is wrong and should be significantly toned down and clarified. Bengio et al. (2013), which this paper cites, provides some theoretical justification already, as do papers on target propagation, such as [1] and [2]. These, as well as additional papers cited in [2] are quite relevant and should also be cited and discussed.

2.	The claim that “it is not the gradient of any function” is also wrong. Each STE is the gradient of a particular function, but is not the gradient of the function used in the forward pass. Please clarify.

3.	The single-layer binarized network architecture studied in this paper can equivalently be framed as a linear function of a collection of single-layer perceptrons with shared weights. Obviously, much work has been done on analyzing the perceptron architecture. Why is none of it discussed in this paper? How does that work relate to the work done in this paper? How does the convolutional layer used here change the results of that related work?

4.	(a) Is there an intuition for why the derivative of the ReLU performs better (i.e., converges) better than the identity? Why does clipping the bottom make it work better? I do not see this explained in the text anywhere and it would be helpful to include this.
(b) Further, depending on the reasoning given, it seems that clipping the top may also be useful (as in the clipped ReLU, which is a shifted and scaled version of the saturated STE discussed in Hubara et al. and [2]). Does your analysis extend to this STE? This would be very useful, as the SSTE/clipped ReLU is the most commonly used STE and the most empirically successful (as validated by your own experiments, as well as in previous work on training binary networks). Also, the SSTE/clipped ReLU is an even better approximation of the step function.
(c) Cai et al. (2017) is not the first use of the clipped ReLU activation function, since it is equivalent to the SSTE when using sign(x) \in {-1, +1} instead of your activation function (\sigma(x) \in {0, 1}) (i.e., you can shift and scale everything to get equivalent results).

5.	In section 3.1, you mention that when using the derivative of the ReLU for the STE then \mu`(x) = \sigma(x). Is this just a coincidence or does this fact help with convergence?

6.	Why did you choose to train your networks initialized with the weights from their full-precision counterparts? When you train using different initializations, does this significantly affect your results?

7.	The improved empirical performance of the clipped ReLU / SSTE is unsurprising but why does the vanilla ReLU STE perform so poorly on CIFAR-10 with ResNet-20 with 2 bit quantization?

8.	In the end, it’s not clear that training single-layer hard-threshold networks is particularly important. Instead, the goal of quantization, etc. is to train multi-layer hard-threshold networks. Can this analysis be extended to such networks? Does it say anything about training such networks currently?

9.	The acknowledgments section is just the text from the style file.

10.	The capitalization is wrong in a number of places in your references.


[1] Difference Target Propagation. Lee, Zhang, Fischer, and Bengio. ECML/PKDD (2015).

[2] Deep Learning as a Mixed Convex-Combinatorial Optimization Problem. Friesen and Domingos. ICLR (2018).


------------------------

After reading the author response, they have sufficiently addressed my main concerns. I think that this is a good paper that will be of interest to those concerned with understanding the training of activation-quantized / hard-threshold neural networks. I have thus increased my score.

---

> ### Author Response · Authors · 2018-11-18
> **Response to Reviewer 2 (1/2)**
>
> We thank the reviewer for the time and constructive comments.
>
> 1. The claim that “we make the first theoretical justification for the concept of STE” is wrong and should be significantly toned down and clarified. Bengio et al. (2013), which this paper cites, provides some theoretical justification already, as do papers on target propagation, such as [1] and [2]. These, as well as additional papers cited in [2] are quite relevant and should also be cited and discussed.
>
> Reply: We agree that the claim is too strong, because the perceptron algorithm uses identity STE and has the convergence guarantee. In the revision, we include the discussions of [1] and [2] as they provide alternative ways for activation quantization. But we would like to point out that the theoretical justification in Bengio et al. (2013) is not for STE, instead it is for the stochastic neuron approach.
>
> 2. The claim that “it is not the gradient of any function” is also wrong. Each STE is the gradient of a particular function, but is not the gradient of the function used in the forward pass. Please clarify.
>
> Reply: Sorry for the confusion. We are not saying that STE is not the gradient of any function. STE is composited in the chain rule which computes the ‘gradient’ of loss function w.r.t. weight variables (which we call coarse gradient in the paper). This coarse gradient is not the gradient of any function including the loss function, because there is a mismatch between backward and forward passes. One main contribution of our paper is to understand why searching in the direction of negative coarse gradient (with proper STE) minimizes the loss function, since this is not the standard gradient descent.
>
> 3. The single-layer binarized network architecture studied in this paper can equivalently be framed as a linear function of a collection of single-layer perceptrons with shared weights. Obviously, much work has been done on analyzing the perceptron architecture. Why is none of it discussed in this paper? How does that work relate to the work done in this paper? How does the convolutional layer used here change the results of that related work?
>
> Reply: Thank you for pointing out the perceptron algorithm that we overlooked. We add the discussions of perceptron algorithm in the revision. It is the second *trainable* linear layer in our model that makes the analysis much more complicated. Our model is indeed a linear combination of a collection of perceptrons, but the way they are mixed is unknown.
>
> 4. (a) Is there an intuition for why the derivative of the ReLU performs better (i.e., converges) better than the identity? Why does clipping the bottom make it work better? I do not see this explained in the text anywhere and it would be helpful to include this.
> (b) Further, depending on the reasoning given, it seems that clipping the top may also be useful (as in the clipped ReLU, which is a shifted and scaled version of the saturated STE discussed in Hubara et al. and [2]). Does your analysis extend to this STE? This would be very useful, as the SSTE/clipped ReLU is the most commonly used STE and the most empirically successful (as validated by your own experiments, as well as in previous work on training binary networks). Also, the SSTE/clipped ReLU is an even better approximation of the step function.
> (c) Cai et al. (2017) is not the first use of the clipped ReLU activation function, since it is equivalent to the SSTE when using sign(x) \in {-1, +1} instead of your activation function (\sigma(x) \in {0, 1}) (i.e., you can shift and scale everything to get equivalent results).
>
> Reply: (a) We think the intuition is that clipped ReLU captures both the minimum and maximum of the original binary function. Or simply put, clipped ReLU is the closest approximation to the binarized ReLU.
>            (b) Thank you for your suggestion. Yes, our analysis now extends to the clipped STE. We added the analysis in the revision.
>            (c) We introduced SSTE as related work in the original paper, but we did not call it SSTE. We mentioned the name SSTE as well in the revision.
>
> 5. In section 3.1, you mention that when using the derivative of the ReLU for the STE then \mu`(x) = \sigma(x). Is this just a coincidence or does this fact help with convergence?
> Reply: We think this is just a coincidence.
>
> 6. Why did you choose to train your networks initialized with the weights from their full-precision counterparts? When you train using different initializations, does this significantly affect your results?
>
> Reply: Initializing the weights from full-precision counterparts is better than random initialization. The difference in the accurices can be noticeable sometimes (>1% on CIFAR-10).

---

> > ### Author Response · Authors · 2018-11-18
> > **Response to Reviewer 2 (2/2)**
> >
> > 7. The improved empirical performance of the clipped ReLU / SSTE is unsurprising but why does the vanilla ReLU STE perform so poorly on CIFAR-10 with ResNet-20 with 2 bit quantization?
> >
> > Reply: For ResNet-20 with 2 bit quantization, ReLU STE suffers the same instability issue (at good minima) as the identity STE does for 4-bit quantization. We added an experiment to demonstrate this in Appendix C. The reason why ReLU is not as good as clipped ReLU is that it does not match the quantized ReLU on the top part.
> >
> > 8. In the end, it’s not clear that training single-layer hard-threshold networks is particularly important. Instead, the goal of quantization, etc. is to train multi-layer hard-threshold networks. Can this analysis be extended to such networks? Does it say anything about training such networks currently?
> >
> > Reply: This is a good question. Theoretically, it is not straightforward to extend our analysis to multi-layer networks. This is the reason why we conduct experiments on LeNet-5, VGG and ResNet architectures in the paper, which complements the theoretical analysis. And in real experiments, we did observe the stability issue of identity STE reported in sec 4.2. This observation is consistent with our theoretical analysis of identity STE for the two-linear-layer model.
> >
> > 9. The acknowledgments section is just the text from the style file.
> > Reply: We will revise it after the decision is made.
> >
> > 10. The capitalization is wrong in a number of places in your references.
> > Reply: Thank you. We fixed them in the revision.

---

> ### Author Response · Authors · 2018-11-27
> **Thank you**
>
> We thank the reviewer for the kind comments.

---

### Official Review · AnonReviewer3 · 2018-11-02
**Interesting approach to correlate STE updates with true loss however implications are weak and assumptions are strong**

**Rating:** 7
**Confidence:** 4

**Review:**

The paper examines the use of STE for learning simple one-layer convolutional networks with binary activations and non overlapping patches. In this setting, the gradients are 0 almost everywhere (gradient of sign(x)) hence it is not clear how to use gradient descent. The approach studied here is to instead use the gradient of an alternative function such as ReLU or identity which is not always 0. The authors prove that if the ReLU's gradient is used then under gaussian distribution, the algorithm will converge to the local minimas/saddle points of the expected squared loss. they also show that the same does not hold for the identity's gradient.

The proof technique is interesting and the results do show the validity of the STE approach. The fact that the loss is provably monotonically decreasing is a strong validation. The paper is clearly written. However, I do have the following concerns/questions:
- The authors claim that their analysis is the first to analyze STE however I would like to point out that [1] studies the same setting (they allow overlapping patches and other distributions) with ReLU activation and show convergence guarantees to the global optima with using identity gradient instead of the ReLu gradient. Also for a single binary output case, using STE as identity equals the perceptron algorithm which is very well studied in literature.
- Restrictive setting: gaussian input, no label noise, non-overlapping architecture. Not clear what the motivation for this setting is. Also binary activations are rarely used in practice. Analysis also seems tied to the gaussian distribution.
- Infinite sample assumption is strong.
- No guarantees for convergence to the optimal solution unlike prior work.
- Assumptions on the weights being lower and upper bounded by a constant at each iteration seems strong unless an explicit projection step is used. Could the authors explain why this is a valid assumption to make?
- In the experimental section, momentum is used whereas it is not mentioned in the analysis. Does the STE perform well without the momentum? It is unclear why quantized ReLU is used.

[1] Surbhi Goel, Adam Klivans, and Raghu Meka. "Learning One Convolutional Layer with Overlapping Patches." ICML 2018.

----------
Apart from one concern (refer to comment), the authors have responded to most of my other comments. Based on this, I think the paper does offer an interesting analysis of the STE approach used for training binary networks, hence I'm increasing my score.

---

> ### Author Response · Authors · 2018-11-18
> **Response to Reviewer 3**
>
> We thank the reviewer for the time and insightful comments.
>
> 1. The authors claim that their analysis is the first to analyze STE however I would like to point out that [1] studies the same setting (they allow overlapping patches and other distributions) with ReLU activation and show convergence guarantees to the global optima with using identity gradient instead of the ReLu gradient. Also for a single binary output case, using STE as identity equals the perceptron algorithm which is very well studied in literature.
>
> Reply: Thank you for bringing to our attention the relevant Convertron paper [1] and perceptron algorithm which use the identity STE. We discussed them in details in the revision. We would like to point out that these two models (perceptron and Convertron) only have one trainable linear layer, whereas ours has two. So our model is more challenging to analyze. Moreover, as the reviewer pointed out, we proved that the loss is descending by using STE in the training, which is meaningful as people observe this in benchmark experiments. In addition, our framework allows us to analyze general STEs such as the derivatives of ReLU and clipped ReLU (in the revision). It is not clear if the analyses from the prior works can do the same thing.
>
> 2. Restrictive setting: gaussian input, no label noise, non-overlapping architecture. Not clear what the motivation for this setting is. Also binary activations are rarely used in practice. Analysis also seems tied to the gaussian distribution.
>
> Reply: We agree that our assumptions are stronger than that in [1]. But our model is more complicated, and [1] considers Leaky ReLU, not the binarzed ReLU which does not have a valid derivative. In light of the new analysis of clipped ReLU STE in the revised version, we believe the Gaussian distribution of input data can be relaxed into any rotation-invariant distribution. People use binarized or general quantized ReLU because this speed ups the prediction of DNNs at inference time, which promotes the energy efficiency.
>
> 3. Infinite sample assumption is strong.
>
> Reply: The reason why we consider infinite sample assumption is that we find the population loss function becomes Lipchitz smooth in this case, which is a surprising fact. To extend our results to the setting with finite training samples, one needs to use probabilistic tools such as concentration inequalities, and figure out the minimal number of samples in order to get a reasonably good solution quality. This requires much more additional technical efforts, and we think it is too much to include all these results in a single paper. Therefore, we plan to do this in our future work.
>
> 4. No guarantees for convergence to the optimal solution unlike prior work.
>
> Reply: In the revision, we prove that for proper initializations (Theorem 2 in the end of appendix), the convergence to global min is guaranteed. No guarantee for convergence to the optimal solution from *random* initialization is not due to the incompetence of the analysis, but because of the presence of spurious local min. The prior works with convergence guarantees to global min do not have a second *trainable* linear layer of the model like ours.
>
> 5. Assumptions on the weights being lower and upper bounded by a constant at each iteration seems strong unless an explicit projection step is used. Could the authors explain why this is a valid assumption to make?
>
> Reply: You are correct, one can use a projection step to avoid the boundedness assumption. For example, we can impose $w$ to be unit-normed (but there is no need to impose upper-bound on $v$ then). In real experiments, the weight vector is typically bounded and away from the zero.
> ----------------------------------------------------------------------------------------------------------------
> *Update*: we are now able to get rid of the upper bound assumption on $||v||$, and will revise it later. The lower bound imposed on $||w||$ is essential for the analysis, because it is the angle between $w$ and $w*$ that contributes to the loss function, and the minimization procedure has no control on $||w||$ itself.
>
> 6. In the experimental section, momentum is used whereas it is not mentioned in the analysis. Does the STE perform well without the momentum?
>
> Reply: This is a good point. Momentum is widely used in benchmarks, which accelerates the training. Just like regular gradient descent, STE needs the help of momentum to achieve the best empirical performance. We did not include it in the analysis because our main interest is study the correlation between the STE and loss function.
>
> 7. It is unclear why quantized ReLU is used.
>
> Reply: People care about quantized ReLU because it speeds up the prediction of DNNs at inference time. We refer the reviewer to the introduction section (the first paragraph) for the background of quantized DNNs.
>
> [1] Surbhi Goel, Adam Klivans, and Raghu Meka. "Learning One Convolutional Layer with Overlapping Patches." ICML 2018.

---

> > ### Comment · AnonReviewer3 · 2018-11-28
> > **Some remaining questions**
> >
> > Thanks for the detailed response to my comments. I've a few additional comments:
> > 1.  The algorithm name is not consistently used (Convotron/Convertron), please fix that. Also, the algorithm seems to work for ReLU as well as leaky ReLU though I agree that it is unclear if it would work for binary activation.
> > 2. For the projection step to ensure the lower bound on ||w||, it is not obvious that it will work. Firstly you are projecting on to a non-convex set, so you will have to guarantee that this step does not hurt your improvement so far.
> >
> > Point 2 still remains a concern for me, however, I am increasing my score based on clarification of most of the other comments.

---

> > > ### Author Response · Authors · 2018-11-28
> > > **Further response**
> > >
> > > We thank the reviewer for the feedback. The response to your new comments is as follows:
> > >
> > > 1. Thank you for pointing this out. We will fix the typos.
> > >
> > > 2. The analytic expression of the population loss function (in Lemma 1) only depends on the angle formed by w and w*, so the projection step (i.e., scaling ||w||) will not change the object value, since the angle is preserved after scaling.

---

### Public Comment · (anonymous) · 2018-10-02

---

### Author Response · Authors · 2018-11-18
**Revisions and Clarifications**

Dear reviewers,

Thank you for your constructive comments. We have revised our paper to discuss relevant references that we overlooked. We highlighted the major changes in red text. The other major changes include

1. We revised the summary of contributions, and compared our work with the prior works using identity STE (the perceptron algorithm (Rosenblatt 1958) and Convertron algorithm (Goel et al. 2018).)

2. We included the analysis of clipped ReLU STE as suggested by Reviewers 1&2 (Lemmas 7&8 and Theorem 1).

3. We proved the convergence to the true weights (global min) using vanilla and clipped ReLU STE with proper initialization (Theorem 2 in the end of Appendix). For random initialization, this is not guaranteed because there exist spurious local min.

Hereby we would like to make some clarifications on the contributions of our paper, since Reviewers 2 & 3 have raised the points that the prior works perceptron algorithm and Convertron algorithm use an identity STE and also have theoretical guarantees.

1. Our model has a second *trainable* linear layer, which results in a more complicated loss function and landscape. In contrast, Perceptron has one linear layer with binary output. While also called one-hidden-layer network, Convertron considers one trainable layer, and the weights in the second linear layer are known and fixed to be 1. Moreover, Convertron deals with Leaky ReLU activation, not the binarized ReLU in the quantization setting.

2. Both perceptron and Convertron algorithms use identity STE. While the identity STE works well for networks with one trainable linear layer, we theoretically prove that identity STE is not good for training two-linear-layer networks, and empirically demonstrate that it is not good for benchmark classifications with quantized ReLU either.

3. We are the first to analyze practically more useful STEs such as derivatives of vanilla and clipped ReLUs (in the revised paper as suggested by Reviewers 1&2). We are the first to prove the descent property of coarse gradient descent associated with these STEs. We discover the instability issue in the training using identity STE in theoretical analysis, which is also observed in our CIFAR-10 4-bit activation experiments reported in section 4.2.

In light of our responses to the reviewers' concerns, we would be very grateful if you would look over our paper again, and reconsider your opinion. We believe our work proposes a novel theoretical framework to analyze general STE, and provides a deeper understanding towards the use of STE in training activation quantized DNNs.

---

### Meta-Review · Area_Chair1 · 2018-12-17
**Progress on the theoretical understanding of straight-through estimation for linear networks**

**Confidence:** 5
**Recommendation:** Accept (Poster)

**Metareview:**

The paper contributes to the understanding of straight-through estimation for single hidden layer neural networks, revealing advantages for ReLU and clipped ReLU over identity activations.  A thorough and convincing theoretical analysis is provided to support these findings.  After resolving various issues during the response period, the reviewers concluded with a unanimous recommendation of acceptance.  Valid criticisms of the presentation quality were raised during the review and response period, and the authors would be well served by continuing to improve the paper's clarity.